

# Temporal evolution of chlorine and minor species related to ozone depletion observed with ground-based FTIR at Syowa Station, Antarctica and satellites during austral fall to spring in 2007 and 2011

Hideaki Nakajima[1,2], Isao Murata[2], Yoshihiro Nagahama[1], Hideharu Akiyoshi[1], Kosuke Saeki[2,3], Masanori Takeda[2], Yoshihiro Tomikawa[4,5], and Nicholas B. Jones[6]

[1]National Institute for Environmental Studies, Tsukuba, Ibaraki, 305-8506, Japan
[2]Graduate School of Environmental Studies, Tohoku University, Sendai, Miyagi, 980-8572, Japan
[3]now at Weathernews Inc., Chiba, 261-0023, Japan
[4]National Institute of Polar Research, Tachikawa, Tokyo, 190-8518, Japan
[5]The Graduate University for Advanced Studies, Tachikawa, Tokyo, 190-8518, Japan
[6]University of Wollongong, Wollongong, New South Wales, 2522, Australia

*Correspondence to*: Hideaki Nakajima (nakajima@nies.go.jp)

**Abstract.**

To understand and project future ozone recovery, understanding of mechanisms related to polar ozone destruction is crucial. For polar stratospheric ozone destruction, chlorine species play an important role, but detailed temporal evolution of chlorine species in the Antarctic winter is not well understood. We retrieved lower stratospheric vertical profiles of $O_3$, $HNO_3$, and HCl from solar spectra taken with a ground-based Fourier-Transform infrared spectrometer (FTIR) installed at Syowa Station, Antarctica (69.0ºS, 39.6ºE) from March to December 2007 and September to November 2011. We analyzed temporal variation of these species combined with ClO, HCl, and $HNO_3$ data taken with the Aura/MLS (Microwave Limb Sounder) satellite sensor, and $ClONO_2$ data taken with the Envisat/MIPAS (The Michelson Interferometer for Passive Atmospheric Sounding) satellite sensor at 18 and 22 km over Syowa Station. When the stratospheric temperature over Syowa Station fell below polar stratospheric cloud (PSC) saturation temperature in early winter, PSCs started to form and heterogeneous reaction on PSCs convert chlorine reservoirs into reactive chemical species. HCl and $ClONO_2$ decrease occurred at both 18 and 22 km, and soon $ClONO_2$ was almost depleted in early winter. When the sun returned to Antarctica in spring, enhancement of ClO and gradual $O_3$ destruction were observed. During the ClO enhanced period, negative correlation between ClO and $ClONO_2$ was observed in the time-series of the data at Syowa Station. This negative correlation was associated with the distance between Syowa Station and the inner edge of the polar vortex. Temporal variation of chlorine species over Syowa Station was affected by both heterogeneous chemistry related to PSC occurrence deep inside the polar vortex, and transport of an $NO_x$-rich airmass from lower latitudinal polar vortex boundary region which can produce additional $ClONO_2$ by reaction between ClO and $NO_2$. We used MIROC3.2 Chemistry-Climate Model (CCM) results to see the comprehensive behavior of chlorine and related species inside the polar vortex and the edge region in more detail. Rapid conversion of chlorine reservoir species (HCl and $ClONO_2$) into $Cl_2$, gradual conversion of $Cl_2$ into $Cl_2O_2$, increase of ClO when sunlight became available, and conversion of



ClO into HCl, was successfully reproduced by the CCM. HCl decrease in the winter polar vortex core continued to occur due to the transport of $ClONO_2$ from the subpolar region (55-65ºS) to higher latitudes (65-75ºS), providing a flux of $ClONO_2$ from more sunlit latitudes into the polar vortex. The deactivation pathways from active ClO into reservoir species (HCl and/or $ClONO_2$) were found to be highly dependent on the availability of ambient $O_3$ and $NO_x$. At an altitude where most ozone was

depleted in Antarctica, most ClO was converted to HCl. However, when there were some $O_3$ and $NO_x$ available, super-recovery of $ClONO_2$ can occur, similar to the case in the Arctic.

## 1. Introduction

Discussion of the detection of "recovery" of the Antarctic ozone hole as the result of CFC restrictions has been attracting

attention. The occurrence of the Antarctic ozone hole is considered to continue at least until the middle of this century. The world's leading Chemistry-Climate Models (CCMs) mean time series indicate that the springtime Antarctic total column ozone will return to 1980 levels between 2045 and 2060 (WMO, 2014). In fact, the recovery time predicted by CCMs has large uncertainty, and observed ozone hole magnitude also shows great year-to-year variability. Although Solomon et al. (2016) and de Laat et al. (2017) reported signs of healing in the Antarctic ozone layer only in September month, there is no statistically

conclusive report on the Antarctic ozone hole recovery (Yang et al., 2008; Kuttippurath et al, 2010).

  To understand ozone depletion processes in polar regions, understanding of the behavior and partitioning of active chlorines ($ClO_x=Cl+Cl_2+ClO+ClOO+Cl_2O_2+HOCl+ClNO_2$) and inert chlorines (HCl and $ClONO_2$) are crucial. Inert chlorine is converted to active chlorine that destroys ozone on polar stratospheric clouds (PSCs) through heterogeneous reactions:

$$ClONO_2 \text{ (g)} + HCl \text{ (s, l)} \rightarrow Cl_2 \text{ (g)} + HNO_3 \qquad (1)$$

$$ClONO_2 \text{ (g)} + H_2O \text{ (l, s)} \rightarrow HOCl \text{ (g)} + HNO_3 \qquad (2)$$

where g, s, and l represents the gas, solid, and liquid phases, respectively.

  Heterogeneous reactions on the surface of particles;

$$N_2O_5 \text{ (g)} + HCl \text{ (s, l)} \rightarrow ClNO_2 \text{ (g)} + HNO_3 \qquad (3)$$

$$HOCl \text{ (g)} + HCl \text{ (s, l)} \rightarrow Cl_2 \text{ (g)} + H_2O \qquad (4)$$

are responsible for additional chlorine activation. When solar illumination is available, $Cl_2$, HOCl, and $ClNO_2$ are photolyzed to produce chlorine atoms by reactions:

$$Cl_2 + hv \rightarrow Cl + Cl \qquad (5)$$

$$HOCl + hv \rightarrow Cl + OH \qquad (6)$$

$$ClNO_2 + hv \rightarrow Cl + NO_2. \qquad (7)$$

The yielded chlorine atoms then start to destroy ozone catalytically through reactions:

$$Cl + O_3 \rightarrow ClO + O_2 \qquad (8)$$

$$ClO + ClO + M \rightarrow Cl_2O_2 + M \qquad (9)$$



$$Cl_2O_2 + h\nu \rightarrow Cl + ClOO \qquad\qquad (10)$$

$$ClOO + M \rightarrow Cl + O_2 + M. \qquad\qquad (11)$$

When the stratospheric temperature gets warmer, and no PSCs are present, gradual inactivation of chlorine starts to occur. Re-formation of $ClONO_2$ and HCl mainly occur through reactions:

$$ClO + NO_2 + M \rightarrow ClONO_2 + M \qquad\qquad (12)$$

$$Cl + CH_4 \rightarrow HCl + CH_3. \qquad\qquad (13)$$

The re-formation of $ClONO_2$ by reaction (12) from active chlorine is much faster than that of HCl by reaction (13), if there are enough $NO_x$ around (Mellqvist et al., 2002; Dufour et al., 2006).

The processes of inactivation of active chlorine are different between in the Arctic and Antarctic. In the Antarctic, the temperature cools below the PSC formation threshold in the whole area of the polar vortex in most years, and almost complete denitrification and chlorine activation occur (WMO, 2007), followed by severe ozone depletion in spring. In the chlorine reservoir recovery phase, HCl is mainly formed by reaction (13) due to the following reasons: The lack of ozone prevents the formation of ClO via reaction (8), and the lack of $NO_2$ due to denitrification prevents the formation of $ClONO_2$ by reaction (12).

On the other hand, in the Arctic, PSC formation occurs in the limited area of the polar vortex due to higher stratospheric temperatures (~10-15K in average) compared with that of Antarctica, and only partial denitrification and chlorine activation occur (Manney et al., 2011; WMO, 2014). In this case, some ozone, ClO, and $NO_2$ are available in the chlorine reservoir recovery phase. Therefore, the $ClONO_2$ amount becomes sometimes higher than that of HCl after PSCs have disappeared due to the rapid reaction in equation (12) (Michelsen et al., 1999; Santee et al., 2003), which results in super-recovery of $ClONO_2$ at the time of chlorine deactivation in spring (Webster et al., 1993). In this way, the distribution ratio of inert chlorine in springtime is related to temperature, PSC amounts, ozone, and $NO_2$ concentrations (Santee et al., 2008; Solomon et al., 2015).

In the polar regions, the ozone and related atmospheric trace components have been monitored since the discovery of the ozone hole. They consist of direct observation by high-altitude aircraft observations (Anderson et al., 1989; Ko et al., 1989; Bonne et al., 2000, etc.), remote sensing observations by satellites (Müller et al., 1996; Michelsen et al., 1999; Höpfner et al., 2004; Dufour et al., 2006; Hayashida et al., 2007, etc.), remote sensing observations from the ground (Farmer et al., 1987; Kreher et al., 1996; Mellqvist et al., 2002; Blumenstock et al., 2006; etc.). Within these observations, ground-based measurements have the characteristic of high temporal resolution. In addition, the Fourier-Transform infrared spectrometer (FTIR) has the capability of measuring several trace components at the same time or in a short time interval. In this paper, we show the results of ground-based FTIR observations of $O_3$ and other trace species at Syowa Station in the Antarctic in 2007 and 2011, combined with the satellite measurements of trace species by Aura/MLS and Envisat/MIPAS, to show the temporal variation and partitioning of active chlorine ($ClO_x$) and chlorine reservoirs (HCl, $ClONO_2$) from fall to spring in the course of the ozone hole formation and dissipation. Finally, simulated distributions of minor species by a MIROC3.2 chemistry-climate model are used to further discuss the behavior of active and inert chlorine species.



## 2. Measurements

The Japanese Antarctic Syowa Station (69.0ºS, 39.6ºE) was established in January 1957. Since then, several scientific observations related to meteorology, upper atmospheric physics, glaciology, biology, geology, seismology, etc. have been performed. The ozone hole was first detected by Dobson spectrometer and ozonesonde measurements from Syowa Station in 1982 (Chubachi, 1984). We installed a Bruker IFS-120M high-resolution Fourier-Transform infrared spectrometer (FTIR) in the Observation Hut at Syowa Station in March 2007. This was the second high-resolution FTIR site in Antarctica in operation after New Zealand's Arrival Heights facility at Scott Station (77.8°S, 166.7°E) (Wood et al., 2002; Wood et al., 2004). The IFS-120M FTIR has a wavenumber resolution of 0.0035 cm$^{-1}$, with two liquid nitrogen cooled detectors (InSb and HgCdTe covering the frequency ranges 2000-5000 and 700-1300 cm$^{-1}$, respectively), and fed by an external solar tracking system. Since Syowa Station is located at a relatively low latitude (69.0°S) compared with Scott Station (77.8°S), there is an advantage of the short (about one month) polar night period, when we cannot measure atmospheric species using the sun as a light source. Since we can resume FTIR measurements from early spring (late July), we can measure chemical species during ozone hole development. Another advantage of Syowa Station is that we can measure both inside and outside of the polar vortex, since the station is sometimes located near the edge of the polar vortex. From March to December 2007, we made 78 days of FTIR measurements in total. Another 19 days of FTIR measurements were performed from September to November 2011. Table 1 shows days when FTIR measurements were made at Syowa Station with the information inside/boundary/outside of the polar vortex defined by the method described in Section 4 using ERA-Interim reanalysis data.

The analysis of the FTIR spectra was done with SFIT2 Version 3.92 program (Rinsland et al., 1998; Hase et al., 2004). SFIT2 retrieves a vertical profile of trace gases using an optimal estimation formulation of Rodgers (2000), implemented with a semi-empirical method which was originally developed for microwave measurements (Parrish et al., 1992; Connor et al., 1995). The SFIT2 forward model fully describes the FTIR instrument response, with absorption coefficients calculated using the algorithm of Norton and Rinsland (1991). The atmosphere is constructed with 30 or more layers from the ground to 100 km, using the FASATM (Gallery et al., 1983) program for atmospheric ray-tracing to account for refractive bending. The retrieval parameters for each gas are shown in Table 2. Temperature and pressure profiles between 0 and 30 km are taken by the Rawin sonde observations flown from Syowa Station on the same day by the Japanese Meteorological Agency (JMA), while values between 30 and 100 km are taken from the CIRA-86 standard atmosphere profile (Rees et al., 1990).

We retrieved vertical profiles of $O_3$, HCl, and $HNO_3$ from the solar spectra. We used monthly averaged ozonesondes profiles (0-30 km) and ILAS-II (Nakajima, 2006; Sugita et al., 2006) profiles (30-100 km) for the a priori of $O_3$, monthly averaged profiles from ILAS-II for $HNO_3$ and monthly averaged profiles from HALOE (Anderson et al., 2000) for HCl. Typical averaging kernels of the SFIT2 retrievals for $O_3$, $HNO_3$, and HCl are shown in Figures 1(a), 1(b), and 1(c), respectively.

ClO profiles were taken from Aura/MLS version 3.3 data (Liversey et al., 2006; Santee et al., 2011; Ziemke et al., 2011; Liversey et al., 2013) whose measurement location is within 300 km radius from Syowa Station and within ±6 hours of the FTIR measurement. $ClONO_2$ profiles were taken from Envisat/MIPAS IMK/IAA version V5R_CLONO2_220 and





V5R_CLONO2_222 (Höpfner et al., 2007), whose measurement criteria are the same as that of Aura/MLS. In order to monitor the appearance of polar stratospheric clouds (PSCs) over Syowa Station, we used CALIPSO PSC data (Pitts et al., 2007; 2009; 2011).

## 3. Validation of retrieved profiles from FTIR spectra with other measurements

We validated retrieved FTIR profiles of $O_3$ with ozonesondes, and $HNO_3$ and HCl with Aura/MLS version 3.3 data (Liversey et al., 2013) for 2007 measurements. We identified the nearest Aura/MLS data from the distance between the Aura/MLS tangent point at 20 km altitude and the point at 20 km altitude for the direction of the sun from Syowa Station at the time of the FTIR measurement. The spatial and temporal collocation criteria used was within 300 km radius and ±6 hours. The ozonesonde and Aura/MLS profiles were interpolated onto a 1 km-grid, then smoothed with a 5 km-wide slit function.

Figures 2(a) and 2(b) show absolute and percentage differences of $O_3$ profiles retrieved from FTIR measurements and those from ozonesonde measurements, respectively, calculated from 14 coincident measurements from September 5 to December 17, 2007. We define the percentage difference D as:

$$D (\%) = 100 * (FTIR-sonde) / ((FTIR+sonde)/2). \qquad (14)$$

The absolute agreement between 15 and 25 km was within -0.02 to 0.40 ppmv. The D value between 15 and 25 km was within -10.4 to +24.4%. The mean D value of $O_3$ for the altitude of interest in this study (18-22 km) was +6.1%, with the minimum of -10.4% and the maximum of +19.2%. FTIR data agree with validation data within error bars at the altitude of interest. Note that relatively large D values between 16 and 18 km are due to small ozone amount in the ozone hole.

Figures 3(a) and 3(b) show absolute and percentage differences of $HNO_3$ profiles retrieved by FTIR measurements and those from Aura/MLS measurements, respectively, calculated from 44 coincident measurements. The agreement between 15 and 25 km was within -0.56 to +0.57 ppb. The D value between 15 and 25 km was within -25.5 to +21.9%. The mean D value for $HNO_3$ for the altitude of interest in this study (18-22 km) was +13.2%, with the minimum of +0.2% and the maximum of +21.9%. However, this positive bias of FTIR data is still within the error bars of FTIR measurements. Livesey et al. (2013) showed that Aura/MLS version 3.3 data has no bias within errors (~0.6-0.7 ppbv at pressure level of 100-3.2 hPa) compared with other measurements.

Figures 4(a) and 4(b) show absolute and percentage differences of HCl profiles retrieved by FTIR measurements and those from Aura/MLS measurements, respectively, calculated from 47 coincident measurements. The agreement between 15 and 25 km was within -0.20 to -0.09 ppbv. The D value between 15 and 25 km was within -34.1 to -3.0%. The mean D value for HCl for the altitude of interest in this study (18-22 km) is -9.7%, with the minimum of -14.6% and the maximum of -3.0%. However, this negative bias of FTIR data is still within the error bars of FTIR measurements. Moreover, Livesey et al. (2013) showed Aura/MLS version 3.3 values are systematically greater than HALOE values by 10-15% with a precision of 0.2-0.6% in the stratosphere, which may partly explain the negative bias of FTIR data compared with MLS data.



Table 3 summarizes validation results of FTIR profiles compared with ozonesonde or Aura/MLS measurements, and possible Aura/MLS biases from literature.

## 4. Results

Figures 5(a) and 5(b) show temporal variations of temperature at 18 and 22 km over Syowa Station using ERA-Interim data (Dee et al., 2011) for 2007 and 2011, respectively. Approximate saturation temperatures for nitric acid trihydrate (NAT) PSC ($T_{NAT}$) and ice PSC ($T_{ICE}$) are also shown on the figures. The dates when PSCs were observed at Syowa Station were identified by CALIPSO data, and indicated by asterisks on the bottom of the figures. Over Syowa Station, PSCs were observed at 15-25 km from the beginning of July (day 183) to the middle of September (day 253) in 2007, and from late June (day 175) to early September (day 251) in 2011.

PSCs were observed only below 20 km after mid-August, due to the sedimentation of PSCs and downwelling of vortex air in late winter. Although temperature above Syowa Station are sometimes below $T_{NAT}$ in June and in late September, no PSC was observed during those periods. This is due to other reason, such as a different time history of temperature for PSC formation, and/or low $HNO_3$ (denitrification) and/or $H_2O$ concentration (dehydration) which are needed for PSC formation in late winter season (Saitoh et al., 2006).

Figures 6, 7, 8, and 9 show temporal variations of HCl, $ClONO_2$, ClO, $Cl_y^*$, $O_3$, and $HNO_3$ over Syowa Station in 2007 and 2011 at altitudes of 18 and 22 km, respectively. $O_3$ (sonde) is observed with the KC96 ozonesonde for 2007 and the ECC-1Z ozonesonde for 2011 by JMA. HCl (MLS) and $HNO_3$ (MLS) observed by Aura/MLS are plotted to complement the data lack of FTIR measurements. $ClONO_2$ is observed by Envisat/MIPAS. Total inorganic chlorine $Cl_y^*$ corresponds to the sum of HCl, $ClONO_2$, and $Cl_x$, where active chlorine species $Cl_x$ is defined as the sum of ClO, Cl, and $2*Cl_2O_2$ (Bonne et al., 2000). Inferred total inorganic chlorine $Cl_y^*$ is calculated from $N_2O$ value (in ppbv) measured by MLS using the polynomial correlation (Bonne et al., 2000) of;

$$Cl_y^*(pptv) = 4.7070*10^{-7}(N_2O)^4 - 3.2708*10^{-4}(N_2O)^3 + 4.0818*10^{-2}(N_2O)^2 - 4.6856(N_2O) + 3225. \qquad (15)$$

Dark shaded and thin shaded days indicate that Syowa station was located outside and in the boundary region of the polar vortex, respectively. Inner and outer edges of the polar vortex were determined as follows:

1) Equivalent latitudes (McIntyre and Palmer, 1984; Butchart and Remsberg, 1986) were computed based on isentropic potential vorticity at 450 K and 560 K isentropic surfaces for 18 km and 22 km using the ERA-Interim reanalysis data (Dee et al., 2011), respectively.

2) Inner and outer edges (at least 5° apart from each other) of the polar vortex were defined by local maxima of the isentropic potential vorticity gradient with respect to equivalent latitude only when a tangential wind speed (i.e., mean horizontal wind speed along the isentropic potential vorticity contour; see Eq. (1) of Tomikawa and Sato (2003)) near the vortex edge exceeds a threshold value (i.e., 20 m s$^{-1}$, see Nash et al. (1996) and Tomikawa et al. (2015)).



3) Then, the polar region is categorized into three categories; i.e., inside the polar vortex (inside of inner edge), the boundary region (between inner and outer edges), and outside the polar vortex (outside of outer edge).

Hereafter, we will discuss the results only when Syowa station was located inside the polar vortex. Note that the Syowa station is often located near the vortex edge and the temporal variations observed over Syowa station sometimes reflect spatial variations, not the chemical evolution. The lack of data for ClO and HCl (MLS) from day 195 to day 219, 2007 and ClONO$_2$ from day 170 to day 216, 2007 (upper panels of Figures 6(a) and 8(a)) is due to large error in Aura/MLS measurements during this period.

Characteristics of variations in each minor species for 2007 and 2011 at altitudes of 18 and 22 km are summarized in Tables 4 (ClO), 5 (HCl), 6 (ClONO$_2$), 7 (Cl$_y^*$), 8 (O$_3$), and 9 (HNO$_3$). In these tables, typical values (mixing ratios), approximate starting and ending days of decrease, variation amounts, etc. are summarized for each species.

The common features found in both 2007 and 2011 at both altitudes of 18 and 22 km can be summarized as follows: ClO is enhanced in August and September and the day-to-day variations were large over this period. HCl was almost zero from late June to early September and the day-to-day variations were small over this period (larger values are related to the polar vortex boundary). HCl and ClONO$_2$ decreased first, then ClO started to increase in winter, while HCl increases and ClO decreases were synchronized in spring. Cl$_y^*$ gradually increased in the polar vortex from late autumn to spring. The Cl$_y^*$ value became larger compared with its mixing ratio outside of the polar vortex in spring. O$_3$ decreased from July to late September when ClO was present. HNO$_3$ showed large decreases from June to July, and then gradually increased in summer. Day-to-day variations of HNO$_3$ from June to August were large.

The following characteristics are evident especially at 18 km (Figures 6 and 7). The day-to-day variations of HCl from late June to early September were as small as 0-0.3 ppb. The recovered values of HCl in spring were larger than those before winter and those outside the polar vortex during the same period. ClONO$_2$ kept near zero even after ClO disappeared, and did not recover to the level before winter until spring. O$_3$ gradually decreased from values of 2.5-3 ppm before winter to values less than one fifth, 0.3-0.5 ppm, in October.

The following characteristics are evident only at 22 km (Figures 8 and 9). The day-to-day variation of HCl from late June to early September were 0-1 ppb, larger than those at 18 km. The recovered values of HCl in spring were nearly the same as those before winter (around 2.2 ppb). ClONO$_2$ recovered to larger values than those before winter after ClO disappeared. From winter to spring, O$_3$ gradually decreased, but the magnitude of the decrease was much smaller than that at 18 km.

As for the temporal increase of ClONO$_2$ in spring during the ClO decreasing phase, we can see a peak of 1.5 ppb at 18 km in 2011, and at 22 km in both 2007 and 2011 around day 270, but we see no temporal increase of ClONO$_2$ at 18 km in 2007.

Figure 9 shows that temporal ClO enhancement and decrease of O$_3$, ClONO$_2$, and HNO$_3$ occurred in early winter (day 150-170) at 22 km in 2011. This small ozone depletion event before winter might be due to an airmass movement from the polar night area to a sunlit area at lower latitudes.



## 5. Discussion

The ratios of observed HCl, ClONO$_2$, ClO, and Cl$_y$ to Cl$_y$* were calculated to discuss the temporal variations of the chlorine partitioning. Here, observed Cl$_y$ is determined as:

Cl$_y$ = HCl + ClONO$_2$ + ClO.

Figures 10 and 11 show temporal variations of the ratios of each species to Cl$_y$* in 2007 (a) and in 2011 (b) at 18 km and 22 km, respectively. For both in 2007 and 2011 at 18 km (Figure 10), the ratio of HCl was 0.6-0.8 and the ratio of ClONO$_2$ was 0.2-0.3 before winter (day 130-140). The partitioning of HCl was three times larger than that of ClONO$_2$ at that time. The ratio of ClO increased to 0.5-0.6 during the enhanced period (day 240-260). The ratio of HCl was 0-0.2 and the ratio of ClONO$_2$ was 0-0.6 during this same period. ClONO$_2$ shows negative correlation with ClO, while HCl kept low even when ClO was low during this period. This negative correlation is shown in Figure 12 later. When ClO was enhanced, the O$_3$ amount gradually decreased, and finally reached <0.5 ppmv (>80% destruction) in October (day 280) (See Figures 6 and 7). The ratios became 0.9-1.0 for HCl and 0-0.1 for ClONO$_2$ after the recovery in spring (after day 290), indicating that almost all chlorine reservoir species became HCl via reaction (13), due to the lack of O$_3$ and NO$_2$ during this period.

For both in 2007 and 2011 at 22 km (Figure 11), the ratio of HCl was 0.4-0.9 and the ratio of ClONO$_2$ was 0.2-0.3 before winter (day 110-140). The partitioning of HCl was two to three times larger than that of ClONO$_2$. The ratio of ClO increased to 0.6-0.7 during the enhanced period (day 220-240). The ratio of HCl was 0-0.3 and the ratio of ClONO$_2$ was 0-0.6 during this period. ClONO$_2$ shows negative correlation with ClO, while HCl kept low even when ClO was low during this period like the case at 18 km. The O$_3$ amount gradually decreased during the ClO enhanced period, but remained >1.5 ppmv (less than half destruction) at this altitude (See Figures 8 and 9). When the ClO enhancement ended, temporal increase (super-recovery) of ClONO$_2$ up to a ratio of 0.5 occurred in early spring (day 260-280). Then, the reservoir ratios became 0.6-0.8 for HCl and 0.2-0.4 for ClONO$_2$ in spring (after day 280). This phenomenon shows that more chlorine deactivation via reaction (12) occurred towards ClONO$_2$ at 22 km rather than at 18km. This is attributed to the existence of O$_3$ and NO$_2$ during this period at 22 km, which was different from the case at 18 km.

In 2011 at 18 km (Figure 10), another temporal increase of ClONO$_2$ up to a ratio of 0.6 occurred in early spring (around day 280) in accordance with HCl increase, then ClONO$_2$ amount gradually decreased to nearly zero after late October (day 300-). This temporal increase in ClONO$_2$ could be attributed to temporal change of the location of Syowa Station in the polar vortex. Although Syowa Station was always located inside the polar vortex from day 195 to 350, the difference between the equivalent latitude over Syowa Station and that at inner edge became less than 10 degrees at around day 280, while it was typically between 15 and 20 degrees in other days. O$_3$ and HNO$_3$ showed higher values around day 280 (see Figure 7), indicating that Syowa Station was located close to the boundary region at this period. Therefore, the temporal increase of ClONO$_2$ in 2011 at 18 km was attributed to spatial variation, not to chemical evolution.

Figure 12 show the correlation between ClO and ClONO$_2$ during the ClO enhanced period (day 220-260) at 18 km in 2007 (a) and 2011 (b), and at 22 km in 2007 (c) and 2011 (d). Solid lines show regression lines obtained by RMA (Reduced Major





Axis) regression. Negative correlations between ClO and ClONO$_2$ are seen in all figures. The cause of this negative correlation might be due to the variation of the relative distance between Syowa Station and the boundary region of the polar vortex. When Syowa Station was located deep inside the polar vortex, there was more ClO and less ClONO$_2$. On the contrary when Syowa Station was located near the vortex edge, there was less ClO and more ClONO$_2$. The potential vorticities (PV) over

Syowa Station shown by color code generally show this tendency, that warm colored higher PV points are located more towards bottom right-hand side. This is further confirmed by 3D model simulation as is shown later in this section.

Figures 13, 14, 15, and 16 show simulated mixing ratios of O$_3$ (a), NO$_2$ (b), HNO$_3$ (c), ClO (d), HCl (e), and ClONO$_2$ (f) by the MIROC3.2 Chemistry-Climate Model (CCM) at 50 hPa (~18 km) for June 24 (day 175), September 1 (day 244), September 6 (day 249), and October 6 (day 279) in 2007, respectively. For a description of the MIROC3.2 CCM, please see Appendix

A for detail. The location of Syowa Station is shown by a white star in each panel. Direct comparisons of mixing ratios of ClO, HCl, ClONO$_2$, Cl$_y$, and O$_3$ measured by FTIR and MLS, and modeled by MIROC3.2 CCM in 2007 and 2011 at 18 and 22 km are shown in Appendix B. In general, the model results are in good agreement with FTIR and satellite observations (Figure B1).

On June 24 (day 175, Figure 13), stratospheric temperatures over Antarctica were already low enough to allow PSCs to

form. Consequently, NO$_2$ and HNO$_3$ in the polar vortex condensed onto PSCs (Figures 13(b) and 13(c)). Note that the depleted area of NO$_2$ was greater than that of HNO$_3$. This might be due to reaction (12) that converts ClO and NO$_2$ to ClONO$_2$ at the edge of the polar vortex, which is shown by the enhanced ClONO$_2$ area at the vortex edge in Figure 13(f). Also, HCl and ClONO$_2$ are depleted in the polar vortex due to the heterogeneous reactions (1), (2), (3), and (4) on the surface of PSCs and aerosols (Figures 13(e) and 13(f)). Some HCl remains near the core of the polar vortex (Figure 13(e)), because the initial

amount of the counter-part of heterogeneous reaction (1) (ClONO$_2$) was less than that of HCl (see Figure 6 and/or 7). The O$_3$ amount was only slightly depleted within the polar vortex on this day (Figure 13(a)).

On September 1 (day 244, Figure 14), amounts of NO$_2$ (Figure 14(b)), HNO$_3$ (Figure 14(c)), HCl (Figure 14(e)), and ClONO$_2$ (Figure 14(f)) all show very depleted values in the polar vortex. The amount of ClO (Figure 14(d)) shows some enhanced values at the outer part of the polar vortex. About 50% ozone depletion was seen throughout the polar vortex. Note that

ClONO$_2$ (Figure 14(f)) shows enhanced values around the boundary region of the polar vortex. This might be due to the reaction (12) at this location. On this day (day 244), Syowa Station was located inside the polar vortex close to the inner vortex edge, where ClO was smaller and ClONO$_2$ was greater than the values deep inside the polar vortex.

On September 6 (day 249, Figure 15), most features were the same as on September 1, but the shape of the polar vortex was different. Consequently, Syowa Station was located deep inside the polar vortex, where ClO was greater and ClONO$_2$ was

smaller than the values around the boundary region of the polar vortex. Hence, the negative correlation between ClO and ClONO$_2$ seen in Figure 12 was due to variation of the relative distance between Syowa Station and the inner edge of the polar vortex.

On October 6 (day 279, Figure 16), ClO enhancement has almost disappeared (Figure 16(d)). Inside the polar vortex, depletion of O$_3$ (Figure 16(a)), NO$_2$ (Figure 16(b)), HNO$_3$ (Figure 16(c)), and ClONO$_2$ (Figure 16(f)) continued. Ozone was



almost fully destroyed at this altitude in the polar vortex. However, the amount of HCl (Figure 16(e)) increased deep inside the polar vortex. This might be due to the recovery of HCl by reaction (13) deep inside the polar vortex, where there is no $O_3$ or $NO_2$ left and reaction (13) was favored compared with reaction (12). At Syowa Station, the amount of HCl was several times greater than that of $ClONO_2$ on this day.

Three-hourly temporal evolution of zonal-mean active chlorine species, $Cl_2O_2$ (b), $Cl_2$ (c), ClO (d), and their sum (ClO+2*$Cl_2O_2$+2*$Cl_2$) (a), and chlorine reservoir species HCl (e) and $ClONO_2$ (f) modeled by MIROC3.2 CCM at 68.4ºS, 71.2ºS, 76.7ºS, and 87.9ºS are plotted in Figure 17. The dates on which the distribution of each species is shown in Figures 13-16, are indicated by vertical dotted lines. In this figure, it is shown that HCl and $ClONO_2$ rapidly decreased at around day 130 at 87.9ºS, when PSCs started to form in the Antarctic polar vortex (Figures 17(e) and 17(f)). The decrease of HCl stopped
when the counter-part of the heterogeneous reaction (1) was missing at around day 140. Consequently, $Cl_2$ was formed (Figures 17(c)). Similar chlorine activation was seen at 76.7ºS about 5-10 days later than at 87.9ºS. Gradual conversion from $Cl_2$ into $Cl_2O_2$ (ClO-dimer) was seen at all latitudes at around day 150-160 (Figures 17(b) and 17(c)), through reaction (5), (8), and (9). At the higher latitude (87.9ºS), conversion from $Cl_2$ to $Cl_2O_2$ was slow, due to lack of sunlight which is needed for reaction (5). Increase of ClO occurred much later after winter (day 190 or later), because sun light is needed to form ClO by
reactions (5) and (8) in the polar vortex (Figure 17(d)). Nevertheless, there were some enhancement of ClO in early winter, day 175, simulated at the edge of the polar vortex (Figure 13(d)) where there was some sunlight available due to the distortion of the shape of the polar vortex. Increase of ClO occurred from lower latitude (68.4ºS) at around day 195, towards higher latitude (87.9ºS) at around day 255 (Figure 17(d)). Diurnal variation of ClO was also seen at latitudes between 68.4ºS and 76.7ºS. When stratospheric temperature increased above PSC saturation temperature at around day 270 (Figure 5(a)), chlorine
activation ended, and ClO was mainly converted into HCl at all latitudes inside the polar vortex (Figures 17(d) and 17(e)). This is because reaction (13) occurs more frequently than reaction (12) inside the polar vortex, as there was little $NO_2$ available due to the depleted $O_3$ amount there.

Continuous loss of HCl was seen at 87.9ºS between days 160 and 200 even after the disappearance of the counterpart of heterogeneous reaction (1) (Figure 17(e)). The cause of this continuous loss was unknown until recently, where a hypothesis
was proposed that includes the effect of the formation of $NO_x$ due to galactic cosmic rays during the winter polar vortex (Grooß et al., 2018). Solomon et al. (2015) proposed a new mechanism on this issue: Continuous transport of $ClONO_2$ from the subpolar regions near 55-65ºS to higher latitudes near 65-75ºS provides a flux of $NO_x$ from more sunlit latitudes into the polar vortex. Our result also shows the same phenomena indicated by some sporadic increase in $ClONO_2$ at around days 158, 179, and 189 at 76.7ºS as shown in Figure 17(f). Subsequently, HCl losses were observed at 76.7ºS and 87.9ºS during these episodes
in Figure 17(e). The continuous loss of HCl at the most polar latitude (87.9ºS) might be due to the gradual mixing of air within the polar vortex during the winter period, when polar vortex was still strong.



## 6. Summary

Lower stratospheric vertical profiles of $O_3$, $HNO_3$, and HCl were retrieved using SFIT2 from solar spectra taken with a ground-based FTIR installed at Syowa Station, Antarctica from March to December 2007 and September to November 2011. This was the first continuous measurements of chlorine species related to the ozone hole from the ground in Antarctica. Retrieved

profiles were validated with Aura/MLS and ozonesonde data. The absolute differences between FTIR and Aura/MLS or ozonesonde measurements were within measurement error bars at the altitude of interest.

To study the temporal variation of chlorine partitioning and ozone destruction from fall to spring in the Antarctic polar vortex, we analyzed temporal variations of measured minor species by FTIR over Syowa Station combined with satellite measurements of ClO, HCl, $ClONO_2$ and $HNO_3$. When the stratospheric temperature over Syowa Station fell below PSC

saturation temperature, PSCs started to form and heterogeneous reaction between HCl and $ClONO_2$ occurred and $ClONO_2$ was almost completely lost at both 18 km and 22 km in early winter. When the sun came back to the Antarctic in spring, enhancement of ClO and gradual $O_3$ destruction were observed. During the ClO enhanced period, negative correlation between ClO and $ClONO_2$ was observed in the time-series of the data at Syowa Station. This negative correlation is associated with the distance between Syowa Station and the inner edge of the polar vortex. Temporal variation of chlorine species over Syowa

Station was affected both by heterogeneous chemistry related to PSC occurrence deep inside the polar vortex, and transport of $NO_x$-rich airmass from lower latitudinal polar vortex boundary region, which can produce additional $ClONO_2$ by reaction (12).

To see the comprehensive behavior of chlorine and related species inside the polar vortex and the boundary region in more detail, results of MIROC3.2 CCM simulation were analyzed. In general, the model results are in good agreement with FTIR and satellite observations. Rapid conversion of chlorine reservoir species (HCl and $ClONO_2$) into $Cl_2$, gradual conversion of

$Cl_2$ into $Cl_2O_2$, increase of ClO when sunlight became available, and conversion of ClO into HCl were successfully reproduced by the CCM. HCl decrease in the winter polar vortex core continued to occur due to the transport of $ClONO_2$ from the subpolar region (55-65ºS) to higher latitudes (65-75ºS), providing a flux of $ClONO_2$ from more sunlit latitudes into the polar vortex.

The deactivation pathways from active ClO into reservoir species (HCl and/or $ClONO_2$) were found to be very dependent on the availability of ambient $O_3$ and $NO_x$. At an altitude where most ozone was depleted in the Antarctic, most ClO was

converted to HCl. However, when there were some $O_3$ and $NO_x$ available, super-recovery of $ClONO_2$ can occur, like the case in the Arctic.

## Appendix A. MIROC3.2 nudged chemistry–climate model

The chemistry-climate model (CCM) used in this study was MIROC3.2 CCM, which was developed on the basis of version

3.2 of the Model for Interdisciplinary Research on Climate (MIROC3.2) general circulation model (GCM). The MIROC3.2 CCM introduces the stratospheric chemistry module of the old version of the CCM that was used for simulations proposed by the chemistry–climate model validation (CCMVal) and the second round of CCMVal (CCMVal2) (WMO, 2007, 2011;



SPARC CCMVal, 2010; Akiyoshi et al., 2009, 2010). The MIROC3.2 CCM is a spectral model with a T42 horizontal resolution ($2.8° \times 2.8°$) and 34 vertical atmospheric layers above the surface. The top layer is located at approximately 80 km (0.01 hPa). Hybrid sigma–pressure coordinates are used for the vertical coordinate. The horizontal wind velocity and temperature in the CCM were nudged toward the ERA–Interim data (Dee et al., 2011) to simulate global distributions of ozone

and other chemical constituents on a daily basis. The transport is calculated by a semi–Lagrangian scheme. The chemical constituents included in this model are $O_x$, $HO_x$, $NO_x$, $ClO_x$, $BrO_x$, hydrocarbons for methane oxidation, heterogeneous reactions for sulfuric-acid aerosols, supercooled ternary solutions, nitric-acid trihydrate, and ice particles. The CCM contains 13 heterogeneous reactions on multiple aerosol types as well as gas-phase chemical reactions and photolysis reactions. The reaction-rate and absorption coefficients are based on JPL–2010 (Sander et al., 2010). See Akiyoshi et al. (2016) for more

details.

**Appendix B.  Direct comparisons of mixing ratios of ClO, HCl, ClONO₂, Cly, and O₃ measured by FTIR and MLS, and modeled by MIROC3.2 CCM**

Mixing ratios of several minor species measured by FTIR, Aura/MLS, and Envisat/MIPAS, are compared with values modeled

by MIROC3.2 CCM. Figure B1 shows daily time series of measured and modeled ClO, HCl, ClONO₂, Cly, and O₃ over Syowa Station at 18 km in 2007 (Figures B1(a)-(e)), and in 2011 (Figures B1(f)-(j)). Figure B2 shows similar values at 22 km. In these figures, $Cl_y$ by Aura/MLS represents the $Cl_y*$ value defined by equation (15) using the $N_2O$ value measured by Aura/MLS. $Cl_y$ from the MIROC3.2 CCM is the sum of total reactive chlorines, i.e., $Cly = Cl + 2*Cl_2 + ClO + 2*Cl_2O_2 + OClO + HCl + HOCl + ClONO_2 + ClNO_2 + BrCl$.

Note that measured ClO values by Aura/MLS show daytime values, while modeled ones by the MIROC3.2 CCM are daily average, which are usually smaller than daytime values, because most ClO is converted to $Cl_2O_2$ by reaction (9) during nighttime. Also, modeled HCl and $Cl_y$ showed systematically smaller values compared with FTIR or MLS measurements. The cause of this discrepancy may be due to either smaller downward advection and/or faster horizontal mixing of airmass across the subtropical barrier in MIROC3.2 CCM (Akiyoshi et al., 2016). Nevertheless, day-to-day relative variations of

measured HCl and ClONO₂ are fairly well reproduced by the MIROC3.2 CCM. Modeled O₃ were in very good agreement with FTIR and/or MLS measurements throughout the year in both altitudes for both years.

**Acknowledgments**

We acknowledge Dr. Takeshi Kinase for making the FTIR observation at Syowa Station in 2011. All the members of the 48[th]

Japanese Antarctic Research Expedition (JARE-48), JARE-52 are acknowledged for their support in making the FTIR observation at Syowa Station. Data provision of Aura/MLS and Envisat/MIPAS are much appreciated. We thank Japan Meteorological Agency for providing the ozonesonde and Dobson data at Syowa Station. Thanks are due to Dr. Yosuke



Yamashita for performing the CTM run and Dr. Eric Dupuy for preparing figures of the model output. The model computations were performed on NEC-SX9/A(ECO) and NEC SX-ACE computers at the CGER, NIES, supported by the Environment Research and Technology Development Funds of the Ministry of the Environment (2-1303) and Environment Restoration and Conservation Agency (2-1709). We thank Dr. Takafumi Sugita for useful discussion and comments.

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



**Tables**

**Table 1. FTIR observation dates at Syowa Station in 2007 and 2011**

| Month | Dates (2007) | Dates (2011) | Number of days inside the polar vortex (2007/2011) | Number of days in the boundary region of the polar vortex (2007/2011) | Number of days outside the polar vortex (2007/2011) | Number of measurement days (2007/2011) |
|---|---|---|---|---|---|---|
| March | 25 | | 0 / 0 | 0 / 0 | 1 / 0 | 1 / 0 |
| April | 1, 3, 4, 5, 8, 24, 26, 28 | | 0 / 0 | 0 / 0 | 8 / 0 | 8 / 0 |
| May | 8, 9, 10, 13, 14, 15, 20, 21, 22 | | 7 / 0 | 0 / 0 | 2 / 0 | 9 / 0 |
| June | | | 0 / 0 | 0 / 0 | 0 / 0 | 0 / 0 |
| July | 29, 30 | | 0 / 0 | 2 / 0 | 0 / 0 | 2 / 0 |
| August | 1, 8, 9, 10, 24, 25, 26, 28, 29 | | 8 / 0 | 1 / 0 | 0 / 0 | 9 / 0 |
| September | 1, 4, 5, 6, 7, 8, 16, 18, 23, 26, 27, 30 | 25, 29, 30 | 12 / 3 | 0 / 0 | 0 / 0 | 12 / 3 |
| October | 6, 10, 11, 14, 19, 20, 25, 26, 27 | 1, 3, 4, 8, 11, 22, 23, 24, 26 | 9 / 9 | 0 / 0 | 0 / 0 | 9 / 9 |
| November | 2, 3, 5, 6, 7, 8, 9, 10, 11, 16, 17, 18, 19, 21, 27, 29, 30 | 1, 2, 3, 9, 11, 16, 19 | 12 / 7 | 1 / 0 | 4 / 0 | 17 / 7 |
| December | 4, 7, 8, 9, 13, 15, 16, 17, 20, 22, 29 | | 8 / 0 | 0 / 0 | 3 / 0 | 11 / 0 |
| **Total** | | | 56 / 19 | 4 / 0 | 18 / 0 | 78 / 19 |



**Table 2.  Retrieval parameters of SFIT2**

| Species | O₃ | HNO₃ | HCl |
|---|---|---|---|
| **Spectroscopy** | HITRAN 2008 | HITRAN 2008 | HITRAN 2008 |
| **PT Profile** | Daily sonde (0-30 km) CIRA 86 (30-100 km) | Daily sonde (0-30 km) CIRA 86 (30-100 km) | Daily sonde (0-30 km) CIRA 86 (30-100 km) |
| **A priori profiles** | Monthly averaged by ozonesonde (0-30 km) & ILAS-II (30-100 km) | Monthly averaged by ILAS-II | Monthly averaged by HALOE |
| **Microwindows (cm⁻¹)** | 1002.578 – 1003.500 1003.900 – 1004.400 1004.578 – 1005.000 | 867.000 – 869.591 872.800 – 874.000 | 2727.730 – 2727.830 2775.700 – 2775.800 2925.800 – 2926.000 |
| **Retrieved interfering species** | O₃ (668), O₃ (686), CO₂, H₂O | H₂O, OCS, NH₃, CO₂, C₂H₆ | CO₂, H₂O, O₃, NO₂ |





**Table 3.  Summary of validation results of FTIR profiles compared with ozonesonde and Aura/MLS measurements, and possible Aura/MLS biases from literatures**

|  | D-value (%) 18-22 km | Min/Max (%) 18-22 km | Agreement 15-25 km (ppm/ppb) | Literature values |
|---|---|---|---|---|
| O₃ | +6.2 | -10.4/+19.2 | -0.02~+0.40 | |
| HNO₃ | +13.2 | +0.2/+21.9 | -0.56~+0.57 | Aura/MLS no bias with errors (0.6 ppb) (Livesey et al., 2011) |
| HCl | -9.7 | -14.6/-3.0 | -0.2~+0.09 | Aura/MLS > HALOE by 10-15%, precision 0.2-0.6 ppb (Livesey et al., 2013) |





**Table 4. Summary of ClO variations**

| Altitude | 18 km | | 22 km | |
|---|---|---|---|---|
| Year | 2007 | 2011 | 2007 | 2011 |
| Period when ClO = 0 (day) | -190 | -200 | -140 | -150 |
| Period of enhanced ClO (day) | 230-260 | 230-260 | 220-240 | 230-250 |
| Variation when ClO enhanced (ppbv) | 0–1.3 | 0–1.5 | 0–2.2 | 0–2.2 |
| Day when ClO gets 0 (day) | 280 | 290 | 280 | 280 |



**Table 5.** **Summary of HCl variations**

| Altitude | 18 km | | 22 km | |
|---|---|---|---|---|
| Year | 2007 | 2011 | 2007 | 2011 |
| Value before winter (ppbv) | 1.5–1.8 | 1.2–1.6 | 2.1–2.4 | 1.8–2.2 |
| Starting day of decrease (day) | 140 | 140 | 130 | 140 |
| Ending day of decrease (day) | 180 | 180 | 180 | 170 |
| Variation when HCl ~ 0 (ppbv) | 0–0.3 | 0–0.3 | 0.1–1.0 | 0.1–0.9 |
| Day when HCl starts to increase | 250 | 250 | 240 | 240 |
| Day when HCl increase stops | 300 | 300 | 280 | 300 |
| Value after increase (ppbv) | 2.6–3.0 | 2.5–2.8 | 2.1–2.4 | 2.0–2.5 |
| Value outside polar vortex (ppbv) | 1.5–2.0 | 1.0–1.8 | 1.5–2.0 | 1.5–2.0 |





**Table 6. Summary of ClONO₂ variations**

| Altitude | 18 km | | 22 km | |
|---|---|---|---|---|
| **Year** | 2007 | 2011 | 2007 | 2011 |
| **Value before winter (ppbv)** | ~0.5 | ~0.4 | 0.6–0.9 | 0.6–0.7 |
| **Starting day of decrease (day)** | 160 | - (after 150) | 160 | between 140 – 150 |
| **Ending day of decrease (day)** | - (after 170) | - | - | 160 |
| **Variation when ClONO₂~0 (ppbv)** | 0–1.5 | 0–1.5 | 0–2.0 | 0–2.0 |
| **Day of ClONO₂ enhancement** | - | 270-300 | 270-280 | 270–280 |
| **Value of ClONO₂ enhancement (ppbv)** | - | 1.5 | 1.5 | 1.5 |
| **Value after enhancement (ppbv)** | 0–0.3 | 0–0.2 | 0.8–1.3 | 0.8–1.1 |
| **Value outside polar vortex (ppbv)** | 0.3–0.4 | 0.2–0.3 | 0.5–0.7 | 0.6–0.8 |



**Table 7. Summary of Cl$_y$\* variations**

| Altitude | 18 km | | 22 km | |
|---|---|---|---|---|
| Year | 2007 | 2011 | 2007 | 2011 |
| Value before winter (ppbv) | 2.3 | 2.2 | 2.8 | 2.9 |
| Value after winter (ppbv) | 3.1 | 2.9 | 3.2 | 3.2 |
| Value outside polar vortex (ppbv) | 1.5–2.0 | 1.5–2.0 | 2.4–2.6 | 2.2–2.8 |



**Table 8. Summary of O$_3$ variations**

| Altitude | 18 km | | 22 km | |
|---|---|---|---|---|
| Year | 2007 | 2011 | 2007 | 2011 |
| Value before winter (ppmv) | 2.5 | 2.5 | 4.0 | 4.0 |
| Starting day of decrease (day) | 190 | 200 | 170 | 170 |
| Ending day of decrease (day) | 280 | 270 | 260 | 270 |
| Minimum value (ppmv) | 0.3 | 0.5 | 2.0 | 1.0 |
| Value after recovery (ppmv) | 0.8 | 0.8 | 2.4–4.0 | 2.0–3.5 |



**Table 9. Summary of HNO₃ variations**

| Altitude | 18 km | | 22 km | |
|---|---|---|---|---|
| **Year** | 2007 | 2011 | 2007 | 2011 |
| **Value before winter (ppbv)** | 6-10 | 8-10 | 15-16 | 13–15 |
| **Starting day of decrease (day)** | 160 | 150 | 140 | 150 |
| **Ending day of decrease (day)** | 190 | 180 | 180 | 180 |
| **Minimum value (ppbv)** | 0 | 0 | 2 | 1 |
| **Value after recovery (ppbv)** | 3–4 | 3–4 | 4–6 | 4–5 |
| **Variation during decrease (ppbv)** | 0–11 (day 200-250) | 0–8 (day 210-240) | 2–14 (day 180-240) | 1–11 (day 200-240) |
| **Variation during decrease (ppbv)** | 1–5 (day 250-300) | 2–4 (day 240-300) | 2–6 (day 240-320) | 2–8 (day 240-300) |



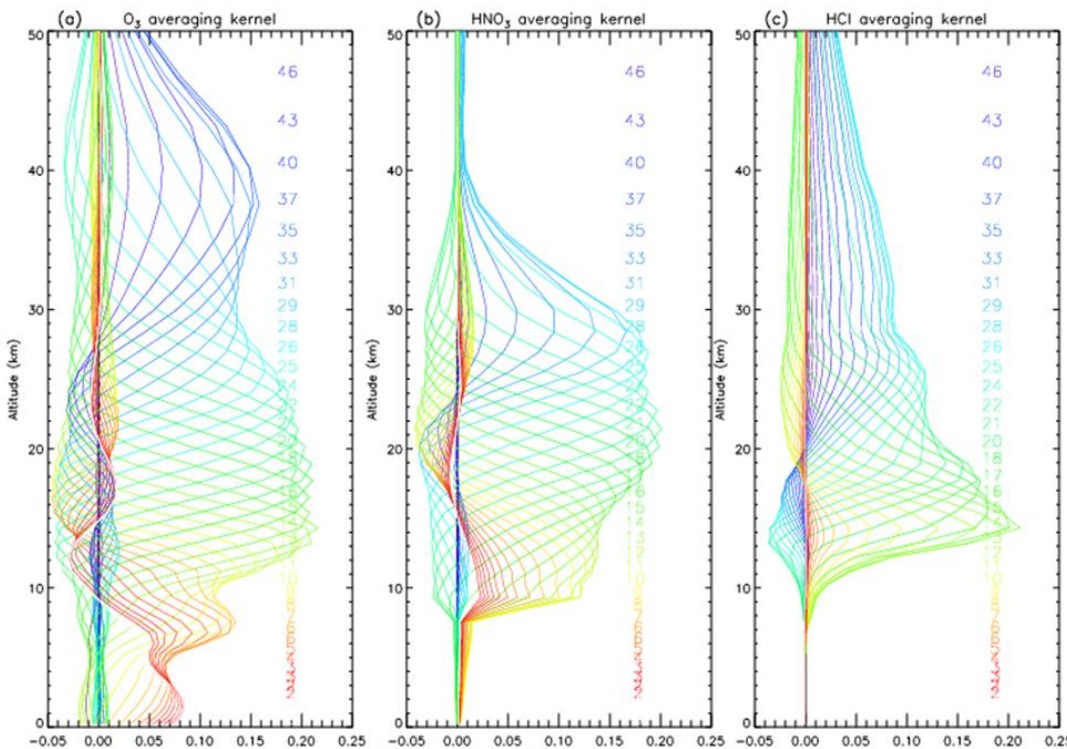

Figure 1. Averaging kernel functions of the SFIT2 retrievals for O₃ (a), HNO₃ (b), and HCl (c).





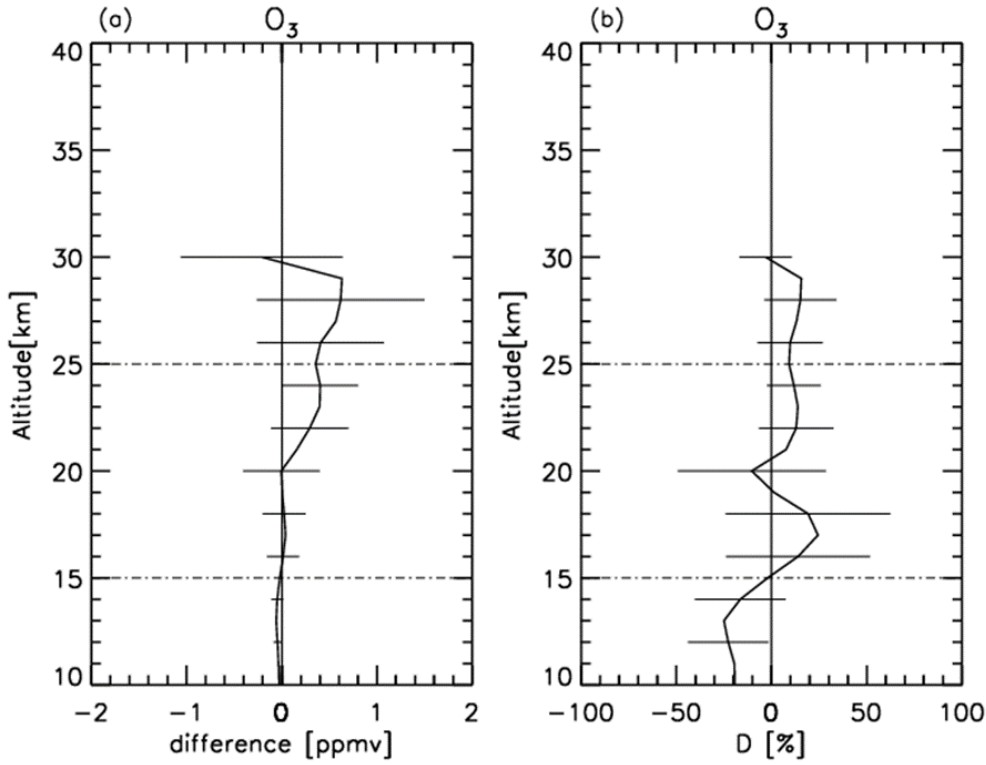

Figure 2. Absolute (a) and percentage (b) differences of O₃ profiles retrieved from FTIR measurements and those from ozonesonde measurements. Horizontal bars indicate the standard deviation of differences at each altitude.





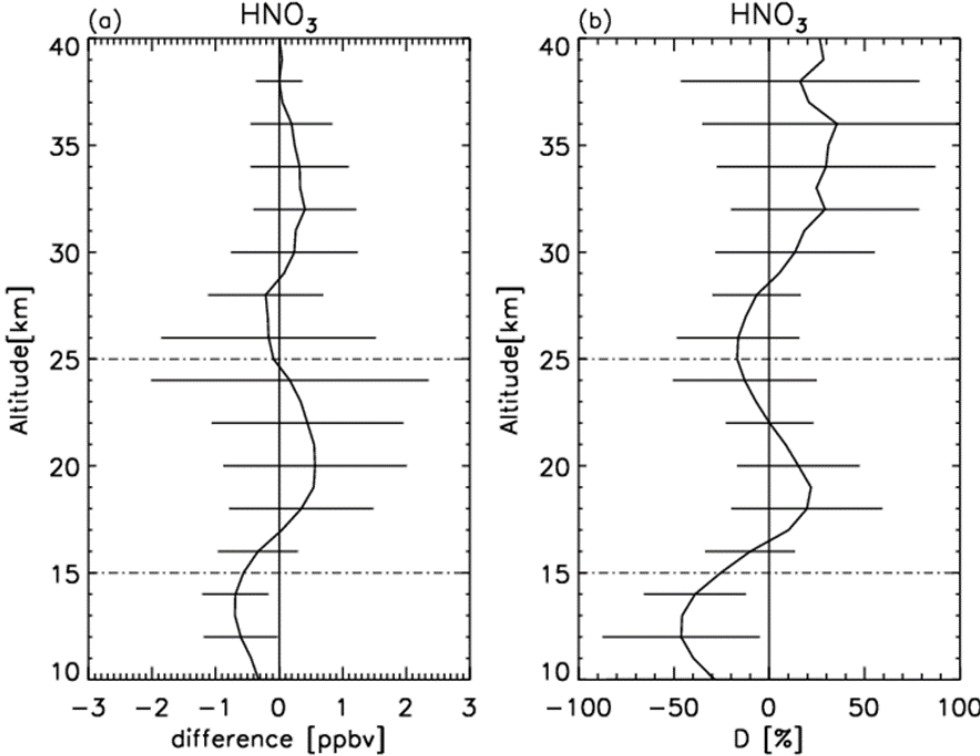

Figure 3. Same as Figure 2 but for HNO₃ profiles retrieved from FTIR measurements and those from Aura/MLS measurements.



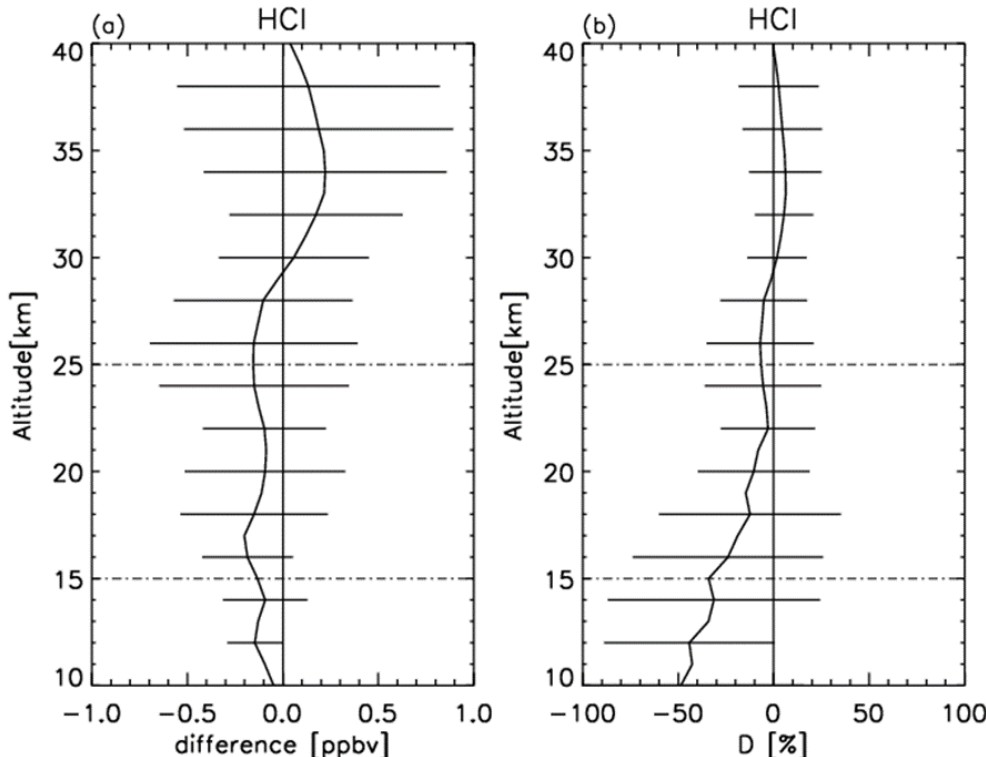

Figure 4. Same as Figure 3 but for HCl profiles.



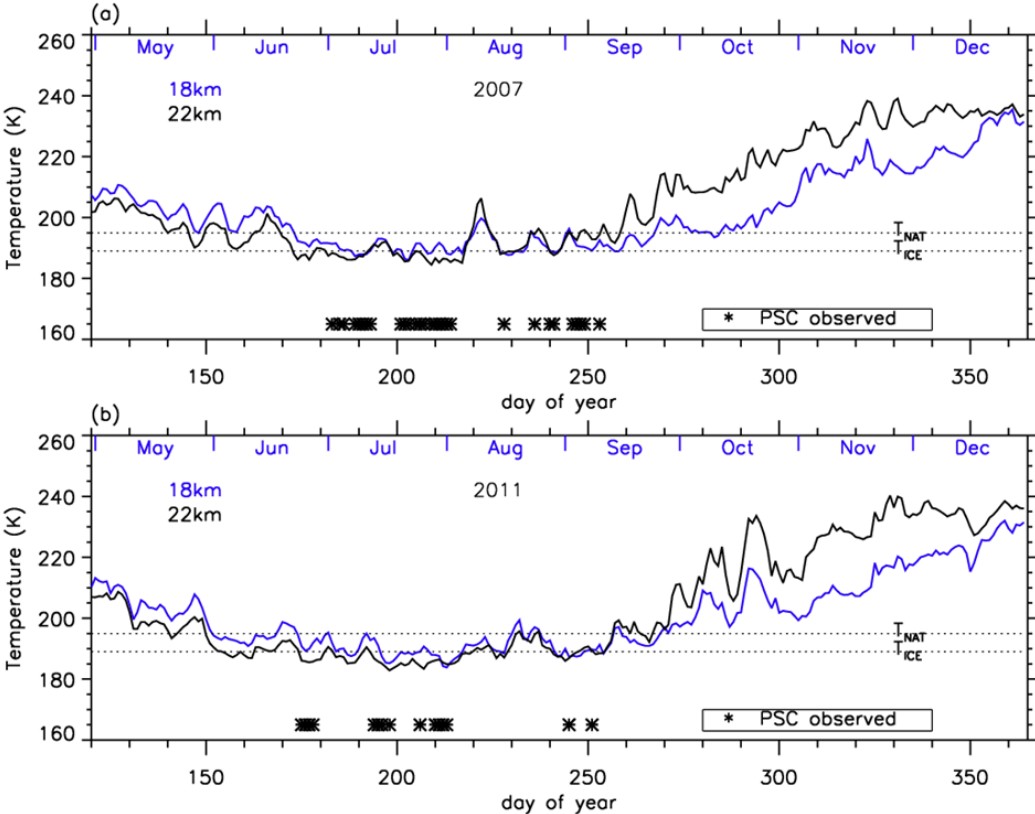

Figure 5. Temporal variation of temperature at 18 and 22 km in (a) 2007 and (b) 2011 over Syowa Station using ERA-Interim data. Approximate saturation temperatures for nitric acid trihydrate PSC ($T_{NAT}$) and ice PSC ($T_{ICE}$) are also plotted on the figures by dotted lines. Dates when PSCs were observed over Syowa Station are indicated by asterisks on the bottom of the figures.





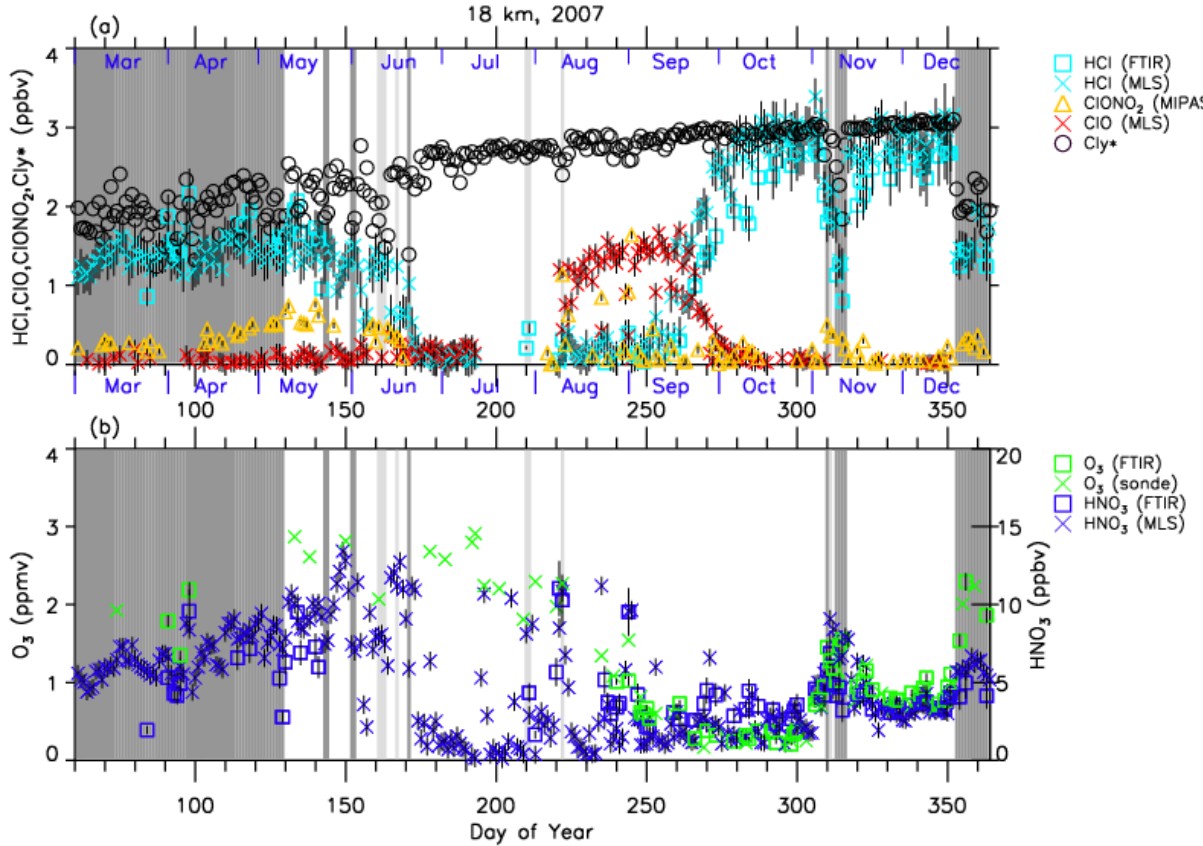

Figure 6. Temporal variations of (a) HCl, ClONO$_2$, ClO, Cl$_y$*, (b) O$_3$, and HNO$_3$ mixing ratios at 18km in 2007 over Syowa Station. O$_3$(FTIR), HCl(FTIR), and HNO$_3$(FTIR) were measured by FTIR at Syowa Station, while HCl(MLS), ClO, and HNO$_3$(MLS) were measured by Aura/MLS. O$_3$(sonde) was measured by ozonesonde. ClONO$_2$ was measured by Envisat/MIPAS. Cl$_y$* is calculated from N$_2$O value. See text in detail. The unit in O$_3$ is ppmv and the others are ppbv. Light and dark shaded areas are the days when Syowa Station is at the boundary region and outside the polar vortex, respectively.





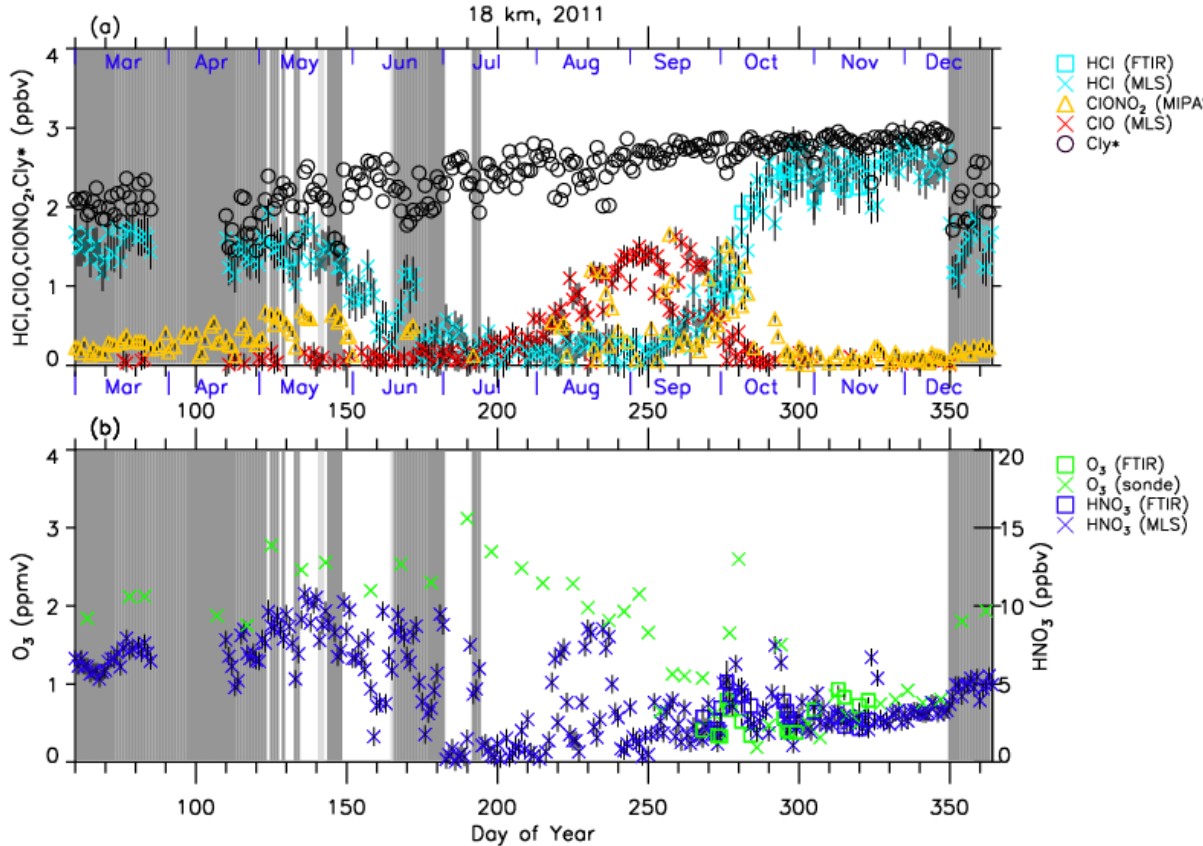

Figure 7. Same as Figure 6 but in 2011.

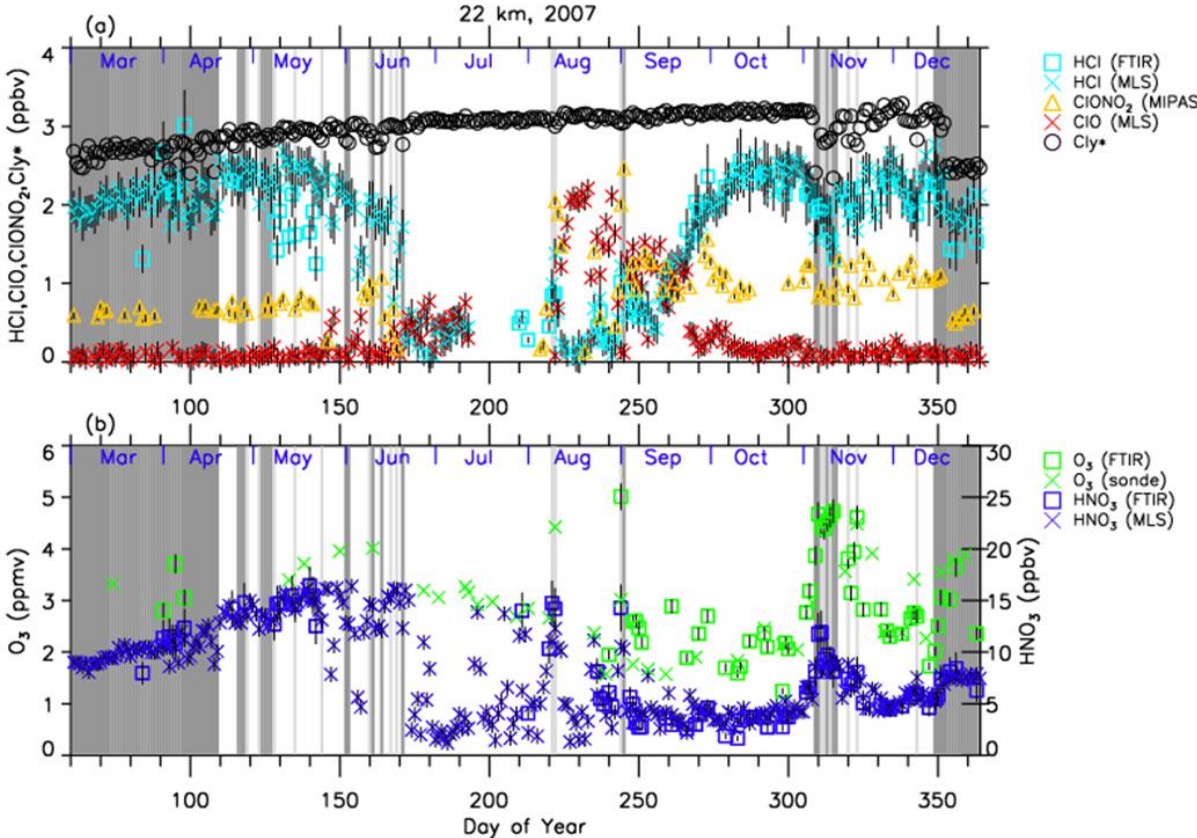

Figure 8. Same as Figure 6 but at 22 km.





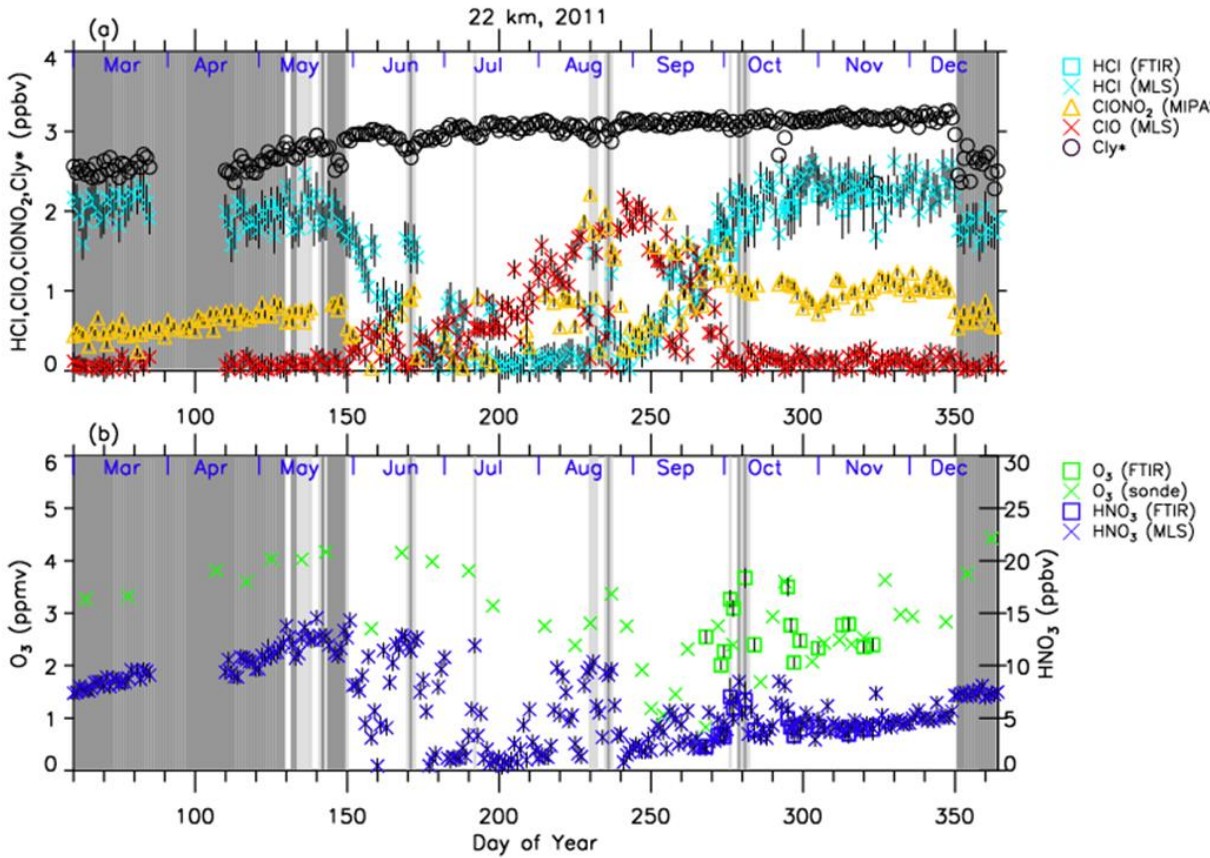

Figure 9. Same as Figure 7 but at 22 km.




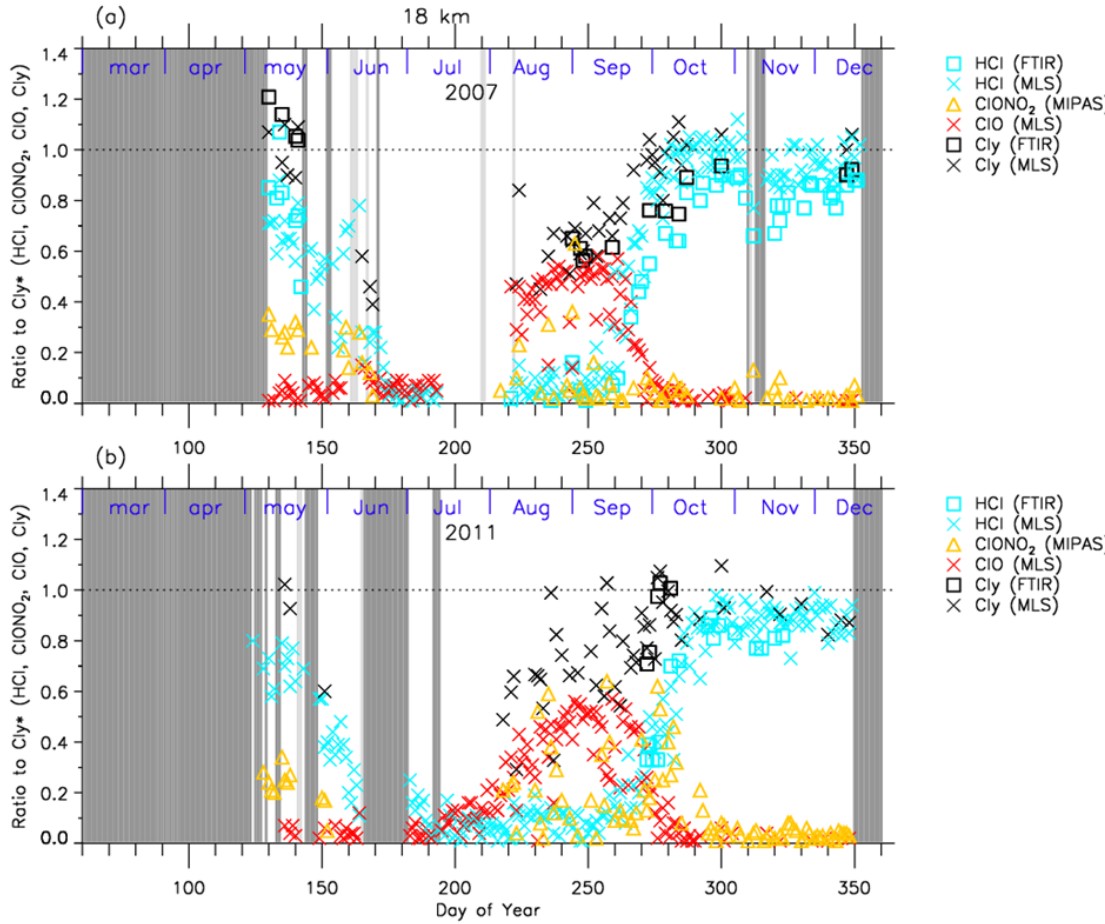

Figure 10. Temporal variations of the ratios of HCl, ClONO₂, ClO, and Cl$_y$(=HCl+ClONO₂+ClO) to total chlorine (Cl$_y$*) over Syowa Station at 18 km in (a) 2007 and in (b) 2011. Shaded areas are the same as Figure 6.



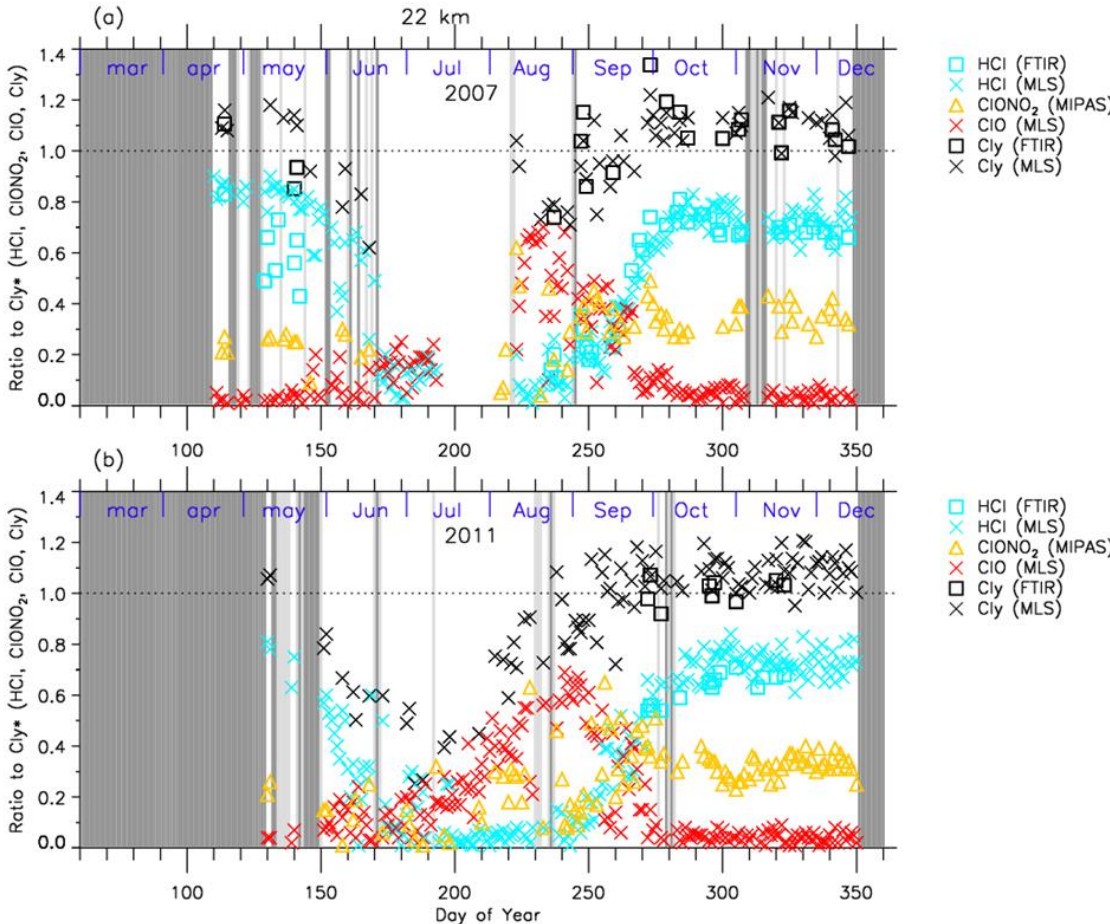

Figure 11. Same as Figure 10 but at 22 km.





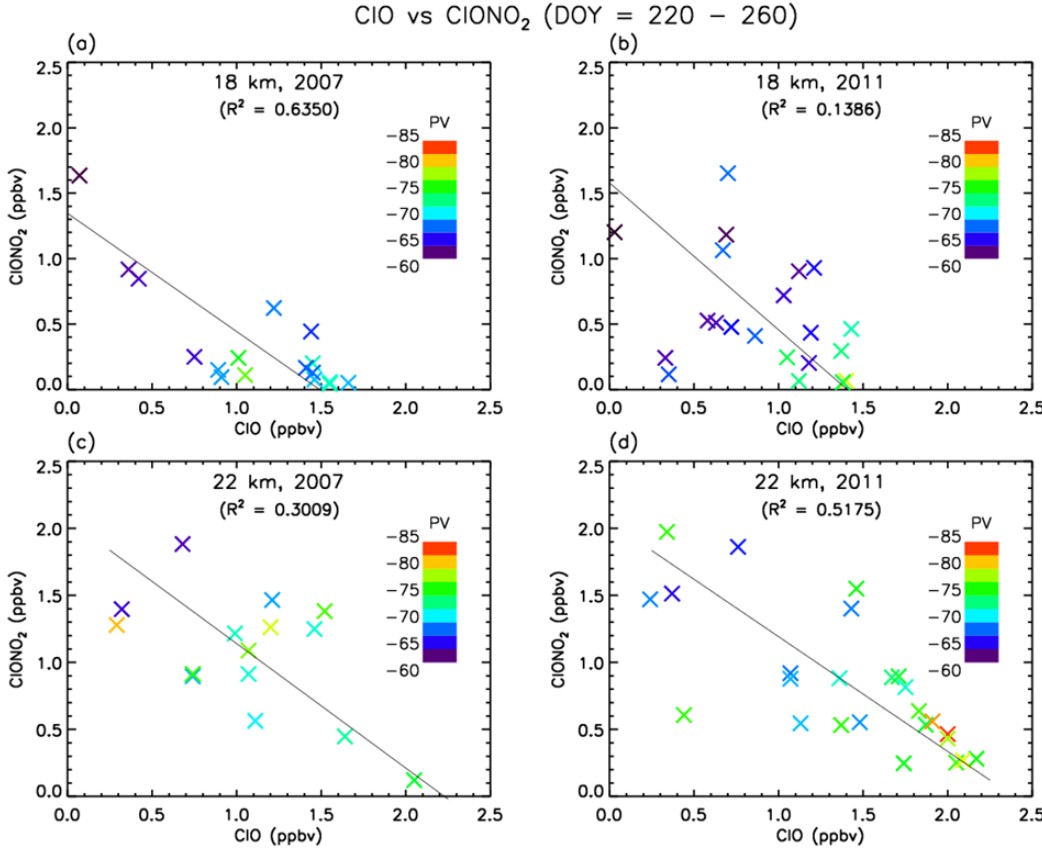

Figure 12. Scatter plot between ClO (Aura/MLS) and ClONO$_2$ (Envisat/MIPAS) mixing ratios during day number 220 – 260 at 18 km and 22 km in 2007 and 2011. Solid lines are regression lines obtained by RMA (Reduced Major Axis) regression. Color represents the potential vorticity over Syowa Station on that day.



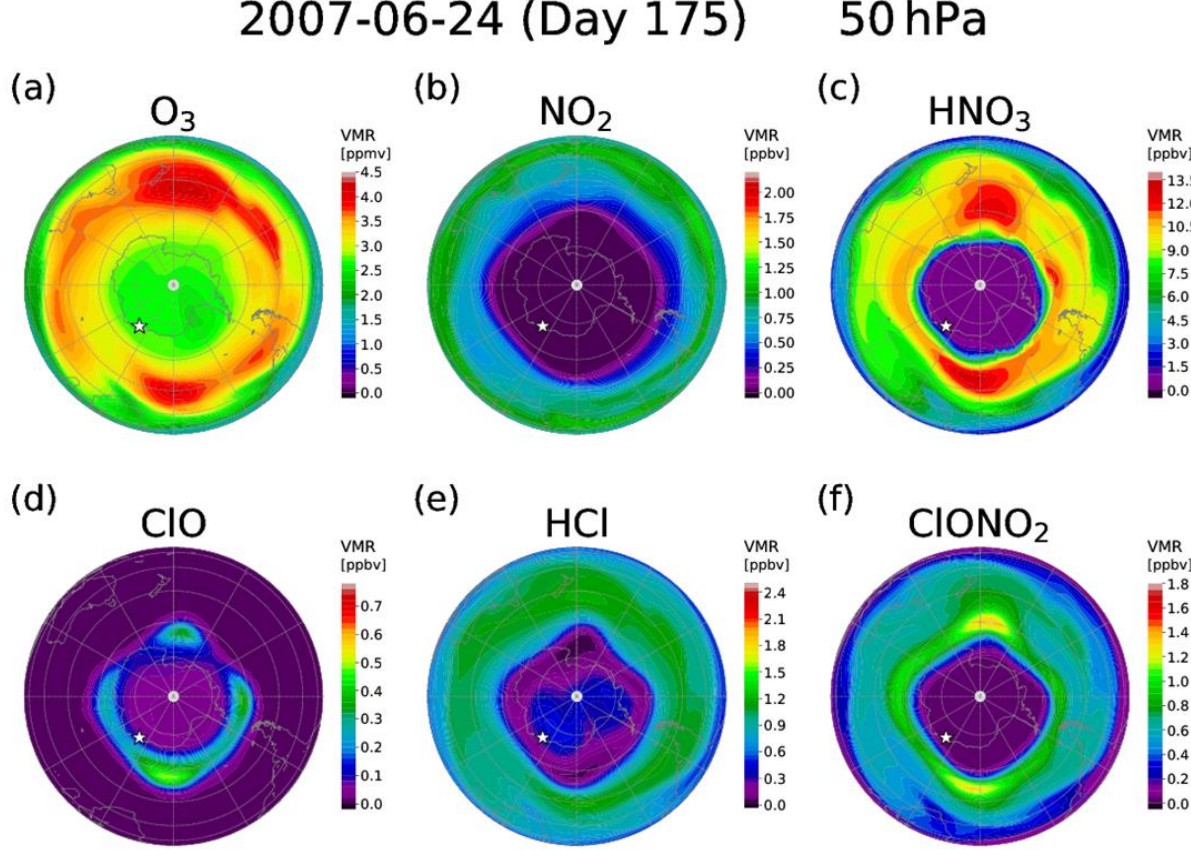

Figure 13. Polar southern hemispheric plot for simulated mixing ratios of O₃ (a), NO₂ (b), HNO₃ (c), ClO (d), HCl (e), and ClONO₂ (f) by a MIROC3.2 chemistry-climate model (CCM) at 50 hPa for June 24 (day 175), 2007. The location of Syowa Station is shown by white star in each panel.



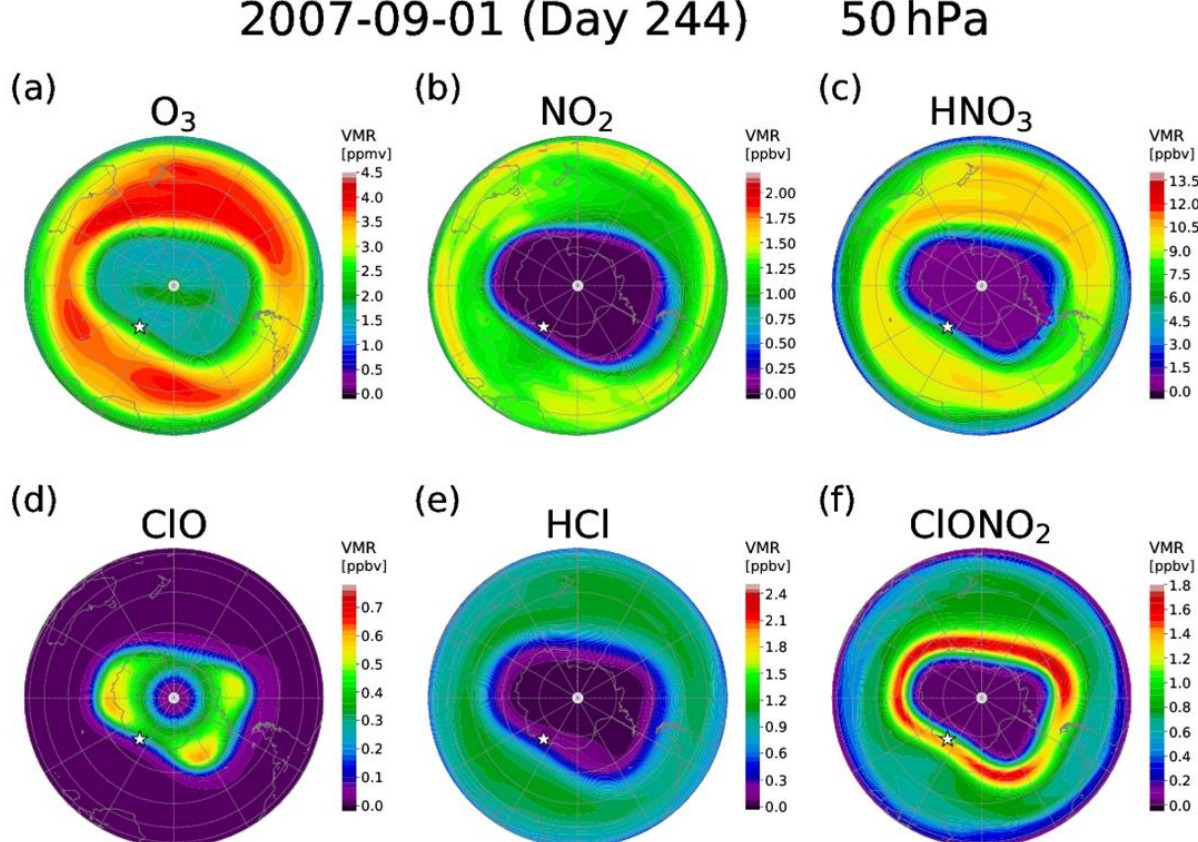

Figure 14. Same as Figure 13 but for September 1 (day 244), 2007.



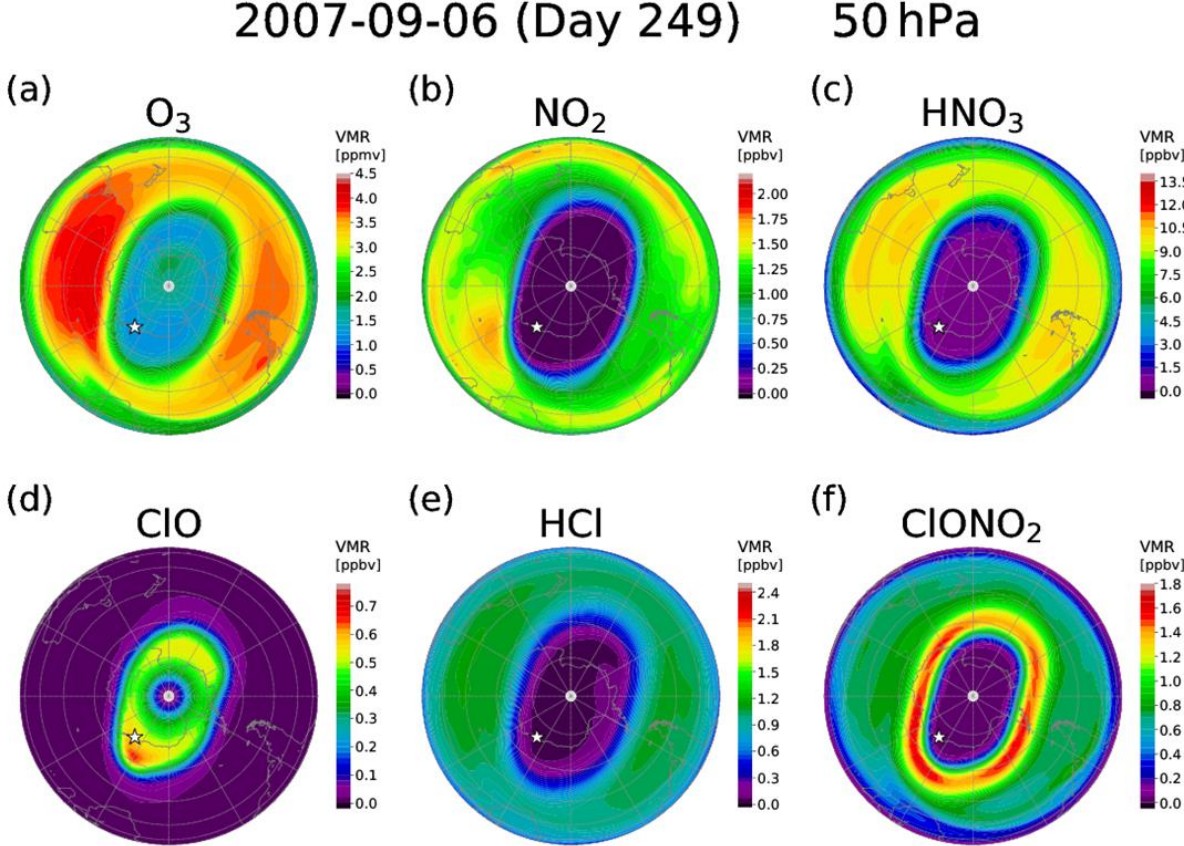

Figure 15.  Same as Figure 13 but for September 6 (day 249), 2007.



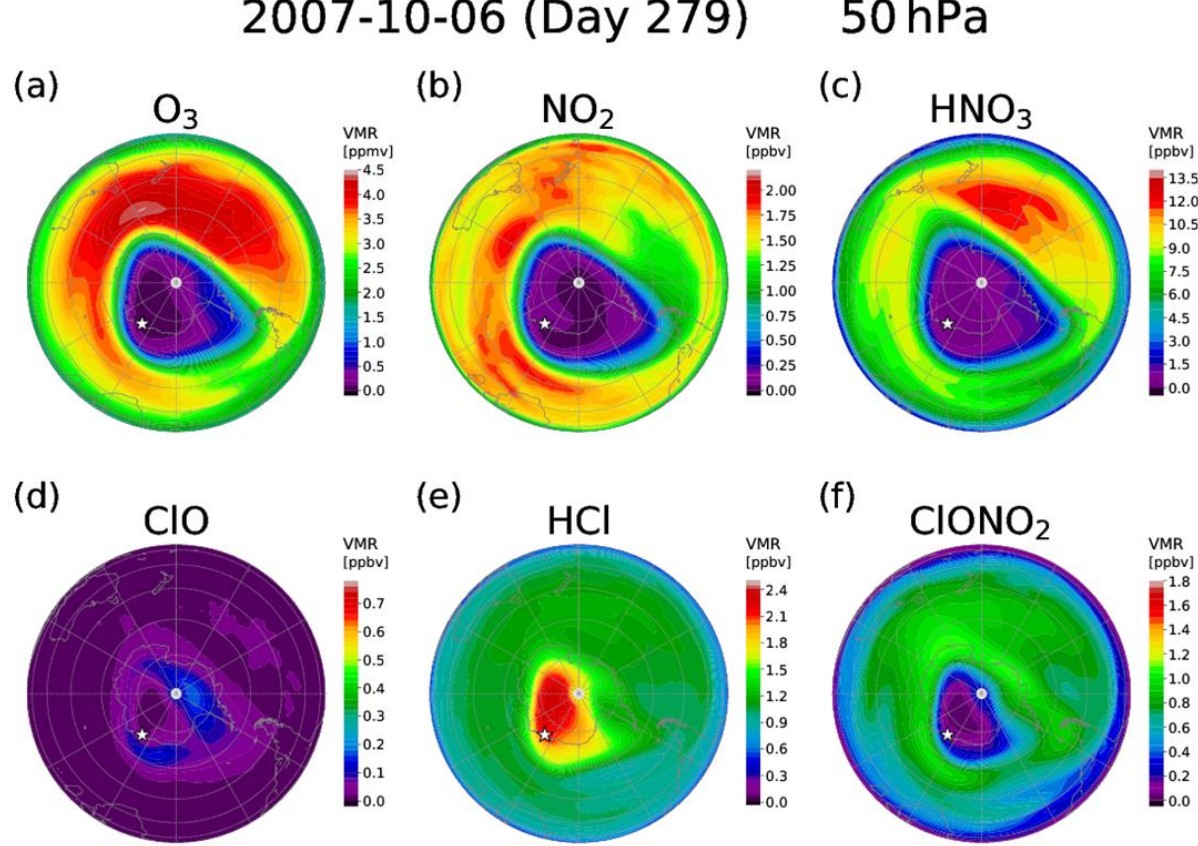

Figure 16.  Same as Figure 13 but for October 6 (day 279), 2007.



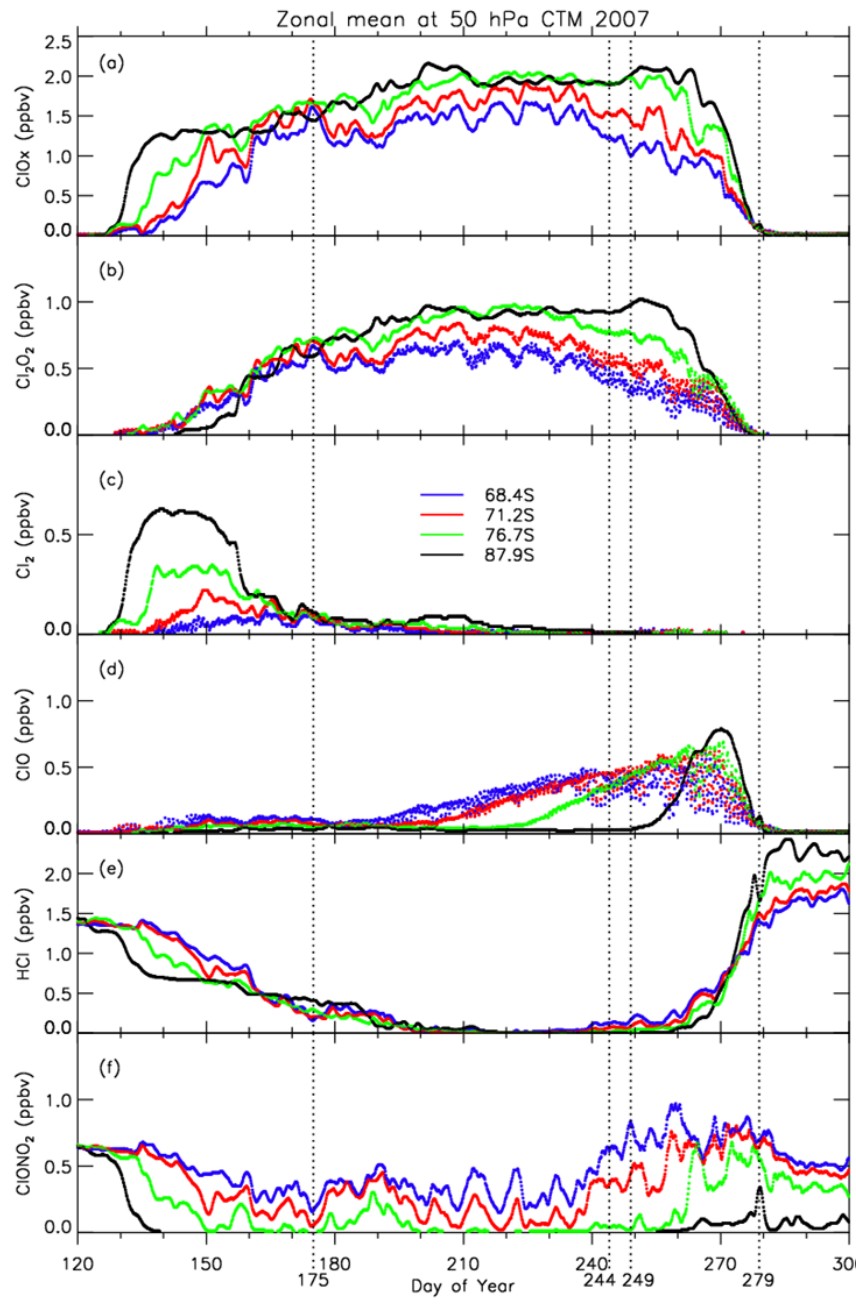

Figure 17. Three-hourly zonal-mean temporal variations of MIROC3.2 CCM outputs for (a) ClO+2*Cl$_2$O$_2$+2*Cl$_2$, (b) Cl$_2$O$_2$, (c) Cl$_2$, (d) ClO, (e) HCl, and (d) ClONO$_2$ during day number 120 – 300 at 50 hPa in 2007.





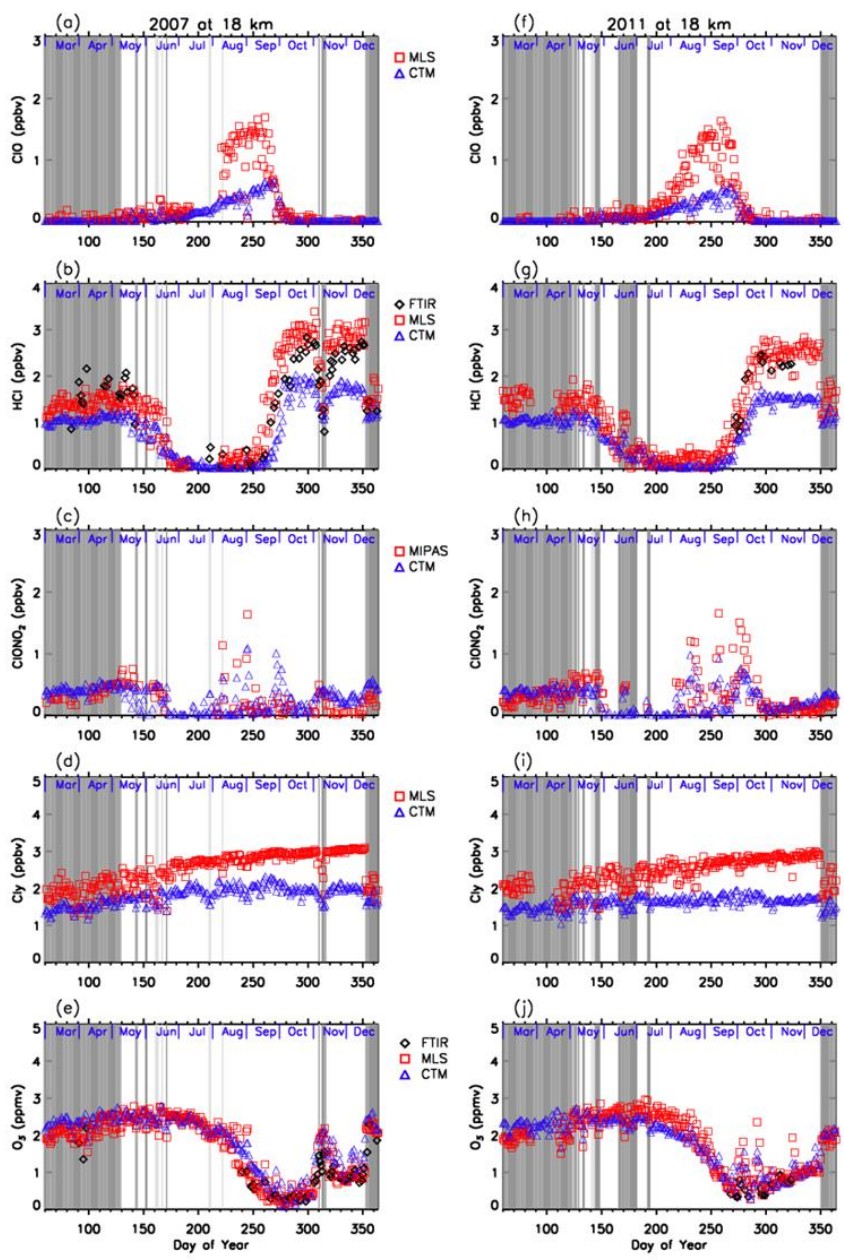

Figure B1. Daily time series of measured and modeled minor species over Syowa Station at 18 km. Black diamonds are data by FTIR, red squares are by Aura/MLS and Envisat/MIPAS, blue triangles are data by MIROC3.2 CCM. Figure B1(a) is for ClO, B1(b) is for HCl, B1(c) is for $ClONO_2$, B1(d) is for Cly, and B1(e) is for $O_3$ in 2007. Figure B1(f) is for ClO, B1(g) is for HCl, B1(h) is for $ClONO_2$, B1(i) is for Cly, and B1(j) is for $O_3$ in 2011.





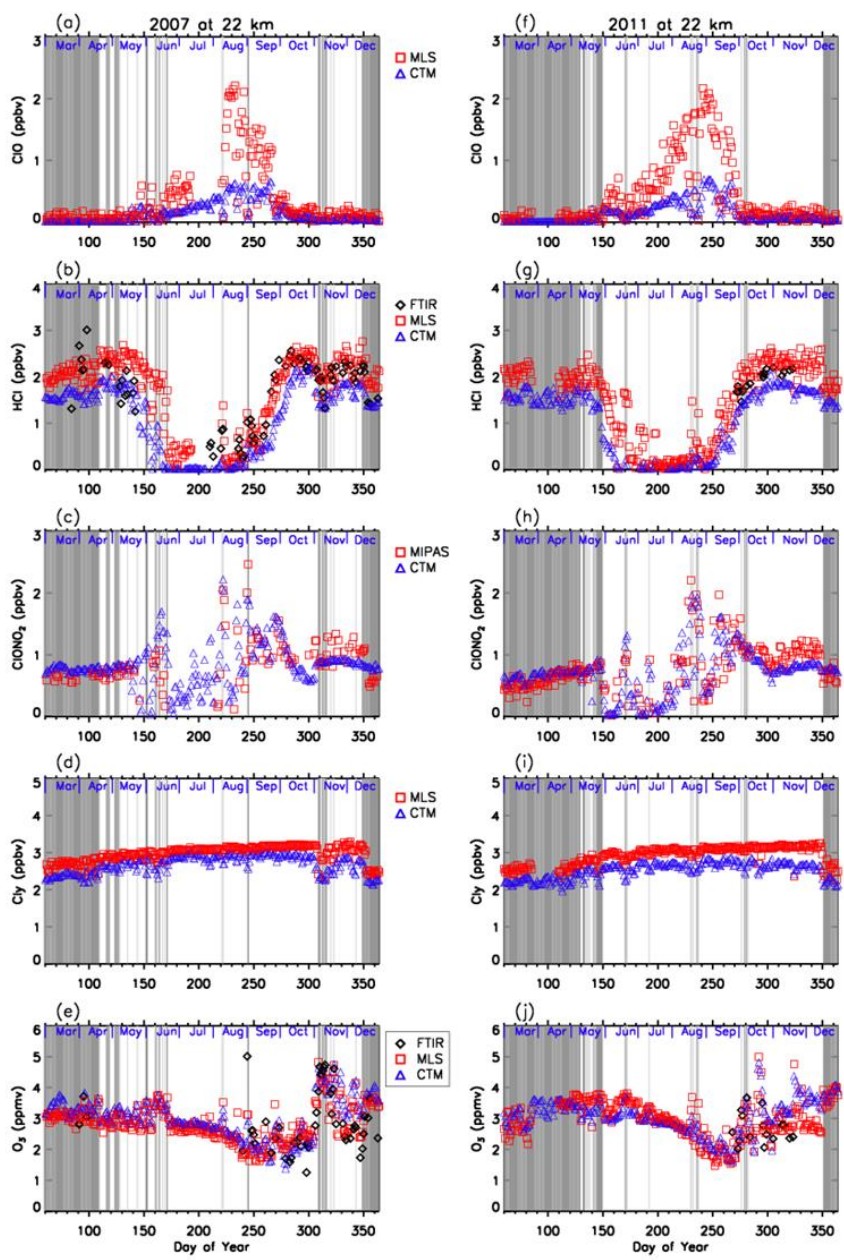

Figure B2. Same as Figure B1 but for 22 km.

