# Peer review of "Temporal evolution of chlorine and minor species related to ozone depletion observed with ground-based FTIR at Syowa Station, Antarctica and satellites during austral fall to spring in 2007 and 2011"

_Atmospheric Chemistry and Physics, 2018_

## Referee Comment (RC1) · Anonymous Referee #1 · 3 Jul 2018

Nakajima et al. investigate the evolution of several trace gases that are important for polar stratospheric chemistry based on measurements and model simulations. Therefore, they use FTIR measurements from Syowa station as well as satellite measurements from different satellite instruments. Additionally, model simulations with a chemistry climate model are used. This is generally an interesting study, but the study lacks of a clear presentation which makes it difficult to judge if the study itself and the results are sound. I would suggest majot revisions before the paper can be

accepted for publication in ACP.

**General comments:**
1. The title is not really representing what actually has been done in this study. In the title only the FTIR measurements from Syowa are mentioned, but also the satellite observations and model simulations comprise a major part of this study. This should be reflected in the title. I would suggest to simply change the title to: "Temporal evolution of chlorine and minor species related to ozone depletion at Syowa station, Antarctic during austral fall to spring in 2007 and 2011" to be more general or to "Temporal evolution of chlorine and minor species related to ozone depletion from observations and model simulations at Syowa station, Antarctic during austral fall to spring in 2007 and 2011" to be more specific.

2. The content of the Appendices are not that unimportant and therefore I would suggest that these parts are moved to the main text while rather other figures, as e.g. parts of Fig 13-16 could be put into the Appendix (see my comment on these figures below for more details).

3. The discussion section is rather a continuation of the result section. These two subsections should be divided into subsections with subsections header that describe what actually is analysed (time series, correlation etc.)

4. There are too many tables and figures in the current version of the manuscript. These tables/figures could be combined. This would not only reduce the number of figures, but also make the manuscript more concise (see also my specific comments below).

**Specific comments:**
P2, L6: …...similar to the case in the Arctic…...This needs more explanation and

maybe also some references.

P2, L13: "….great year-to-year variability." I think this only holds for the Arctic and not for the Antarctic. The distinction between Arctic and Antarctic should be made here and the differences between both regions should be discussed.

P2, L18: References should be added.

P3, L3: Please add which temperatures are needed. Do you mean with the term "inactivation" the "deactivation"?

P3, L15-16: This sentence should be rephrased and maybe even better split into two sentences. It is not clear what you mean here. Also in Antarctica PSCs should occur within the polar vortex. Or is there a shift between the cold area and the vortex? If yes this should be said more clearly. However, you discuss here the temperature, so I guess you just mean that there are less PSCs in the Arctic due to the generally higher temperatures there.

P3, L30ff: Section 3 describes the validation of the FTIR measurements, but in the introduction nothing is mentioned about this validation.

P4, Section2: I would suggest to make two subsections, one for FTIR and one for the satellites and to also extend the description of the satellite data sets since these data sets are not that unimportant for this study.

P4, L5: I would suggest to discuss the discovery of the ozone hole based on the typical papers as e.g. Farman et al., 1985, Nature, first and then to discuss the measurements of the ozone hole over Syowa.

[Figure]

P4, L23ff: The abbreviations FASATM, CIRA-86, ILAS-II have not been introduced yet. The abbreviations of Aura/MLS and Envisat/MIPAS have been introduced in the abstract, but should be introduced once again here in the main text of the manuscript.

P4, L27ff: O3, HNO3, HCl were used as a priori. What about ClO and ClONO2. Were these also used as a priori?

P5, L5: The abbreviation CALIPSO has not been introduced yet.

P4-5. General: Different satellite instruments were used for different trace gases. It should be motivated why different instruments have been used and it also should be discussed if instrumental differences influence the results. What is the quality of the used data products?

P7, L12ff: A motivation is missing why two years and two altitudes have been considered. So far I could not see that the results are that much different so that it is justified two show two different altitudes and two different years. The differences between these two years and two altitudes needs to be more clearly described.

P8, L1ff: As mentioned above the discussion is rather a continuation of the result section. This section should be divided into subsections that describe what actually is analysed (time series, correlations etc.)

P9, L7ff: The comparison to model simulations is not really a comparison. It's rather a different way of analyses of the winter using a different data set than the ones before.

P10, L21-22: "….as there was little NO2 available due to the depleted O3 amount

here." This needs definitely to be explained. What is the relation between these two species?

P11, 3-4: "This study was ($\rightarrow$ were) the first continuous measurement of chlorine species related to the ozone hole from the ground in Antarctica" I this is really the case, then this should be also mentioned in the abstract.

P11, L25-26: "...like the case in the Arctic $\rightarrow$ like it is the case in the Arctic". That needs more explanation. It has not been stated what the usual case in the Arctic is and what generally the differences between Arctic and Antarctic are.

P11, L26ff: Conclusions should be given.

P11, L26ff: References should be given. The processes and interplay between the species are already known and documented. At statement what actually are the new findings of this specific study is missing.

Appendix A and B: Move this to these parts to the main text.

Tables: Too many tables! I would suggest to combine Table 4 to 9, thus creating a long table. Add a column for the species and then put the current tables under each other.

Figures 2-4: There are also too many figures. Figures 2-4 could be combined by putting these under each other.

Figure 8: Is it really necessary to show here two years? Couldn't one of the years put into the appendix or a supplement to the paper since the results do not seem to be that much different.

Figure 10: I would suggest to use lines instead of symbols. The interplay between the trace gases would then become much better visible (the same holds for Figure 6-9).

Figures 13-16: Why is a complete different representation of the model results used than for the measurement data? How are these comparable? I would suggest to combine Figures 13-16 and not to show all trace gases, but a selection. Put the gases in the rows and the dates in the columns. The remaining gases could be put into the appendix or a supplement to the paper.

Figures B1 and B2: These figures should be put into the manuscript instead of the appendix.

**Technical corrections:**
P1, L14: please add "the", so that it reads "understanding of the mechanisms. . ...".
P1, L29: change sentence to: "by reaction of ClO with NO2".
P2, L19ff: The numbering of the reactions. Doing this differently than just using number as e.g. using R1 to R13 would be more concise. However, how to do the numbering of equations should be checked with the ACP guidelines.
P3, L3: when stratospheric temperature gets warmer → when stratospheric temperatures get warmer
P3, L4: mainly occur → mainly occurs
P3, L9: between in → either "between" or "in", but not both.
P3, L23: Use a different beginning for the sentence as it is done now. Maybe it is even easier to combine this sentence with the one before so that it reads:". . . . . . . ...discovery of the ozone hole by direct observations from high-altitude aircraft."
P3, L23ff: If a not complete references list is given because there are too many references, these should be referenced with "e.g." in front and not with "etc." at the end.

P3, L28: trace components → trace gas components or better to write trace gases

P3, L30: trace species → trace gas species or trace gases

P3, L32: "during the course of the winter" or "during the ozone hole formation and dissipation period", at least "in the" should be replaced by "during".

P3, L33: rephrase sentence to ".....Finally, distributions of minor species simulated with the MIROC3.2 chemistry-climate model . . . . . ..".

P4, L30: Figures 1(a), (b), and (c) → Figures 1 (a)-(c) .

P5, L11: Same here for Figure 2.

P5, L16ff: Instead of just "D value" it should read "relative difference D" or just "relative difference".

P5, L26: Figures 4(a) amd 4(b) show → Figure 4 shows

P5, L29: the maximum → a maximum

P6, L5: Rephrase sentence to "Figure 5 shows the temporal variation of temperature and chlorine species. . ..."

P6, L7: on the figures → in the figures

P6, L12: temperature → temperatures

P6, L12: are → were

P6, L16: Figures 6,7,8 and 9 → Figures 6-9

P6, L18: MLS in parentheses obsolete since it is written ". . ..observed by Aura/MLS..."

P6; L21: correlation → equation. More correctly it should be written "polynomial equation derived from the correlation of Cly and N2O".

P6, L22: Skip "of" and replace the comma by a colon

P7, L12: is → was

P7, L25 and 28: ppb → ppbv

P8, L5: Rephrase sentence to "Figures 10 and 11 show the temporal variation of the ratios of the chlorine species to Cly*" or "Figures 10 and 11 show the temporal variation of the ratios between the chlorine species and Cly*".

P8, L25: include "the" so that it reads "...with HCl increase, then the ClONO2 amount gradually. . ...."

P9, L7: Figures 13, 14, 15, 15 → Figures 13-16
P9, L7: (a), (b), (c), (d), (e), (f) are obsolete.
P9, L16: reaction (12) → Reaction 12
P9, L18: (1), (2), (3), (4) → (1)-(4)
P9, L34: simplify this sentence with listing the trace gases and then adding Figure 16
in parentheses.
P10, L8: comma before "are" obsolete.
P10, L12: comma before through obsolete
P10, L13: At higher latitudes → "At this latitude" or "At latitudes > 87.9"
P10, L14: after → in
P10, L15: enhancement → enhancements
P11, L6: altitude → altitudes
Figure 6 caption: remove (FTIR) and (MLS) and (sonde) since it is enough if this suffix
is given in the figure legend. Change last three sentences to "See text for details. The
unit of O3 is ppmv and of the other gases ppbv. The light and dark shaded areas are
the days when Syowa station was at the boundary region and outside the polar vortex,
respectively.
Figure 8 and 9: The months should be added here as it done for figures 6 and 7.
Figure 12: "day number 220 – 260" → add time period as dates

---

## Referee Comment (RC2) · Anonymous Referee #2 · 5 Jul 2018

The manuscript by Nakajima et al. focuses on ground based FTIR observations of trace species in 18 and 22 km altitude for the years 2007 and 2011 made from the Japanese Syowa station on Antarctica. These data are compared with co-located MLS data and monthly average ozone sonde data. The manuscript further shows simulations with the chemistry climate model MIROC3.2 nudged to ERA-Interim data for the observation years 2007 and 2011.

[Figure]

Although there are likely no severe issues with the data or the simulations, the scientific goal of the manuscript remains completely unclear. I find that it is not sufficient, just to present the observations and simulations without addressing open scientific questions. The authors present no new novel concepts, ideas, or tools. The only thing that is said that they present the first ground based continuous measurements of chlorine species in Antarctica.

The model data are displayed in addition to the observations but not on the same altitude levels and they are not used to interpret the observations. There is no obvious connection between the simulations and the observations besides two appendix figures that contain the time series of the chlorine compounds and ozone over Syowa station from model and observations.

With respect to the interpretation of the data, it seems that only few aspects of the observations and the simulations are mentioned. Most of them are in line of what is expected and shown in standard chemistry model runs over the last two decades. The simulations seem to be in line with the observations. But it remains completely unclear, what the message of the paper is.

Furthermore, there are many small inaccuracies in the text, some of them are summarised below. Also, many informations are not given precisely, such that the reviewer needs to guess, what the authors meant. Because of the missing scientific concept, I would not recommend the paper for publication in Atmos. Chem. Phys.

**General**

The introductions lists some textbook knowledge but it is not clear, why it is at all important in the context of the manuscript.

The use of ClO data as done in this paper is problematic, since one needs to take into

account the diurnal cycle typically involving $Cl_2O_2$ as a nighttime reservoir. Observations (fig 6-12) are likely for different local times and are therefore not comparable. As in the 3-hourly model data of fig 17, no diurnal cycle is visible, it is likely a zonal average over daytime and nighttime data. That is not useful for comparison. Potentially this is the cause of the ClO difference in figs B1 and B2.

Figures 13-16 show the model output on the 50 hPa level for 4 different times. It is not so clear, what can be learned from these figures. Also it would be better to not use a different vertical coordinate (pressure) for the model results as opposed to the observations (altitude).

fig 10/11 shows $Cl_y$ (MLS) relative to $Cl_y$* (MLS $N_2O$), even though that should be described more clearly. In the context $Cl_y$ (without star) is defined as ClO + $ClONO_2$ + HCl. However, there are no $ClONO_2$ observations of MLS. This needs to be clarifies.

**Details**

page 1/line 16 "is not well understood": I don't think that this statement is justified.

1/22 PSC saturation temperature: you likely mean "PSC existence temperature".

2/3: "from active chlorine" or "from ClO" ?

2/17: the expression "inert chlorines" for HCl and $ClONO_2$ is not typical, please use the wording "chlorine reservoirs" (as in 3/31).

3/4-14: The chlorine deactivation into $ClONO_2$ or HCl is mentioned, but not that it depends on ozone (Douglass et al. 1997, Grooß et al., 2005 JAS, Grooß et al., 2011 etc;).

3/19: the phrase "super-recovery" is not ideal. It is sometimes used for ozone but not often for $ClONO_2$.

3/22: ozone has been monitored before the discovery of the ozone hole. (otherwise the ozone hole would not have been discovered).

4/18: "analysis" Do you mean retrieval of tracer profiles from the FTIR spectra?

4/22: how many layers exactly?

5/12: As you only show 14 coincident measurements within a period of 3 months, I would not call this chapter "Validation".

5/31: You show results from MLS Version 3.3 data. Why do you not use version 4.x?

6/8: How exactly do you identify the coincident CALIPSO PSCs? Orbit within a certain distance from Syowa station? PSCs of what type at what altitude?

6/12-15: This seems to be a speculation. It is not clear how this statement is proven in this context.

6/16: You use the term "temporal variations" several times, where I think it is (only) a time series.

6/16: define the expression $Cl_y$*.

6/17: add "for all ground-based and satellite based observations used in this study" (or similar).

8/5 ratios of each species with respect to $Cl_y$*.

9/1ff: It has not been said over what time the data were collected, how the anti-correlation was evaluated (MIPAS and MLS have different orbits).

page 9/line 1-6 (anti-correlation of MIPAS $ClONO_2$ and MLS ClO, fig 12): If is said that this is due to the PV (eq. latitude) dependence. Could it also be that this occurs because of the time dependence of the deactivation throughout the days 220-260? The slope of the regression line is not given in the text nor a statement of what would be expected from the model. What does this slope or correlation mean scientifically?

[Figure]

This correlation in the phase of chlorine deactivation is definitely no surprise.

9/24 "About 50This statement is inaccurate (do you mean at the 50 hPa level equally from the vortex edge to the core? Or also at the other levels? It is at least rather hard to read a number of percentage ozone loss from this figure.

9/33f "Inside the polar vortex,depletion of $O_3$, $NO_2$, $HNO_3$, and $ClONO_2$ continued" Most of the species are already near zero. It is not clear from the figure how you see continuing depletion. Also I would not expect further ozone depletion, if active chlorine returned more or less to zero.

10/23-28: The continuous loss of HCl seems to look differently than in the study by Grooß et al. that you mention. The conclusions of that study are not given properly. It is not clear, whether you include an additional process like ionisation by cosmic rays or cross vortex edge $ClONO_2$ flow due to Solomon et al. in your model. Or does the HCl just deplete because of the large diffusivity that is present due to the low model resolution (2.8×2.8 degrees)?

11/18f: I do not see this good agreement between model and FTIR in the figures.

figure 5: How was $T_{NAT}$ and $T_{ice}$ derived? What data for $HNO_3$ and $H_2O$ were used?

figure 12: There must be something wrong with the colour coding of the PV in the panels. I would expect about a factor 2 difference in PV between 18 and 22 km and also PV values significantly below -85 PVU at 22 km.

figure 17: "Three-hourly zonal-mean temporal variations" What do you mean by variations? It only looks like zonal mean values.

figures B1/B2: Here the model is labelled CTM. Is it really? In the paper you always talk about a CCM.

**[ACPD](about:blank)**

Interactive
comment

**Typos/grammar**

6/12: other reason -> "other reasons" or "an other reason"

8/32 shows

---

## Referee Comment (RC3) · Anonymous Referee #2 · 6 Jul 2018

there was a typo and a formatting issue in my review, therefore these two points are repeated:

**General**

fig 10/11 shows $Cl_y$ (MLS) relative to $Cl_y$* (MLS N2O), even though that should be described more clearly. In the context Cly (without star) is defined as ClO + ClONO$_2$ + HCl. However, there are no ClONO$_2$ observations of MLS. This needs to be clarified.

**Details**

9/24 "About 50% ozone depletion was seen throughout the polar vortex."
This statement is inaccurate. Do you mean at the 50 hPa level equally from the vortex edge to the core? Or also at the other levels? It is at least rather hard to read a number of percentage ozone loss from this figure.

---

## Author Comment (AC3) · 10 Sep 2018

**Comments to the revised version**

We thank the two reviewers for their comments that helped us to improve the paper. We addressed the points and changes made to the revised manuscript in the separate "Reply to Reviewers #1 and #2" which are posted in the interactive discussion.

5 For clarity, we here include a version of the paper in which the additions and changes to the previous ACPD manuscript are marked by colored letters. Red-colored changes are mainly for the comments by Reviewer-#1, and blue-colored changes are for those by Reviwer-#2.

**Temporal evolution of chlorine and minor species related to ozone depletion  at Syowa Station, Antarctica  during austral fall to spring in 2007 and 2011**

[revised manuscript text omitted]

$$ClONO_2 \ (g) + HCl \ (s, l) \rightarrow Cl_2 \ (g) + HNO_3 \qquad (R1)$$
$$ClONO_2 \ (g) + H_2O \ (l, s) \rightarrow HOCl \ (g) + HNO_3 \qquad (R2)$$

where g, s, and l represents the gas, solid, and liquid phases, respectively (Solomon et al., 1986; Nakajima et al., 2016).

Heterogeneous reactions on the surface of particles;

$$N_2O_5 \ (g) + HCl \ (s, l) \rightarrow ClNO_2 \ (g) + HNO_3 \qquad (R3)$$
$$HOCl \ (g) + HCl \ (s, l) \rightarrow Cl_2 \ (g) + H_2O \qquad (R4)$$

are responsible for additional chlorine activation. When solar illumination is available, $Cl_2$, HOCl, and $ClNO_2$ are photolyzed to produce chlorine atoms by reactions:

$$Cl_2 + hv \rightarrow Cl + Cl \qquad (R5)$$
$$HOCl + hv \rightarrow Cl + OH \qquad (R6)$$

$$ClNO_2 + hv \rightarrow Cl + NO_2. \qquad (R7)$$

The yielded chlorine atoms then start to destroy ozone catalytically through reactions:

$$Cl + O_3 \rightarrow ClO + O_2 \qquad (R8)$$

$$ClO + ClO + M \rightarrow Cl_2O_2 + M \qquad (R9)$$

5 $$Cl_2O_2 + hv \rightarrow Cl + ClOO \qquad (R10)$$

$$ClOO + M \rightarrow Cl + O_2 + M. \qquad (R11)$$

When the stratospheric temperature get warmer than nitric acid trihydrate (NAT) PSC saturation temperature, and no PSCs are present, gradual deactivation of chlorine starts to occur. Re-formation of $ClONO_2$ and HCl mainly occur through reactions:

10 $$ClO + NO_2 + M \rightarrow ClONO_2 + M \qquad (R12)$$

$$Cl + CH_4 \rightarrow HCl + CH_3. \qquad (R13)$$

The re-formation of $ClONO_2$ by reaction (R12) from active chlorine is much faster than that of HCl by reaction (R13), if there are enough $NO_x$ around (Mellqvist et al., 2002; Dufour et al., 2006). But the formation rates of $ClONO_2$ and HCl are also related to ozone concentration. Douglass et al. (1995) showed that HCl increases more rapidly in the Antarctic polar vortex

15 than in the Arctic polar vortex due to lower ozone concentrations in the Antarctic polar vortex. Low ozone reduces the rate of reaction (R8), and then Cl/ClO ratio becomes high. Low ozone also reduces the rate of the following reaction:

$$NO + O_3 \rightarrow NO_2 + O_2. \qquad (R14)$$

This makes NO/$NO_2$ ratio high and increases Cl/ClO ratio by the following reaction:

$$ClO + NO \rightarrow Cl + NO_2. \qquad (R15)$$

20 High Cl/ClO ratio leads rapid HCl formation by reaction (R13) and reduces the formation ratio of $ClONO_2$ by reaction (R12).

The processes of deactivation of active chlorine are different between typical conditions in the Arctic and those in the Arctic. In the Antarctic, the temperature cools below the PSC formation threshold in the whole area of the polar vortex in most years, and almost complete denitrification and chlorine activation occur (WMO, 2007), followed by severe ozone depletion in spring. In the chlorine reservoir recovery phase, HCl is mainly formed by reaction (R13) due to the

25  lack of ozone (typically less than 0.5 ppmv) by the mechanism described in the previous paragraph (Grooß et al., 2011).

On the other hand, in the Arctic, typically less PSC formation occurs in the polar vortex  due to generally higher stratospheric temperatures (~10-15K in average) compared with that of Antarctica.

[revised manuscript text omitted]

The retrievalanalysis of the FTIR spectra was done with SFIT2 Version 3.92 program (Rinsland et al., 1998; Hase et al., 2004). SFIT2 retrieves a vertical profile of trace gases using an optimal estimation formulation of Rodgers (2000), implemented with a semi-empirical method which was originally developed for microwave measurements (Parrish et al., 1992; Connor et al., 1995). The SFIT2 forward model fully describes the FTIR instrument response, with absorption coefficients calculated using the algorithm of Norton and Rinsland (1991). The atmosphere is constructed with 4730 or more layers from the ground to 100 km, using the FSCASATM (Gallery et al., 1983) program for atmospheric ray-tracing to account for refractive bending. The retrieval parameters for each gas are shown in Table 2. Temperature and pressure profiles between 0 and 30 km are taken by the Rawin sonde observations flown from Syowa Station on the same day by the Japanese Meteorological Agency (JMA), while values between 30 and 100 km are taken from the COSPAR International Reference Atmosphere 1986 (CIRA-86) standard atmosphere profile (Rees et al., 1990).

We retrieved vertical profiles of $O_3$, HCl, and $HNO_3$ from the solar spectra. We used monthly averaged ozonesondes profiles (0-30 km) and Improve Limb Atmospheric Spectrometer-II (ILAS-II) (Nakajima, 2006; Nakajima et al., 2006; Sugita et al., 2006) profiles (30-100 km) for the a priori of $O_3$, monthly averaged profiles from ILAS-II for $HNO_3$ and monthly averaged profiles from HALOE (Anderson et al., 2000) for HCl. Typical averaging kernels of the SFIT2 retrievals for $O_3$, $HNO_3$, and HCl are shown in Figures 1(a), (b), and , 1(b), and 1(c), respectively.

**2.2 Satellite measurements**

The Earth Observing System (EOS) Microwave Limb Sounder (MLS) onboard the Aura satellite was launched on 15 July 2004, to monitor several atmospheric chemical species in upper troposphere to mesosphere (Waters et al., 2006). The Aura orbit is sun-synchronous at 705 km altitude with an inclination of 98°, 13:45 ascending (north-going) equator-crossing time, and 98.8-min period. Vertical profiles are measured every ~165 km along the suborbital track, horizontal resolution is ~200-600 km along-track, ~3-10 km across-track, and vertical resolution is ~3-4 km in the lower to middle stratosphere (Froidevaux et al., 2006).

ClO, HCl, and $HNO_3$ profiles used in this study were taken from Aura/MLS version 3.3 data (Liversey et al., 2006; Santee et al., 2011; Ziemke et al., 2011; Liversey et al., 2013). The MLS data was selected whose measurement location is within 300 km radius from Syowa Station and within ±6 hours of the FTIR measurement.

–Michelson Interferometer for Passive Atmospheric Sounding (MIPAS) is a Fourier transform spectrometer sounding the thermal emission of the earth's atmosphere between 685 and 2410 cm$^{-1}$ (14.6-4.15 µm) in limb geometry (Fischer and Oelhaf, 1996). The maximum optical path difference of MIPAS is 20 cm. The field-of-view of the instrument at the tangent points is

about 3 km in the vertical and 30 km in the horizontal.  In the standard observation mode in one limb-scan, 17 tangent points are observed with nominal altitudes 6, 9, 12, …, 39, 42, 47, 52, 60, and 68 km.  In this mode, about 73 limb scans are recorded per orbit.  The measurements of each orbit cover nearly the complete latitude range from about 87°S to 89°N.  MIPAS was put on board the European Environmental Satellite (Envisat), which was launched on 1 March 2002, and was put into a polar sun-synchronous orbit at an altitude of about 800 km with an inclination of 98.55° (von Clarmann et al., 2003).  On its descending node, the satellite crosses the equator at 10:00 local time.  Envisat performs 14.3 orbits per day, which results in a good global coverage.  $ClONO_2$ profiles which we used in this study were taken from Envisat/MIPAS IMK/IAA version V5R_CLONO2_220 and V5R_CLONO2_222 (Höpfner et al., 2007).  Thewhose measurement criteria of the MIPAS data used in this study are the same as that of Aura/MLS.

    –The Cloud-Aerosol-Lidar and Infrared Pathfinder Satellite Observations (CALIPSO) satellite was launched on 28 April 2006.  On CALIPSO satellite, Cloud-Aerosol Lidar with Orthogonal Polarization (CALIOP) instrument was on board, to monitor aerosols, clouds, and polar stratospheric clouds (PSCs) (Pitts et al., 2007).  CALIOP is a two-wavelength, polarization sensitive lidar that provides high vertical resolution profiles of backscatter coefficient at 532 and 1064 nm, as well as two orthogonal (parallel and perpendicular) polarization components at 532 nm (Winker et al., 2007).  
[revised manuscript text omitted]
 altitude of 18 km was selected because it was the altitude where the most ozone depletion was occurred. The altitude of 22 km was selected to show the difference of the behavior of minor atmospheric species with 18 km where about half of the ozone was depleted.

The common features found in both 2007 and 2011 at both altitudes of 18 and 22 km can be summarized as follows: ClO was enhanced in August and September and the day-to-day variations were large over this period. HCl was almost zero from late June to early September and the day-to-day variations were small over this period (larger values are related to the polar vortex boundary). HCl and $ClONO_2$ decreased first, then ClO started to increase in winter, while HCl increases and ClO decreases were synchronized in spring. $Cl_y*$ gradually increased in the polar vortex from late autumn to spring. The $Cl_y*$

value became larger compared with its mixing ratio outside of the polar vortex in spring. $O_3$ decreased from July to late September when ClO was present. $HNO_3$ showed large decreases from June to July, and then gradually increased in summer. Day-to-day variations of $HNO_3$ from June to August were large.

The following characteristics are evident especially at 18 km (Figures 46 and 57). The day-to-day variations of HCl from late June to early September were as small as 0-0.3 ppbv. The recovered values of HCl in spring were larger than those before winter and those outside the polar vortex during the same period. $ClONO_2$ kept near zero even after ClO disappeared, and did not recover to the level before winter until spring. $O_3$ gradually decreased from values of 2.5-3 ppmv before winter to values less than one fifth, 0.3-0.5 ppmv, in October.

The following characteristics are evident only at 22 km (Figures 68 and 79). The day-to-day variation of HCl from late June to early September were 0-1 ppbv, larger than those at 18 km. The recovered values of HCl in spring were nearly the same as those before winter (around 2.2 ppbv). $ClONO_2$ recovered to larger values than those before winter after ClO disappeared. From winter to spring, $O_3$ gradually decreased, but the magnitude of the decrease was much smaller than that at 18 km.

As for the temporal increase of $ClONO_2$ in spring during the ClO decreasing phase, we can see a peak of 1.5 ppbv at 18 km in 2011, and at 22 km in both 2007 and 2011 around day 270, but we see no temporal increase of $ClONO_2$ at 18 km in 2007.

Figure 79 shows that temporal ClO enhancement and decrease of $O_3$, $ClONO_2$, and $HNO_3$ occurred in early winter (day 150-170) at 22 km in 2011. This small ozone depletion event before winter might be due to an airmass movement from the polar night area to a sunlit area at lower latitudes. Table 4 summarized the characteristics of variation of minor atmospheric species for 2007 and 2011 at altitudes of 18 and 22 km.

**5. Discussion**

The ratios of observed HCl, $ClONO_2$, ClO, and $Cl_y$ with respect to $Cl_y^*$ were calculated to discuss the temporal variations of the chlorine partitioning. Here, observed $Cl_y$ is determined as:

$$Cl_y \text{ (FTIR)} = HCl \text{ (FTIR)} + ClONO_2 \text{ (MIPAS)} + ClO \text{ (MLS)}. \qquad (3)$$

$$Cl_y \text{ (MLS)} = HCl \text{ (MLS)} + ClONO_2 \text{ (MIPAS)} + ClO \text{ (MLS)}. \qquad (4)$$

Figures 810 and 911 show the time seriestemporal variations of the ratios of each chlorine species with respect to $Cl_y^*$ in 2007 (a) and in 2011 (b) at 18 km and 22 km, respectively. For both in 2007 and 2011 at 18 km (Figure 810), the ratio of HCl was 0.6-0.8 and the ratio of $ClONO_2$ was 0.2-0.3 before winter (day 130-140). The partitioning of HCl was three times larger than that of $ClONO_2$ at that time. The ratio of ClO increased to 0.5-0.6 during the enhanced period (day 240-260). The ratio of HCl was 0-0.2 and the ratio of $ClONO_2$ was 0-0.6 during this same period. $ClONO_2$ shows negative correlation with ClO, while HCl kept low even when ClO was low during this period. This negative correlation is shown in Figure 102 later. When ClO was enhanced, the $O_3$ amount gradually decreased, and finally reached <0.5 ppmv (>80% destruction) in October (day

280) (See Figures 6 and 7). The ratios became 0.9-1.0 for HCl and 0-0.1 for ClONO$_2$ after the recovery in spring (after day 290), indicating that almost all chlorine reservoir species became HCl via reaction (13), due to the lack of O$_3$ and NO$_2$ during this period. The ratios of Cl$_y$ (FTIR) and Cl$_y$ (MLS) were both around 0.7 at the time of ClO enhanced period (day 230-260). The remaining chlorine are thought to be Cl$_2$O$_2$, which will be shown in model simulation in Section 5 later. The ratio of Cl$_y$  became close to 1 after the recovery period (after day 280).

For both in 2007 and 2011 at 22 km (Figure 9), the ratio of HCl was 0.4-0.9 and the ratio of ClONO$_2$ was 0.2-0.3 before winter (day 110-140). The partitioning of HCl was two to three times larger than that of ClONO$_2$. The ratio of ClO increased to 0.6-0.7 during the enhanced period (day 220-240). The ratio of HCl was 0-0.3 and the ratio of ClONO$_2$ was 0-0.6 during this period. ClONO$_2$ shows negative correlation with ClO, while HCl kept low even when ClO was low during this period like the case at 18 km. The O$_3$ amount gradually decreased during the ClO enhanced period but remained >1.5 ppmv (less than half destruction) at this altitude (See Figures 6 and 7). When the ClO enhancement ended, temporal increase  of ClONO$_2$ up to a ratio of 0.5 occurred in early spring (day 260-280). Then, the reservoir ratios became 0.6-0.8 for HCl and 0.2-0.4 for ClONO$_2$ in spring (after day 280). This phenomenon shows that more chlorine deactivation via reaction (12) occurred towards ClONO$_2$ at 22 km rather than at 18km. This is attributed to the existence of O$_3$ and NO$_2$ during this period at 22 km, which was different from the case at 18 km. The ratios of Cl$_y$ (FTIR) and Cl$_y$ (MLS) were both around 0.8 at the time of ClO enhanced period (day 230-250). The remaining chlorine are thought to be Cl$_2$O$_2$. The ratio of Cl$_y$  became around 1.1 after the recovery period (after day 270). The reason why observed Cl$_y$ values exceed calculated Cl$_y$* values might be due to the difference in N$_2$O-Cl$_y$ correlation at this altitude from the one in  the equation (2).

In 2011 at 18 km (Figure 8), another temporal increase of ClONO$_2$ up to a ratio of 0.6 occurred in early spring (around day 280) in accordance with HCl increase, then the ClONO$_2$ amount gradually decreased to nearly zero after late October (day 300-). This temporal increase in ClONO$_2$ could be attributed to temporal change of the location of Syowa Station in the polar vortex. Although Syowa Station was always located inside the polar vortex from day 195 to 350, the difference between the equivalent latitude over Syowa Station and that at inner edge became less than 10 degrees at around day 280, while it was typically between 15 and 20 degrees in other days. O$_3$ and HNO$_3$ showed higher values around day 280 (see Figure 5), indicating that Syowa Station was located close to the boundary region at this period. Therefore, the temporal increase of ClONO$_2$ in 2011 at 18 km was attributed to spatial variation, not to chemical evolution.

**5.  Discussion**

Figure 10 show the correlation between ClO and ClONO$_2$ during the ClO enhanced period (August 8-September 17; day 220-260) at 18 km in 2007 (a) and 2011 (b), and at 22 km in 2007 (c) and 2011 (d). Note that MLS ClO and MIPAS ClONO$_2$

data were sampled on the same day at the nearest orbit for both satellites to Syowa Station. The maximum differences between these two satellites' observational times and locations are 9.0 hours in time and 587 km in distance. Mean differences are 6.8 hours in time and 270 km in distance, respectively. Solid lines show regression lines obtained by RMA (Reduced Major Axis) regression. Negative correlations of slope ~ -1.0 between ClO and $ClONO_2$ are seen in all figures. The cause of this negative

5   correlation might be due to the variation of the relative distance between Syowa Station and the boundary region of the polar vortex. When Syowa Station was located deep inside the polar vortex, there was more ClO and less $ClONO_2$. On the contrary when Syowa Station was located near the vortex edge, there was less ClO and more $ClONO_2$. The potential vorticities (PV) over Syowa Station shown by color code generally show this tendency, that warm colored higher PV points are located more towards bottom right-hand side. This is further confirmed by 3D model simulation as is shown later in this section.

10     Figures 11 shows simulated mixing ratios of $O_3$ , $NO_2$ , $HNO_3$ , ClO , HCl , and $ClONO_2$  by the MIROC3.2 Chemistry-Climate Model (CCM) at 50 hPa (~18 km) for June 24 (day 175), September 1 (day 244), September 6 (day 249), and October 6 (day 279) in 2007. For a description of the MIROC3.2 CCM, please see Appendix A for detail. The location of Syowa Station is shown by a white star in each panel. Direct comparisons of mixing ratios of ClO, HCl, $ClONO_2$, $Cl_y$, and $O_3$ measured by FTIR and MLS, and modeled by MIROC3.2 CCM in 2007 and 2011 at

15   18 and 22 km are shown in Appendix B. In general, the model results are in good agreement with FTIR and satellite observations (Figure B1). Hereafter, the result of MIROC3.2 CCM is discussed.

    On June 24 (day 175), stratospheric temperatures over Antarctica were already low enough to allow PSCs to form. Consequently, $NO_2$ and $HNO_3$ in the polar vortex condensed onto PSCs . Note that the depleted area of $NO_2$ was greater than that of $HNO_3$. This might be due to reaction (R12) that converts ClO and $NO_2$ to $ClONO_2$ at the

20   edge of the polar vortex, which is shown by the enhanced $ClONO_2$ area at the vortex edge in Figure 11. Also, HCl and $ClONO_2$ are depleted in the polar vortex due to the heterogeneous reactions (R1), (R2), (R3), and (R4) on the surface of PSCs and aerosols . Some HCl remains near the core of the polar vortex , because the initial amount of the counter-part of heterogeneous reaction (R1) ($ClONO_2$) was less than that of HCl (see Figure 4 and/or 5). The $O_3$ amount was only slightly depleted within the polar vortex on this day .

25   On September 1 (day 244), amounts of $NO_2$ , $HNO_3$ , HCl , and $ClONO_2$  all show very depleted values in the polar vortex. The amount of ClO  shows some enhanced values at the outer part of the polar vortex. Development of ozone depletion was seen in the polar vortex. Note that $ClONO_2$  shows enhanced values around the boundary region of the polar vortex. This might be due to the reaction (R12) at this location. On this day (day 244), Syowa Station was located inside the polar vortex close to the

30   inner vortex edge, where ClO was smaller and $ClONO_2$ was greater than the values deep inside the polar vortex.

    On September 6 (day 249), most features were the same as on September 1, but the shape of the polar vortex was different. Consequently, Syowa Station was located deep inside the polar vortex, where ClO was greater and $ClONO_2$ was smaller than the values around the boundary region of the polar vortex. Hence, the negative correlation between ClO and

$ClONO_2$ seen in Figure 12 was due to variation of the relative distance between Syowa Station and the inner edge of the polar vortex.

On October 6 (day 279), ClO enhancement has almost disappeared .  Inside the polar vortex,  $O_3$ , $NO_2$ , $HNO_3$ , and $ClONO_2$  showed very low values.  Ozone was almost fully destroyed at this altitude in the polar vortex.  However, the amount of HCl  increased deep inside the polar vortex.  This might be due to the recovery of HCl by reaction (R13) deep inside the polar vortex, where there is no $O_3$ or $NO_2$ left and reaction (R13) was favoured compared with reaction (R12).  At Syowa Station, the amount of HCl was several times greater than that of $ClONO_2$ on this day.

Three-hourly time series of zonal-mean active chlorine species, $Cl_2O_2$ (b), $Cl_2$ (c), ClO (d), and their sum (ClO+2*$Cl_2O_2$+2*$Cl_2$) (a), and chlorine reservoir species HCl (e) and $ClONO_2$ (f) modeled by MIROC3.2 CCM at 68.4ºS, 71.2ºS, 76.7ºS, and 87.9ºS are plotted in Figure 17.  The dates on which the distribution of each species is shown in Figure 1 are indicated by vertical dotted lines.  In this figure, it is shown that HCl and $ClONO_2$ rapidly decreased at around day 130 at 87.9ºS, when PSCs started to form in the Antarctic polar vortex (Figures 17(e) and 17(f)).  The decrease of HCl stopped when the counter-part of the heterogeneous reaction (R1) was missing at around day 140.  Consequently, $Cl_2$ was formed (Figure 17(c)).   Similar chlorine activation was seen at 76.7ºS about 5-10 days later than at 87.9ºS.  Gradual conversion from $Cl_2$ into $Cl_2O_2$ (ClO-dimer) was seen at all latitudes at around day 150-160 (Figures 17(b) and 17(c)) through reaction (R5), (R8), and (R9).  At 87.9ºS, conversion from $Cl_2$ to $Cl_2O_2$ was slow, due to lack of sunlight which is needed for reaction (R5).  Increase of ClO occurred much later after winter (day 190 or later), because sun light is needed to form ClO by reactions (R5) and (R8) in the polar vortex (Figure 17(d)).  Nevertheless, there were some enhancement of ClO in early winter, day 175, simulated at the edge of the polar vortex (Figure 1) where there was some sunlight available due to the distortion of the shape of the polar vortex.  Increase of ClO occurred from lower latitude (68.4ºS) at around day 195, towards higher latitude (87.9ºS) at around day 255 (Figure 17(d)).  Diurnal variation of ClO was also seen at latitudes between 68.4ºS and 76.7ºS.  When stratospheric temperature increased above PSC saturation temperature at around day 270 (Figure 5(a)), chlorine activation ended, and ClO was mainly converted into HCl at all latitudes inside the polar vortex (Figures 17(d) and 17(e)).  This is because reaction (R13) occurs more frequently than reaction (R12) inside the polar vortex due to the depleted $O_3$ amount there as was described in Section 1 (Douglass et al., 1995).

[revised manuscript text omitted]

Grooß, J. -U., Brautzsch, K., Pommrich, R., Solomon, S., and Müller, R.: Stratospheric ozone chemistry in the Antarctic: what
   determines the lowest ozone values reached and their recovery?, Atmos. Chem. Phys., 11, 12,217-12,226, doi:10.5194/acp-
   11-12217-2011, 2011.

[revised manuscript text omitted]

20    von Clarmann, T., Glatthor, N., Grabowski, U., Höpfner, M., Kellmann, S., Kiefer, M., Linden, A., Tsidu, G. M., Milz, M., Steck, T., Stiller, G. P., Wand, D. Y., and Fischer, H., Retrieval of temperature and tangent altitude pointing from limb emission spectra recorded from space by the Michelson Interferometer for Passive Atmospheric Sounding (MIPAS), J. Geophys. Res., 108, 4736, doi:10.1029/2003JD003602, 2003.

Waters, J. W., et al.: The Earth Observing System Microwave Limb Sounder (EOS MLS) on the Aura satellite, IEEE Trans.
25    Geosci. Remote Sens., 44, 1075-1092, 2006.

Webster, C. R., May, R. D., Toohey, D. W., Avallone, L. M., Anderson, J. G., Newman, P., Lait, L., Schoeberl, M. R., Elkins, J. W., and Chan, K. R.: Chlorine chemistry on polar stratospheric cloud particles in the Arctic winter, Science, 261, 1130-1134, 1993.

Winker, D. M., McGill, M., and Hunt, W. H.: Initial performance assessment of CALIOP, Geophys. Res. Lett., 34, L19803,
30    doi:10.1029/2007GL30135, 2007.

WMO: Scientific Assessment of Ozone Depletion: 2006, Global Ozone Research and Monitoring Project—Report No. 50, 572 pp., Geneva, Switzerland, 2007.

WMO: Scientific Assessment of Ozone Depletion: 2010, Global Ozone Research and Monitoring Project—Report No. 52, 516 pp., Geneva, Switzerland, 2011.

WMO: Scientific Assessment of Ozone Depletion: 2014, Global Ozone Research and Monitoring Project—Report No. 55, 416 pp., Geneva, Switzerland, 2014.

Wood, S. W., Bodeker, G. E., Boyd, I. S., Jones, N. B., Connor, B. J., Johnston, P. V., Matthews, W. A., Nichol, S. E., Murcray, F. J., Nakajima, H., and Sasano, Y.: Validation of version 5.20 ILAS $HNO_3$, $CH_4$, $N_2O$, $O_3$, and $NO_2$ using ground-based measurements at Arrival Heights and Kiruna, J. Geophys. Res., 107, 8208, 10.1029/2001jd000581, 2002.

Wood, S. W., Batchelor, R. L., Goldman, A., Rinsland, C. P., Connor, B. J., Murcray, F. J., Stephen, T. M., and Heuff, D. N.: Ground-based nitric acid measurements at Arrival Heights, Antarctica, using solar and lunar Fourier transform infrared observations, J. Geophys. Res., 109, D18307, 10.1029/2004jd004665, 2004.

Yang, E.-S., Cunnold, D. M., Newchurch, M. J., Salawitch, R. J., McCormick, M. P., Russell III, J. M., Zawodny, J. M., and Oltmans, S. J.: First stage of Antarctic ozone recovery, J. Geophys.. Res., 113, D20308, 10.1029/2007JD009675, 2008.

Ziemke, J. R., Chandra, S., Labow, G. J., Bhartia, P. K., Froidevaux, L., and Witte, J. C.: A global climatology of tropospheric and stratospheric ozone derived from Aura OMI and MLS measurements, Atmos. Chem. Phys., 11, 9237-9251, 10.5194/acp-11-9237-2011, 2011.

**Tables**

**Table 1.  FTIR observation dates at Syowa Station in 2007 and 2011**

| Month | Dates (2007) | Dates (2011) | Number of days inside the polar vortex (2007/2011) | Number of days in the boundary region of the polar vortex (2007/2011) | Number of days outside the polar vortex (2007/2011) | Number of measurement days (2007/2011) |
|---|---|---|---|---|---|---|
| **March** | 25 | | 0 / 0 | 0 / 0 | 1 / 0 | 1 / 0 |
| **April** | 1, 3, 4, 5, 8, 24, 26, 28 | | 0 / 0 | 0 / 0 | 8 / 0 | 8 / 0 |
| **May** | 8, 9, 10, 13, 14, 15, 20, 21, 22 | | 7 / 0 | 0 / 0 | 2 / 0 | 9 / 0 |
| **June** | | | 0 / 0 | 0 / 0 | 0 / 0 | 0 / 0 |
| **July** | 29, 30 | | 0 / 0 | 2 / 0 | 0 / 0 | 2 / 0 |
| **August** | 1, 8, 9, 10, 24, 25, 26, 28, 29 | | 8 / 0 | 1 / 0 | 0 / 0 | 9 / 0 |
| **September** | 1, 4, 5, 6, 7, 8, 16, 18, 23, 26, 27, 30 | 25, 29, 30 | 12 / 3 | 0 / 0 | 0 / 0 | 12 / 3 |
| **October** | 6, 10, 11, 14, 19, 20, 25, 26, 27 | 1, 3, 4, 8, 11, 22, 23, 24, 26 | 9 / 9 | 0 / 0 | 0 / 0 | 9 / 9 |
| **November** | 2, 3, 5, 6, 7, 8, 9, 10, 11, 16, 17, 18, 19, 21, 27, 29, 30 | 1, 2, 3, 9, 11, 16, 19 | 12 / 7 | 1 / 0 | 4 / 0 | 17 / 7 |
| **December** | 4, 7, 8, 9, 13, 15, 16, 17, 20, 22, 29 | | 8 / 0 | 0 / 0 | 3 / 0 | 11 / 0 |
| **Total** | | | 56 / 19 | 4 / 0 | 18 / 0 | 78 / 19 |

**Table 2. Retrieval parameters of SFIT2**

| Species | O$_3$ | HNO$_3$ | HCl |
|---|---|---|---|
| Spectroscopy | HITRAN 2008 | HITRAN 2008 | HITRAN 2008 |
| PT Profile | Daily sonde (0-30 km) CIRA 86 (30-100 km) | Daily sonde (0-30 km) CIRA 86 (30-100 km) | Daily sonde (0-30 km) CIRA 86 (30-100 km) |
| A priori profiles | Monthly averaged by ozonesonde (0-30 km) & ILAS-II (30-100 km) | Monthly averaged by ILAS-II | Monthly averaged by HALOE |
| Microwindows (cm$^{-1}$) | 1002.578 – 1003.500 1003.900 – 1004.400 1004.578 – 1005.000 | 867.000 – 869.591 872.800 – 874.000 | 2727.730 – 2727.830 2775.700 – 2775.800 2925.800 – 2926.000 |
| Retrieved interfering species | O$_3$ (668), O$_3$ (686), CO$_2$, H$_2$O | H$_2$O, OCS, NH$_3$, CO$_2$, C$_2$H$_6$ | CO$_2$, H$_2$O, O$_3$, NO$_2$ |

**Table 3. Summary of validation results of FTIR profiles compared with ozonesonde and Aura/MLS measurements, and possible Aura/MLS biases from literatures**

| | D-value (%) 18-22 km | Min/Max (%) 18-22 km | Agreement 15-25 km (ppmv/ppbv) | Literature values |
|---|---|---|---|---|
| $O_3$ | +6.2 | -10.4/+19.2 | -0.02~+0.40 | |
| $HNO_3$ | +13.2 | +0.2/+21.9 | -0.56~+0.57 | Aura/MLS no bias with errors (0.6 ppbv) (Livesey et al., 2011) |
| $HCl$ | -9.7 | -14.6/-3.0 | -0.2~+0.09 | Aura/MLS > HALOE by 10-15%, precision 0.2-0.6 ppbv (Livesey et al., 2013) |

**Table 4. Summary of minor atmospheric species variations**

| Altitude | 18 km | | 22 km | |
|---|---|---|---|---|
| Year | 2007 | 2011 | 2007 | 2011 |
|  |  |  |  |  |
| Period of enhanced ClO (day) | 230-260 | 230-260 | 220-240 | 230-250 |
| Variation when ClO enhanced (ppbv) | 0–1.3 | 0–1.5 | 0–2.2 | 0–2.2 |
| HCl value before winter (ppbv) | 1.5–1.8 | 1.2–1.6 | 2.1–2.4 | 1.8–2.2 |
| HCl starting-ending day of decrease (day) | 140-180 | 140-180 | 130-180 | 140-170 |
| Variation when HCl ~ 0 (ppbv) | 0–0.3 | 0–0.3 | 0.1–1.0 | 0.1–0.9 |
| HCl starting-ending day of increase (day) | 250-300 | 250-300 | 240-280 | 240-300 |
| HCl Value after increase (ppbv) | 2.6–3.0 | 2.5–2.8 | 2.1–2.4 | 2.0–2.5 |
| HCl Value outside polar vortex (ppbv) | 1.5–2.0 | 1.0–1.8 | 1.5–2.0 | 1.5–2.0 |
| $ClONO_2$ Value before winter (ppbv) | ~0.5 | ~0.4 | 0.6–0.9 | 0.6–0.7 |
| Variation when $ClONO_2$~0 (ppbv) | 0–1.5 | 0–1.5 | 0–2.0 | 0–2.0 |
| Day of $ClONO_2$ enhancement | - | 270-300 | 270-280 | 270–280 |
| Value of $ClONO_2$ enhancement (ppbv) | - | 1.5 | 1.5 | 1.5 |
| $ClONO_2$ value after enhancement (ppbv) | 0–0.3 | 0–0.2 | 0.8–1.3 | 0.8–1.1 |
| $ClONO_2$ value outside polar vortex (ppbv) | 0.3–0.4 | 0.2–0.3 | 0.5–0.7 | 0.6–0.8 |
| $O_3$ value before winter (ppmv) | 2.5 | 2.5 | 4.0 | 4.0 |
| $O_3$ starting-ending day of decrease (day) | 190-280 | 200-270 | 170-260 | 170-270 |
| $O_3$ minimum value (ppmv) | 0.3 | 0.5 | 2.0 | 1.0 |
| $O_3$ value after recovery (ppmv) | 0.8 | 0.8 | 2.4–4.0 | 2.0–3.5 |
| $HNO_3$ value before winter (ppbv) | 6-10 | 8-10 | 15-16 | 13–15 |
| $HNO_3$ starting-ending day of decrease (day) | 160-190 | 150-180 | 140-180 | 150-180 |
| $HNO_3$ minimum value (ppbv) | 0 | 0 | 2 | 1 |
| $HNO_3$ value after recovery (ppbv) | 3–4 | 3–4 | 4–6 | 4–5 |
|  |  |  |  |  |

**Table 5. Summary of HCl variations**

| Altitude | 18 km | | 22 km | |
|---|---|---|---|---|
| Year | 2007 | 2011 | 2007 | 2011 |
| Value before winter (ppbv) | 1.5–1.8 | 1.2–1.6 | 2.1–2.4 | 1.8–2.2 |
| Starting day of decrease (day) | 140 | 140 | 130 | 140 |
| Ending day of decrease (day) | 180 | 180 | 180 | 170 |
| Variation when HCl ~ 0 (ppbv) | 0–0.3 | 0–0.3 | 0.1–1.0 | 0.1–0.9 |
| Day when HCl starts to increase | 250 | 250 | 240 | 240 |
| Day when HCl increase stops | 300 | 300 | 280 | 300 |
| Value after increase (ppbv) | 2.6–3.0 | 2.5–2.8 | 2.1–2.4 | 2.0–2.5 |
| Value outside polar vortex (ppbv) | 1.5–2.0 | 1.0–1.8 | 1.5–2.0 | 1.5–2.0 |

**Table 6.  Summary of ClONO$_2$ variations**

| Altitude | 18 km | | 22 km | |
|---|---|---|---|---|
| Year | 2007 | 2011 | 2007 | 2011 |
| Value before winter (ppbv) | ~0.5 | ~0.4 | 0.6–0.9 | 0.6–0.7 |
| Starting day of decrease (day) | 160 | (after 150) | 160 | between 140 – 150 |
| Ending day of decrease (day) | (after 170) | - | - | 160 |
| Variation when ClONO$_2$~0 (ppbv) | 0–1.5 | 0–1.5 | 0–2.0 | 0–2.0 |
| Day of ClONO$_2$ enhancement | - | 270–300 | 270–280 | 270–280 |
| Value of ClONO$_2$ enhancement (ppbv) | - | 1.5 | 1.5 | 1.5 |
| Value after enhancement (ppbv) | 0–0.3 | 0–0.2 | 0.8–1.3 | 0.8–1.1 |
| Value outside polar vortex (ppbv) | 0.3–0.4 | 0.2–0.3 | 0.5–0.7 | 0.6–0.8 |

**Table 7. Summary of Cl$_y$* variations**

| Altitude | 18 km | | 22 km | |
|---|---|---|---|---|
| Year | 2007 | 2011 | 2007 | 2011 |
| Value before winter (ppbv) | 2.3 | 2.2 | 2.8 | 2.9 |
| Value after winter (ppbv) | 3.1 | 2.9 | 3.2 | 3.2 |
| Value outside polar vortex (ppbv) | 1.5–2.0 | 1.5–2.0 | 2.4–2.6 | 2.2–2.8 |

**Table 8. Summary of O$_3$ variations**

| Altitude | 18 km | | 22 km | |
|---|---|---|---|---|
| Year | 2007 | 2011 | 2007 | 2011 |
| Value before winter (ppmv) | 2.5 | 2.5 | 4.0 | 4.0 |
| Starting day of decrease (day) | 190 | 200 | 170 | 170 |
| Ending day of decrease (day) | 280 | 270 | 260 | 270 |
| Minimum value (ppmv) | 0.3 | 0.5 | 2.0 | 1.0 |
| Value after recovery (ppmv) | 0.8 | 0.8 | 2.4–4.0 | 2.0–3.5 |

**Table 9. Summary of HNO$_3$ variations**

| Altitude | 18 km | | 22 km | |
|---|---|---|---|---|
| Year | 2007 | 2011 | 2007 | 2011 |
| Value before winter (ppbv) | 6-10 | 8-10 | 15-16 | 13-15 |
| Starting day of decrease (day) | 160 | 150 | 140 | 150 |
| Ending day of decrease (day) | 190 | 180 | 180 | 180 |
| Minimum value (ppbv) | 0 | 0 | 2 | 1 |
| Value after recovery (ppbv) | 3-4 | 3-4 | 4-6 | 4-5 |
| Variation during decrease (ppbv) | 0-11 (day 200-250) | 0-8 (day 210-240) | 2-14 (day 180-240) | 1-11 (day 200-240) |
| Variation during decrease (ppbv) | 1-5 (day 250-300) | 2-4 (day 240-300) | 2-6 (day 240-320) | 2-8 (day 240-300) |

[Figure]

Figure 1. Averaging kernel functions of the SFIT2 retrievals for $O_3$ (a), $HNO_3$ (b), and HCl (c).

[Figure]

Figure 2. Absolute (a) and percentage (b) differences of $O_3$ profiles retrieved from FTIR measurements and those from ozonesonde measurements. Horizontal bars indicate the standard deviation of differences at each altitude. Absolute (c) and percentage (d) differences of $HNO_3$ profiles retrieved from FTIR measurements and those from Aura/MLS measurements. Absolute (e) and percentage (f) differences of HCl profiles retrieved from FTIR measurements and those from Aura/MLS measurements.

[Figure]

Figure 35. Time seriesemporal variation of temperatures at 18 km in (a) 2007 and (b) 2011, and at 22 km in (ca) 2007 and (db) 2011 over Syowa Station using ERA-Interim data. Approximate saturation temperatures for nitric acid trihydrate PSC ($T_{NAT}$) and ice PSC ($T_{ICE}$) calculated by assuming 6 ppbv $HNO_3$ and 4.5 ppmv $H_2O$ are also plotted ion the figures by dotted lines. Dates when PSCs were observed over Syowa Station are indicated by asterisks on the bottom of the figures.

[Figure]

Figure 46. Time seriesemporal variations of (a) HCl, ClONO₂, ClO, Cl$_y$*, (b) O₃, and HNO₃ mixing ratios at 18 km in 2007 over Syowa Station. O₃(FTIR), HCl(FTIR), and HNO₃(FTIR) were measured by FTIR at Syowa Station, while HCl(MLS), ClO, and HNO₃(MLS) were measured by Aura/MLS. O₃(sonde) was measured by ozonesonde. ClONO₂ was measured by Envisat/MIPAS. Cl$_y$* is calculated from N₂O value. See text in detail. The unit ofin O₃ is ppmv and the other gases are ppbv. The Llight and dark shaded areas are the days when Syowa Station iwas at the boundary region and outside the polar vortex, respectively.

[Figure]

Figure 57.  Same as Figure 46 but in 2011.

[Figure]

Figure 68. Same as Figure 46 but at 22 km.

[Figure]

Figure 79.  Same as Figure 57 but at 22 km.

[Figure]

Figure 810. Time seriesemporal variations of the ratios of HCl, ClONO$_2$, ClO, and Cl$_y$(=HCl+ClONO$_2$+ClO) to total chlorine (Cl$_y$*) over Syowa Station at 18 km in (a) 2007 and in (b) 2011. Shaded areas are the same as Figure 46.

[Figure]

Figure 911.  Same as Figure 810 but at 22 km.

[Figure]

Figure 10. Scatter plot between ClO (Aura/MLS) and ClONO$_2$ (Envisat/MIPAS) mixing ratios between August 8 and September 17 (day  220 – 260) at 18 km and 22 km in 2007 and 2011. Solid lines are regression lines obtained by RMA (Reduced Major Axis) regression. Color represents the potential vorticity over Syowa Station on that day.

[Figure]

Figure 11.  Polar southern hemispheric plot for simulated mixing ratios of O₃ , NO₂ , HNO₃ , ClO , HCl , and ClONO₂  by a MIROC3.2 chemistry-climate model (CCM) at 50 hPa for June 24 (day 175), September 1 (day 244), September 6 (day 249), and October 6 (day 279), 2007.  The location of Syowa Station is shown by white star in each panel.

[Figure]

Figure 127. Three-hourly zonal-mean time seriestemporal variations of MIROC3.2 CCM outputs for (a) ClO+2*Cl$_2$O$_2$+2*Cl$_2$, (b) Cl$_2$O$_2$, (c) Cl$_2$, (d) ClO, (e) HCl, and (d) ClONO$_2$ during day number 120 – 300 at 50 hPa in 2007.

[Figure]

Figure B1. Daily time series of measured and modeled minor species over Syowa Station at 18 km. Black diamonds are data by FTIR, red squares are by Aura/MLS and Envisat/MIPAS, blue triangles are data by MIROC3.2 CCM. Figure B1(a) is for ClO, B1(b) is for HCl, B1(c) is for $ClONO_2$, B1(d) is for Cly, and B1(e) is for $O_3$ in 2007. Figure B1(f) is for ClO, B1(g) is for HCl, B1(h) is for $ClONO_2$, B1(i) is for Cly, and B1(j) is for $O_3$ in 2011.

[Figure]

Figure B2. Same as Figure B1 but for 22 km.

---

## Author Comment (AC2)

**Reply to reviewer #2**

We thank anonymous referee #2 for his/her constructive review that helped to improve the contents of our paper. The review comments by anonymous referee #2 are numbered and repeated below as *in italic letters*, followed by our answers.

*(1). Although there are likely no severe issues with the data or the simulations, the scientific goal of the manuscript remains completely unclear. I find that it is not sufficient, just to present the observations and simulations without addressing open scientific questions. The authors present no new novel concepts, ideas, or tools. The only thing that is said that they present the first ground based continuous measurements of chlorine species in Antarctica.*

We believe that this study contains several new findings in the characteristics of chlorine species over the whole period of Antarctic ozone hole. One is that our measurement was the first continuous measurement of chlorine species related to the ozone hole from the ground in Antarctica. This description was now added in the Abstract and in the Conclusion. Another new finding is that the deactivation pathways from active ClO into reservoir species (HCl and/or ClONO$_2$) in the Antarctica depends on the availability of ambient O$_3$, and they are different in different altitudes (18 and 22 km in our case), which has never observed in the Antarctic station before. Other new finding is that day-to-day variations in ClO and ClONO$_2$ over Syowa Station are negatively correlated (see (new) Figure 10), associated with the distance between Syowa Station and the inner edge of the polar vortex. These new findings are summarized in the Abstract and Conclusions.

*(2). The model data are displayed in addition to the observations but not on the same altitude levels and they are not used to interpret the observations. There is no obvious connection between the simulations and the observations besides two appendix figures that contain the time series of the chlorine compounds and ozone over Syowa station from model and observations.*

Yes, you are right. The FTIR and satellite observations are discussed in altitude grid, while the MIROC3.2 CCM output was displayed in a single pressure grid (50 hPa), which was close to the observation altitude of 18 km. The purpose of using the CCM output in this study is to understand the behavior of chlorine species in the whole polar vortex scale (which is now typically shown in (new) Figure 11). By looking at variations of chlorine species in a single station (e.g., (new) Figures 4-9), it is difficult to understand the reason of negative correlation between ClO and ClONO$_2$. The purpose of this study is definitely not the comparison or validation of the MIROC3.2 CCM with observations. Therefore, the comparison between CCM and observations are placed in the Appendix.

*(3). With respect to the interpretation of the data, it seems that only few aspects of the observations and the simulations are mentioned. Most of them are in line of what is expected and shown in standard chemistry model*

*runs over the last two decades. The simulations seem to be in line with the observations. But it remains completely unclear, what the message of the paper is.*

As we said above, the main focus of this study is to show whole winter behavior of chlorine species over Syowa Station covering whole ozone hole period. The time series over Syowa Station shows temporal variation of chlorine species over a single station, while the CCM results shows vortex wide spatial distribution of chlorine species. The CCM results are also used to help understanding the characteristics of chlorine species and their (negative) correlations. Moreover, MIROC3.2 CCM succeeded to reproduce the continuous HCl loss in the core of polar vortex in winter period, which was briefly mentioned in the manuscript.

*(4). Furthermore, there are many small inaccuracies in the text, some of them are summarized below. Also, many informations are not given precisely, such that the reviewer needs to guess, what the authors meant. Because of the missing scientific concept, I would not recommend the paper for publication in Atmos. Chem. Phys.*

Thank you for pointing out inaccuracies in the manuscript. We tried to correct them as much as possible which are described below. The scientific concept of this study is the first observational study of temporal evolution of chlorine species throughout the Antarctic winter, springtime ozone hole period, and recovery phase at Syowa Station, which is located near the edge of the polar vortex.

***General***

*(5). The introductions lists some textbook knowledge but it is not clear, why it is at all important in the context of the manuscript.*

Another reviewer also pointed out the shortage of description in the Introduction part. We revised the Introduction part to show more clearly what we want to discuss in this paper.

*(6). The use of ClO data as done in this paper is problematic, since one needs to take into account the diurnal cycle typically involving $Cl_2O_2$ as a nighttime reservoir. Observations (fig 6-12) are likely for different local times and are therefore not comparable. As in the 3-hourly model data of fig 17, no diurnal cycle is visible, it is likely a zonal average over daytime and nighttime data. That is not useful for comparison. Potentially this is the cause of the ClO difference in figs B1 and B2.*

We used ClO data from only daytime measurements of Aura/MLS instrument. Since Aura/MLS is in a sun-synchronized orbit, it always observes at 13:45 local time. In the 3-hourly model data in (new) Figure 12, some diurnal cycles are visible in $Cl_2O_2$ (Fig. 12(b)) and ClO (Fig. 12(d)) data at 68.4S (blue dots) and 71.2S (red dots). The large differences in ClO between CCM and MLS in Figures B1 and B2 might be due to the daily average of the ClO data in the CCM, as you suggested, which is explained in the Appendix B.

*(7). Figures 13-16 show the model output on the 50 hPa level for 4 different times. It is not so clear, what can be learned from these figures. Also it would be better to not use a different vertical coordinate (pressure) for the model results as opposed to the observations (altitude).*

As is mentioned above, the purpose of showing this figure (new Figure 11) is to show that the time series over Syowa Station includes both chemical evolution and dynamical effect of the spatial distribution due to the movement of the vortex edge, because Syowa Station is located at relatively lower latitude (69S).   See the difference in ClO and $ClONO_2$ values at the location of Syowa Station (shown by stars in Fig. 11) between Sep. 1 and Sep. 6.   These situations are difficult to understand by only looking at data over single station like in Figs. 4-9.   In order to show general characteristics of the situation relative to the location of polar vortex, difference in altitude grid would not affect a lot.

*(8). fig 10/11 shows $Cl_y$ (MLS) relative to $Cl_y$\* (MLS $N_2O$), even though that should be described more clearly. In the context $Cl_y$ (without star) is defined as $ClO + ClONO_2 + HCl$. However, there are no $ClONO_2$ observations of MLS. This needs to be clarified.*

$ClONO_2$ was measured by MIPAS.   In order to show $Cl_y$ (FTIR) and $Cl_y$ (MLS) more clearly, we added new equations (3) and (4) in the text.

**Details**

*(9). page 1/line 16 "is not well understood." I don't think that this statement is justified.*

I think this statement is valid.   For example, there are several issues related to Antarctic chlorine chemistry which are not fully well understood yet.   For example, recent studies by Grooß et al. (2018, ACP), Müller et al. (2018, ACP), and Zafar et al. (2018, Tellus) raised some new questions which need to be understood in future.

*(10). 1/22 PSC saturation temperature: you likely mean "PSC existence temperature."*

Yes.   It was corrected as suggested.

*(11). 2/3: "from active chlorine" or "from ClO"?*

It was corrected to 'from active chlorine into …'

*(12). 2/17: the expression "inert chlorines" for HCl and $ClONO_2$ is not typical, please use the wording "chlorine reservoirs" (as in 3/31).*

They are corrected to 'chlorine reservoirs'.

*(13). 3/4-14: The chlorine deactivation into $ClONO_2$ or HCl is mentioned, but not that it depends on ozone (Douglass et al. 1997, Grooß et al., 2005 JAS, Grooß et al., 2011 etc;).*

Now, the chlorine deactivation processes into $ClONO_2$ or HCl depending on ambient ozone amount is described in the Introduction using Douglass et al. (1995) and Grooß et al. (2011) as references.

*(14). 3/19: the phrase "super-recovery" is not ideal. It is sometimes used for ozone but not often for $ClONO_2$.*

We rephrased 'super-recovery' into 'additional increase of $ClONO_2$ than initial value' throughout the manuscript.

*(15). 3/22: ozone has been monitored before the discovery of the ozone hole. (otherwise the ozone hole would not have been discovered).*

You are right. However, it is also true that measurement quality and quantity on ozone and related atmospheric trace gas species have increased a lot after the discovery of the ozone hole. In order to avoid your point, the word 'intensively' was added in the manuscript.

*(16). 4/18: "analysis" Do you mean retrieval of tracer profiles from the FTIR spectra?*

Yes. It was corrected to 'retrieval'.

*(17). 4/22: how many layers exactly?*

We checked the SFIT2 program and found that it was actually constructed with 47 layers. It was corrected in the text.

*(18). 5/12: As you only show 14 coincident measurements within a period of 3 months, I would not call this chapter "Validation."*

Initially we compared FTIR profiles with MLS $O_3$ measurements, which gave much more comparison cases (~50 like the case for $HNO_3$ or HCl). However, there was a suggestion to compare with ozonesondes rather than MLS, because the accuracy of ozonesonde profiles are much more reliable than satellite measurements. Since ozonesonde measurements were done only once per week, there were only 14 coincidences in total. Nevertheless, we believe that comparison with ozonesondes is better than with satellite measurements.

*(19). 5/31: You show results from MLS Version 3.3 data. Why do you not use version 4.x?*

At the time when we have done this study, MLS Version 3.3 was the latest data product. In the technical documentation of MLS data products (Livesey et al., 2018), it was stated that there are some improvements for MLS Ver. 4.2 compared with Ver. 3.* product, such as improved composition profiles in cloudy regions, or more suitable reference surface for geopotential heights. However, there are only minor changes in the data product for the focus of this study ($O_3$, $ClO$, $HCl$, and $HNO_3$).

*(20). 6/8: How exactly do you identify the coincident CALIPSO PSCs? Orbit within a certain distance from Syowa station? PSCs of what type at what altitude?*

The nearest CALIPSO orbit to Syowa Station was selected. The maximum distance between Syowa Station and the nearest CALIPSO orbit is about 500 km, and the maximum time difference is +/- 12 hours. Any type of PSCs at any altitude were counted. The description was added.

*(21). 6/12-15: This seems to be a speculation. It is not clear how this statement is proven in this context.*

You are right. The description was rephrased as 'This may be due to other reason, …'.

*(22). 6/16: You use the term 'temporal variations' several times, where I think it is (only) a time series.*

We changed 'temporal variation' into 'time series' at several places.

*(23). 6/16: define the expression $Cl_y*$.*

$Cl_y*$ is defined a few lines below (inferred inorganic chlorine) and in equation (2).

*(24). 6/17: add 'for all ground-based and satellite based observations used in this study' (or similar).*

The description was added in the manuscript as suggested.

*(25). 8/5 ratios of each species with respect to $Cl_y*$.*

It was now rephrased as suggested.

*(26). 9/1ff: It has not been said over what time the data were collected, how the anticorrelation was evaluated (MIPAS and MLS have different orbits).*

MLS and MIPAS data were sampled on the same day at the nearest orbit for both satellites to Syowa Station.

The maximum differences between these two satellites' observational times and locations are 9.0 hours in time and 587 km in distance. Mean differences are 6.8 hours in time and 270 km in distance, respectively. This information is now described in the manuscript.

*(27). page 9/line 1-6 (anti-correlation of MIPAS ClONO₂ and MLS ClO, fig 12): If is said that this is due to the PV (eq. latitude) dependence. Could it also be that this occurs because of the time dependence of the deactivation throughout the days 220-260? The slope of the regression line is not given in the text nor a statement of what would be expected from the model. What does this slope or correlation mean scientifically? This correlation in the phase of chlorine deactivation is definitely no surprise.*

The slope of the regression line is now shown on the (new) Figure 10. It is about -1.0 for all panels, which is expected if we assume constant $Cl_y$ value in the polar vortex. As we described in the manuscript, this negative correlation between $ClONO_2$ and $ClO$ is the result of relative distance between Syowa Station and polar vortex edge, not because of the time dependence of the chlorine deactivation.

*(28). 9/24 'About 50% ozone depletion was seen throughout the polar vortex.' This statement is inaccurate. Do you mean at the 50 hPa level equally from the vortex edge to the core? Or also at the other levels? It is at least rather hard to read a number of percentage ozone loss from this figure.*

We agree that this statement is inaccurate. We now rephrased to: 'Development of ozone depletion was seen in the polar vortex.'

*(29). 9/33f 'Inside the polar vortex, depletion of O₂, NO₂, HNO₃, and ClONO₂ continued' Most of the species are already near zero. It is not clear from the figure how you see continuing depletion. Also I would not expect further ozone depletion, if active chlorine returned more or less to zero.*

We agree. We now rephrased to: 'Inside the polar vortex, $O_3$, $NO_2$, $HNO_3$, and $ClONO_2$ showed very low values.'

*(30). 10/23-28: The continuous loss of HCl seems to look differently than in the study by Grooß et al. that you mention. The conclusions of that study are not given properly. It is not clear, whether you include an additional process like ionisation by cosmic rays or cross vortex edge ClONO₂ flow due to Solomon et al. in your model. Or does the HCl just deplete because of the large diffusivity that is present due to the low model resolution (2.8×2.8 degrees)?*

At the time when we submitted this paper, the paper of Grooß et al., (2018) was in the discussion phase of ACPD. Now, Grooß et al., (2018) was published in ACP, and the conclusion was a bit modified. We modified the draft to more accurately refer the conclusion of new Grooß et al., (2018) paper in ACP. In our MIROC3.2 CCM run, we did not include any ionization process by cosmic rays. We agree with you on that

*'the HCl just deplete because of the large diffusivity that is present due to the low model resolution (2.8×2.8 degrees)'.* In addition, our model uses a hybrid sigma-pressure levels for the vertical coordinates. This also may lead to a larger diffusivity than in the theta coordinates.

*(31). 11/18f: I do not see this good agreement between model and FTIR in the figures.*

You are right. We now rephrased to: 'The modeled $O_3$ and day-to-day variations of HCl and $ClONO_2$ are in good agreement with FTIR and satellite observations.

*(32). figure 5: How was $T_{NAT}$ and $T_{ice}$ derived? What data for $HNO_3$ and $H_2O$ were used?*

We calculated $T_{NAT}$ and $T_{ice}$ by assuming 6 ppbv $HNO_3$ and 4.5 ppmv $H_2O$. The explanation was added in the manuscript.

*(33). figure 12: There must be something wrong with the colour coding of the PV in the panels. I would expect about a factor 2 difference in PV between 18 and 22 km and also PV values significantly below -85PVU at 22 km.*

We checked the actual PV values used in this plot. Actually, the PV values for this figure ranged as follows: (new) Figure 10 (a): -61.4 ~ -76.3, (b): -61.1 ~ -78.6, (c): -60.4 ~ -80.5, (d): -61.2 ~ -82.7. Therefore, we used the same color scales for both 18 and 22 km.

*(34). figure 17: Three-hourly zonal-mean temporal variations'What do you mean by variations? It only looks like zonal mean values.*

The description was now rephrased as: 'Three-hourly zonal-mean time series of …'

*(35). figures B1/B2: Here the model is labelled CTM. Is it really? In the paper you always talk about a CCM.*

CCM is correct. The label was corrected in the figures.

***Typos/grammar***
*(36). 6/12: other reason -> ̈other reasons'or ̈an other reason"*

Corrected to: 'other reasons'.

*(37). 8/32 shows*

Corrected as suggested.

---

## Author Comment (AC4)

**Temporal evolution of chlorine and minor species related to ozone depletion at Syowa Station, Antarctica during austral fall to spring in 2007 and 2011**

Hideaki Nakajima1,2, Isao Murata2, Yoshihiro Nagahama1, Hideharu Akiyoshi1, Kosuke Saeki2,3, Masanori Takeda2, Yoshihiro Tomikawa4,5, and Nicholas B. Jones6

1National Institute for Environmental Studies, Tsukuba, Ibaraki, 305-8506, Japan
 2Graduate School of Environmental Studies, Tohoku University, Sendai, Miyagi, 980-8572, Japan
 3now at Weathernews Inc., Chiba, 261-0023, Japan
 4National Institute of Polar Research, Tachikawa, Tokyo, 190-8518, Japan
 5The Graduate University for Advanced Studies, Tachikawa, Tokyo, 190-8518, Japan
 6University of Wollongong, Wollongong, New South Wales, 2522, Australia

Correspondence to: Hideaki Nakajima (nakajima@nies.go.jp)

**Abstract.**

5

10

To understand and project future ozone recovery, understanding of the mechanisms related to polar ozone destruction is

- 15 crucial. For polar stratospheric ozone destruction, chlorine species play an important role, but detailed temporal evolution of chlorine species in the Antarctic winter is not well understood. We retrieved lower stratospheric vertical profiles of O3, HNO3, and HCl from solar spectra taken with a ground-based Fourier-Transform infrared spectrometer (FTIR) installed at Syowa Station, Antarctica (69.0°S, 39.6°E) from March to December 2007 and September to November 2011. This was the first continuous measurements of chlorine species related to the ozone hole from the ground in Antarctica. We analyzed temporal
- 20 variation of these species combined with CIO, HCl, and HNO3 data taken with the Aura/MLS (Microwave Limb Sounder) satellite sensor, and CIONO2 data taken with the Envisat/MIPAS (The Michelson Interferometer for Passive Atmospheric Sounding) satellite sensor at 18 and 22 km over Syowa Station. When the stratospheric temperature over Syowa Station fell below polar stratospheric cloud (PSC) existence temperature in early winter, PSCs started to form and heterogeneous reaction on PSCs convert chlorine reservoirs into reactive chemical species. HCl and CIONO2 decrease occurred at both 18 and 22 km,
- and soon ClONO2 was almost depleted in early winter. When the sun returned to Antarctica in spring, enhancement of ClO and gradual O3 destruction were observed. During the ClO enhanced period, negative correlation between ClO and ClONO2 was observed in the time-series of the data at Syowa Station. This negative correlation was associated with the distance between Syowa Station and the inner edge of the polar vortex. We used MIROC3.2 Chemistry-Climate Model (CCM) results to see the comprehensive behavior of chlorine and related species inside the polar vortex and the edge region in more detail.
- 30 Rapid conversion of chlorine reservoir species (HCl and ClONO2) into Cl2, gradual conversion of Cl2 into Cl2O2, increase of ClO when sunlight became available, and conversion of ClO into HCl, was successfully reproduced by the CCM. HCl decrease in the winter polar vortex core continued to occur due to the transport of ClONO2 from the subpolar region to higher latitudes,

providing a flux of  $CIONO_2$  from more sunlit latitudes into the polar vortex. Temporal variation of chlorine species over Syowa Station was affected by both heterogeneous chemistry related to PSC occurrence deep inside the polar vortex, and transport of an  $NO_x$ -rich airmass from lower latitudinal polar vortex boundary region which can produce additional  $CIONO_2$ by reaction of CIO with  $NO_2$ . The deactivation pathways from active chlorine into reservoir species (HCl and/or  $CIONO_2$ )

5 were found to be highly dependent on the availability of ambient O3. At an altitude (18 km) where most ozone was depleted in Antarctica, most ClO was converted to HCl. However, at an altitude (22 km) when there were some O3 available, additional increase of ClONO2 from initial value can occur, similar to the case in the Arctic.

**1. Introduction**

- 10 Discussion of the detection of "recovery" of the Antarctic ozone hole as the result of CFC restrictions has been attracting attention. The occurrence of the Antarctic ozone hole is considered to continue at least until the middle of this century. The world's leading Chemistry-Climate Models (CCMs) mean time series indicate that the springtime Antarctic total column ozone will return to 1980 levels between 2045 and 2060 (WMO, 2014). In fact, the recovery time predicted by CCMs has large uncertainty, and observed ozone hole magnitude also shows year-to-year variability. Although Solomon et al. (2016) and de
- 15 Laat et al. (2017) reported signs of healing in the Antarctic ozone layer only in September month, there is no statistically conclusive report on the Antarctic ozone hole recovery (Yang et al., 2008; Kuttippurath et al, 2010).

To understand ozone depletion processes in polar regions, understanding of the behavior and partitioning of active chlorines  $(CIO_x=CI+CI_2+CIO+CIOO+CI_2O_2+HOCI+CINO_2)$  and chlorine reservoirs (HCl and CIONO\_2) are crucial. Chlorine reservoir is converted to active chlorine that destroys ozone on polar stratospheric clouds (PSCs) through heterogeneous reactions:

 $CIONO_{2}(g) + HCl(s, l) \rightarrow Cl_{2}(g) + HNO_{3}$ (R1)

$$CIONO_2(g) + H_2O(l, s) \rightarrow HOCl(g) + HNO_3$$
(R2)

where g, s, and l represents the gas, solid, and liquid phases, respectively (Solomon et al., 1986; Nakajima et al., 2016).

Heterogeneous reactions on the surface of particles;

$$N_{2}O_{5}(g) + HCl(s, l) \rightarrow ClNO_{2}(g) + HNO_{3}$$
(R3)

30

$$HOCl (g) + HCl (s, l) \rightarrow Cl_2 (g) + H_2O$$
(R4)

are responsible for additional chlorine activation. When solar illumination is available,  $Cl_2$ , HOCl, and ClNO2 are photolyzed to produce chlorine atoms by reactions:

| $Cl_2 + hv \rightarrow Cl + Cl$      | (R5) |
|--------------------------------------|------|
| $HOCl + hv \rightarrow Cl + OH$      | (R6) |
| $ClNO_2 + hv \rightarrow Cl + NO_2.$ | (R7) |

The yielded chlorine atoms then start to destroy ozone catalytically through reactions:

$$Cl + O_3 \rightarrow ClO + O_2$$
 (R8)

| $ClO + ClO + M \rightarrow Cl_2O_2 + M$ | (R9)  |
|-----------------------------------------|-------|
| $Cl_2O_2 + hv \rightarrow Cl + ClOO$    | (R10) |
| $ClOO + M \rightarrow Cl + O_2 + M.$    | (R11) |

When the stratospheric temperatures get warmer than nitric acid trihydrate (NAT) PSC saturation temperature, and no PSCs
are present, gradual deactivation of chlorine starts to occur. Re-formation of ClONO2 and HCl mainly occurs through reactions:

$$CIO + NO_2 + M \rightarrow CIONO_2 + M$$
(R12)

$$Cl + CH_4 \rightarrow HCl + CH_3.$$
 (R13)

The re-formation of ClONO2 by reaction (R12) from active chlorine is much faster than that of HCl by reaction (R13), if there are enough NOx around (Mellqvist et al., 2002; Dufour et al., 2006). But the formation rates of ClONO2 and HCl are also related to ozone concentration. Douglass et al. (1995) showed that HCl increases more rapidly in the Antarctic polar vortex than in the Arctic polar vortex due to lower ozone concentrations in the Antarctic polar vortex. Low ozone reduces the rate of reaction (R8), and then Cl/ClO ratio becomes high. Low ozone also reduces the rate of the following reaction:

$$NO + O_3 \rightarrow NO_2 + O_2. \tag{R14}$$

15 This makes NO/NO2 ratio high and increases Cl/ClO ratio by the following reaction:

$$ClO + NO \rightarrow Cl + NO_2.$$
 (R15)

High Cl/ClO ratio leads rapid HCl formation by reaction (R13) and reduces the formation ratio of ClONO2 by reaction (R12). The processes of deactivation of active chlorine are different between typical conditions in the Antarctic and those in the Arctic. In the Antarctic, the temperature cools below the PSC formation threshold in the whole area of the polar vortex in

20 most years, and almost complete denitrification and chlorine activation occur (WMO, 2007), followed by severe ozone depletion in spring. In the chlorine reservoir recovery phase, HCl is mainly formed by reaction (R13) due to the lack of ozone (typically less than 0.5 ppmv) by the mechanism described in the previous paragraph (Grooß et al., 2011).

On the other hand, in the Arctic, typically less PSC formation occurs in the polar vortex due to generally higher stratospheric temperatures (~10-15K in average) compared with that of Antarctica. Then only partial denitrification and chlorine activation

- 25 occur (Manney et al., 2011; WMO, 2014). In this case, some ozone, CIO, and NO2 are available in the chlorine reservoir recovery phase. Therefore, the CIONO2 amount becomes sometimes higher than that of HCl after PSCs have disappeared due to the rapid reaction (R12) (Michelsen et al., 1999; Santee et al., 2003), which results in additional increase of CIONO2 than initial value at the time of chlorine deactivation in spring (Webster et al., 1993). In this way, the partitioning of chlorine reservoir in springtime is related to temperature, PSC amounts, ozone, and NO2 concentrations (Santee et al., 2008; Solomon
- 30 et al., 2015).

In the polar regions, the ozone and related atmospheric trace gas species have been intensively monitored by several techniques since the discovery of the ozone hole. These measurements consist of direct observations by high-altitude aircrafts (e.g., Anderson et al., 1989; Ko et al., 1989; Bonne et al., 2000), remote sensing observations by satellites (e.g., Müller et al., 1996; Michelsen et al., 1999; Höpfner et al., 2004; Dufour et al., 2006; Hayashida et al., 2007), remote sensing observations

from the ground (e.g., Farmer et al., 1987; Kreher et al., 1996; Mellqvist et al., 2002; Blumenstock et al., 2006). Within these observations, ground-based measurements have the characteristic of high temporal resolution. In addition, the Fourier-Transform infrared spectrometer (FTIR) has the capability of measuring several trace gas species at the same time or in a short time interval. In this paper, we show the results of ground-based FTIR observations of O3 and other trace gas species at Syowa

5 Station in the Antarctic in 2007 and 2011, combined with the satellite measurements of trace gas species by Aura/MLS and Envisat/MIPAS, to show the temporal variation and partitioning of active chlorine (ClOx) and chlorine reservoirs (HCl, ClONO2) from fall to spring during the ozone hole formation and dissipation period. The methods of FTIR and satellite measurements are described in Section 2. The validation of FTIR measurements is described in Section 3. The results of FTIR and satellite measurements are described in Section 4. Finally, in Section 5, distributions of minor species simulated

10 with the MIROC3.2 chemistry-climate model are used to further discuss the behavior of active and inert chlorine species.

**2. Measurements**

**2.1 FTIR measurements**

The Japanese Antarctic Syowa Station (69.0°S, 39.6°E) was established in January 1957. Since then, several scientific observations related to meteorology, upper atmospheric physics, glaciology, biology, geology, seismology, etc. have been performed. The ozone hole was first detected by Dobson spectrometer and ozonesonde measurements from Syowa Station in 1982 (Chubachi, 1984) and Halley Bay (Farman et al., 1985). We installed a Bruker IFS-120M high-resolution Fourier-Transform infrared spectrometer (FTIR) in the Observation Hut at Syowa Station in March 2007. This was the second highresolution FTIR site in Antarctica in operation after New Zealand's Arrival Heights facility at Scott Station (77.8°S, 166.7°E)

- 20 (Wood et al., 2002; Wood et al., 2004). The IFS-120M FTIR has a wavenumber resolution of 0.0035 cm-1, with two liquid nitrogen cooled detectors (InSb and HgCdTe covering the frequency ranges 2000-5000 and 700-1300 cm-1, respectively), and fed by an external solar tracking system. Since Syowa Station is located at a relatively low latitude (69.0°S) compared with Scott Station (77.8°S), there is an advantage of the short (about one month) polar night period, when we cannot measure atmospheric species using the sun as a light source. Since we can resume FTIR measurements from early spring (late Julv).
- 25 we can measure chemical species during ozone hole development. Another advantage of Syowa Station is that we can measure both inside and outside of the polar vortex, since the station is sometimes located near the edge of the polar vortex. From March to December 2007, we made 78 days of FTIR measurements in total. Another 19 days of FTIR measurements were performed from September to November 2011. Table 1 shows days when FTIR measurements were made at Syowa Station with the information inside/boundary/outside of the polar vortex defined by the method described in Section 4 using ERA-
- 30 Interim reanalysis data.

The retrieval of the FTIR spectra was done with SFIT2 Version 3.92 program (Rinsland et al., 1998; Hase et al., 2004). SFIT2 retrieves a vertical profile of trace gases using an optimal estimation formulation of Rodgers (2000), implemented with

a semi-empirical method which was originally developed for microwave measurements (Parrish et al., 1992; Connor et al., 1995). The SFIT2 forward model fully describes the FTIR instrument response, with absorption coefficients calculated using the algorithm of Norton and Rinsland (1991). The atmosphere is constructed with 47 layers from the ground to 100 km, using the FSCATM (Gallery et al., 1983) program for atmospheric ray-tracing to account for refractive bending. The retrieval

5 parameters for each gas are shown in Table 2. Temperature and pressure profiles between 0 and 30 km are taken by the Rawin sonde observations flown from Syowa Station on the same day by the Japanese Meteorological Agency (JMA), while values between 30 and 100 km are taken from the COSPAR International Reference Atmosphere 1986 (CIRA-86) standard atmosphere profile (Rees et al., 1990).

We retrieved vertical profiles of O3, HCl, and HNO3 from the solar spectra. We used monthly averaged ozonesondes profiles
(0-30 km) and Improve Limb Atmospheric Spectrometer-II (ILAS-II) (Nakajima, 2006; Nakajima et al., 2006; Sugita et al., 2006) profiles (30-100 km) for the a priori of O3, monthly averaged profiles from ILAS-II for HNO3 and monthly averaged profiles from HALOE (Anderson et al., 2000) for HCl. Typical averaging kernels of the SFIT2 retrievals for O3, HNO3, and HCl are shown in Figures 1(a), (b), and (c), respectively.

**15 **2.2 Satellite measurements**

20

The Earth Observing System (EOS) Microwave Limb Sounder (MLS) onboard the Aura satellite was launched on 15 July 2004, to monitor several atmospheric chemical species in upper troposphere to mesosphere (Waters et al., 2006). The Aura orbit is sun-synchronous at 705 km altitude with an inclination of 98°, 13:45 ascending (north-going) equator-crossing time, and 98.8-min period. Vertical profiles are measured every ~165 km along the suborbital track, horizontal resolution is ~200-600 km along-track, ~3-10 km across-track, and vertical resolution is ~3-4 km in the lower to middle stratosphere (Froidevaux

- et al., 2006). ClO, HCl, and HNO3 profiles used in this study were taken from Aura/MLS version 3.3 data (Liversey et al., 2006; Santee et al., 2011; Ziemke et al., 2011; Liversey et al., 2013). The MLS data was selected whose measurement location is within 300 km radius from Syowa Station and within ±6 hours of the FTIR measurement.
- Michelson Interferometer for Passive Atmospheric Sounding (MIPAS) is a Fourier transform spectrometer sounding the thermal emission of the earth's atmosphere between 685 and 2410 cm-1 (14.6-4.15 μm) in limb geometry (Fischer and Oelhaf, 1996). The maximum optical path difference of MIPAS is 20 cm. The field-of-view of the instrument at the tangent points is about 3 km in the vertical and 30 km in the horizontal. In the standard observation mode in one limb-scan, 17 tangent points are observed with nominal altitudes 6, 9, 12, ..., 39, 42, 47, 52, 60, and 68 km. In this mode, about 73 limb scans are recorded per orbit. The measurements of each orbit cover nearly the complete latitude range from about 87°S to 89°N. MIPAS was
- 30 put on board the European Environmental Satellite (Envisat), which was launched on 1 March 2002, and was put into a polar sun-synchronous orbit at an altitude of about 800 km with an inclination of 98.55° (von Clarmann et al., 2003). On its descending node, the satellite crosses the equator at 10:00 local time. Envisat performs 14.3 orbits per day, which results in a good global coverage. ClONO2 profiles which we used in this study were taken from Envisat/MIPAS IMK/IAA version

V5R CLONO2 220 and V5R CLONO2 222 (Höpfner et al., 2007). The measurement criteria of the MIPAS data used in this study are the same as that of Aura/MLS.

The Cloud-Aerosol-Lidar and Infrared Pathfinder Satellite Observations (CALIPSO) satellite was launched on 28 April 2006. On CALIPSO satellite, Cloud-Aerosol Lidar with Orthogonal Polarization (CALIOP) instrument was on board, to

5 monitor aerosols, clouds, and polar stratospheric clouds (PSCs) (Pitts et al., 2007). CALIOP is a two-wavelength, polarization

15

sensitive lidar that provides high vertical resolution profiles of backscatter coefficient at 532 and 1064 nm, as well as two orthogonal (parallel and perpendicular) polarization components at 532 nm (Winker et al., 2007). In order to monitor the appearance of PSCs over Syowa Station, we used CALIOP PSC data (Pitts et al., 2007; 2009; 2011).

**3. Validation of retrieved profiles from FTIR spectra with other measurements 10**

We validated retrieved FTIR profiles of O3 with ozonesondes, and HNO3 and HCl with Aura/MLS version 3.3 data (Liversey et al., 2013) for 2007 measurements. We identified the nearest Aura/MLS data from the distance between the Aura/MLS tangent point at 20 km altitude and the point at 20 km altitude for the direction of the sun from Syowa Station at the time of the FTIR measurement. The spatial and temporal collocation criteria used was within 300 km radius and  $\pm 6$  hours. The ozonesonde and Aura/MLS profiles were interpolated onto a 1 km-grid, then smoothed with a 5 km-wide slit function.

Figures 2(a)-(b) show absolute and percentage differences of  $O_3$  profiles retrieved from FTIR measurements and those from ozonesonde measurements, respectively, calculated from 14 coincident measurements from September 5 to December 17, 2007. We define the percentage difference D as:

D(%) = 100 \* (FTIR-sonde) / ((FTIR+sonde)/2).(1)

20 The absolute agreement between 15 and 25 km was within -0.02 to 0.40 ppmv. The relative difference D between 15 and 25 km was within -10.4 to +24.4%. The mean D of  $O_3$  for the altitude of interest in this study (18-22 km) was +6.1%, with the minimum of -10.4% and the maximum of +19.2%. FTIR data agree with validation data within error bars at the altitude of interest. Note that relatively large D values between 16 and 18 km are due to small ozone amount in the ozone hole.

Figures 2(c)-(d) show absolute and percentage differences of HNO3 profiles retrieved by FTIR measurements and those from

- 25 Aura/MLS measurements, respectively, calculated from 44 coincident measurements. The agreement between 15 and 25 km was within -0.56 to +0.57 ppby. The D between 15 and 25 km was within -25.5 to +21.9%. The mean D for HNO3 for the altitude of interest in this study (18-22 km) was +13.2%, with the minimum of +0.2% and the maximum of +21.9%. However, this positive bias of FTIR data is still within the error bars of FTIR measurements. Livesey et al. (2013) showed that Aura/MLS version 3.3 data has no bias within errors (~0.6-0.7 ppbv at pressure level of 100-3.2 hPa) compared with other measurements.
- 30 Figures 2(e)-(f) show absolute and percentage differences of HCl profiles retrieved by FTIR measurements and those from Aura/MLS measurements, respectively, calculated from 47 coincident measurements. The agreement between 15 and 25 km was within -0.20 to -0.09 ppby. The D between 15 and 25 km was within -34.1 to -3.0%. The mean D for HCl for the altitude

of interest in this study (18-22 km) is -9.7%, with a minimum of -14.6% and a maximum of -3.0%. However, this negative bias of FTIR data is still within the error bars of FTIR measurements. Moreover, Livesey et al. (2013) showed Aura/MLS version 3.3 values are systematically greater than HALOE values by 10-15% with a precision of 0.2-0.6% in the stratosphere, which may partly explain the negative bias of FTIR data compared with MLS data.

5

Table 3 summarizes validation results of FTIR profiles compared with ozonesonde or Aura/MLS measurements, and possible Aura/MLS biases from literature.

**4. Results**

Figures 3(a)-(d) show the time series of temperatures at 18 and 22 km over Syowa Station using ERA-Interim data (Dee et al.,

- 10 2011) for 2007 and 2011. Approximate saturation temperatures for NAT PSC ( $T_{NAT}$ ) and ice PSC ( $T_{ICE}$ ) calculated by assuming 6 ppbv HNO3 and 4.5 ppmv H2O are also shown in the figures. The dates when PSCs were observed at Syowa Station identified by the nearest CALIOP data of that day were indicated by asterisks on the bottom of the figures. Over Syowa Station, PSCs were observed at 15-25 km from the beginning of July (day 183) to the middle of September (day 253) in 2007, and from late June (day 175) to early September (day 251) in 2011.
- 15 PSCs were observed only below 20 km after mid-August, due to the sedimentation of PSCs and downwelling of vortex air in late winter. Although temperatures above Syowa Station were sometimes below TNAT in June and in late September, no PSC was observed during those periods. This may be due to other reasons, such as a different time history of temperature for PSC formation, and/or low HNO3 (denitrification) and/or H2O concentration (dehydration) which are needed for PSC formation in late winter season (Saitoh et al., 2006).
- Figures 4-7 show time series of HCl, ClONO2, ClO, Cly\*, O3, and HNO3 over Syowa Station in 2007 and 2011 at altitudes of 18 and 22 km for all ground-based and satellite based observations used in this study, respectively. O3 (sonde) is observed with the KC96 ozonesonde for 2007 and the ECC-1Z ozonesonde for 2011 by JMA. HCl and HNO3 observed by Aura/MLS are plotted to complement the data lack of FTIR measurements. ClONO2 is observed by Envisat/MIPAS. Total inorganic chlorine Cly\* corresponds to the sum of HCl, ClONO2, and Clx, where active chlorine species Clx is defined as the sum of ClO,
- 25 Cl, and 2\*Cl2O2 (Bonne et al., 2000). Inferred total inorganic chlorine Cly\* is calculated from N2O value (in ppbv) measured by MLS using the polynomial equation derived from the correlation of Cly and N2O (Bonne et al., 2000):

 $Cl_{y}^{*}(pptv) = 4.7070^{*}10^{-7}(N_{2}O)^{4} - 3.2708^{*}10^{-4}(N_{2}O)^{3} + 4.0818^{*}10^{-2}(N_{2}O)^{2} - 4.6856(N_{2}O) + 3225.$  (2)

Dark shaded and thin shaded days indicate that Syowa station was located outside and in the boundary region of the polar vortex, respectively. Inner and outer edges of the polar vortex were determined as follows:

30 1) Equivalent latitudes (McIntyre and Palmer, 1984; Butchart and Remsberg, 1986) were computed based on isentropic potential vorticity at 450 K and 560 K isentropic surfaces for 18 km and 22 km using the ERA-Interim reanalysis data (Dee et al., 2011), respectively.

2) Inner and outer edges (at least 5° apart from each other) of the polar vortex were defined by local maxima of the isentropic potential vorticity gradient with respect to equivalent latitude only when a tangential wind speed (i.e., mean horizontal wind speed along the isentropic potential vorticity contour; see Eq. (1) of Tomikawa and Sato (2003)) near the vortex edge exceeds a threshold value (i.e., 20 m s-1, see Nash et al. (1996) and Tomikawa et al. (2015)).

5

3) Then, the polar region is categorized into three categories; i.e., inside the polar vortex (inside of inner edge), the boundary region (between inner and outer edges), and outside the polar vortex (outside of outer edge).

Hereafter, we will discuss the results only when Syowa station was located inside the polar vortex. Note that the Syowa station is often located near the vortex edge and the temporal variations observed over Syowa station sometimes reflect spatial variations, not the chemical evolution. The lack of data for CIO and HCl (MLS) from day 195 to day 219, 2007 and CIONO2

10 from day 170 to day 216, 2007 (upper panels of Figures 4(a) and 6(a)) is due to large error in Aura/MLS or Envisat/MIPAS measurements during this period.

The altitude of 18 km was selected because it was the altitude where the most ozone depletion was occurred. The altitude of 22 km was selected to show the difference of the behavior of minor atmospheric species with 18 km where about half of the ozone was depleted. The common features found in both 2007 and 2011 at both altitudes of 18 and 22 km can be summarized

- 15 as follows: CIO was enhanced in August and September and the day-to-day variations were large over this period. HCl was almost zero from late June to early September and the day-to-day variations were small over this period (larger values are related to the polar vortex boundary). HCl and ClONO2 decreased first, then ClO started to increase in winter, while HCl increases and ClO decreases were synchronized in spring. Cly\* gradually increased in the polar vortex from late autumn to spring. The Cly\* value became larger compared with its mixing ratio outside of the polar vortex in spring. O3 decreased from
- 20 July to late September when ClO was present. HNO3 showed large decreases from June to July, and then gradually increased in summer. Day-to-day variations of HNO3 from June to August were large.

The following characteristics are evident especially at 18 km (Figures 4 and 5). The day-to-day variations of HCl from late June to early September were as small as 0-0.3 ppbv. The recovered values of HCl in spring were larger than those before winter and those outside the polar vortex during the same period. ClONO2 kept near zero even after ClO disappeared, and did

not recover to the level before winter until spring.  $O_3$  gradually decreased from values of 2.5-3 ppmv before winter to values less than one fifth, 0.3-0.5 ppmv, in October.

The following characteristics are evident only at 22 km (Figures 6 and 7). The day-to-day variation of HCl from late June to early September were 0-1 ppbv, larger than those at 18 km. The recovered values of HCl in spring were nearly the same as those before winter (around 2.2 ppbv). ClONO2 recovered to larger values than those before winter after ClO disappeared.

30 From winter to spring,  $O_3$  gradually decreased, but the magnitude of the decrease was much smaller than that at 18 km. As for the temporal increase of ClONO2 in spring during the ClO decreasing phase, we can see a peak of 1.5 ppbv at 18 km

in 2011, and at 22 km in both 2007 and 2011 around day 270, but we see no temporal increase of ClONO2 at 18 km in 2007. Figure 7 shows that temporal ClO enhancement and decrease of O3, ClONO2, and HNO3 occurred in early winter (day 150-

170) at 22 km in 2011. This small ozone depletion event before winter might be due to an airmass movement from the polar

night area to a sunlit area at lower latitudes. Table 4 summarized the characteristics of variation of minor atmospheric species for 2007 and 2011 at altitudes of 18 and 22 km.

The ratios of observed HCl, ClONO2, ClO, and  $Cl_y$  with respect to  $Cl_y^*$  were calculated to discuss the temporal variations of the chlorine partitioning. Here, observed  $Cl_y$  is determined as:

5  $Cl_y (FTIR) = HCl (FTIR) + ClONO_2 (MIPAS) + ClO (MLS)$  (3)  $Cl_y (MLS) = HCl (MLS) + ClONO_2 (MIPAS) + ClO (MLS).$  (4)

Figures 8 and 9 show the time series of the ratios of each chlorine species with respect to  $Cl_y^*$  in 2007 (a) and in 2011 (b) at 18 km and 22 km, respectively. For both in 2007 and 2011 at 18 km (Figure 8), the ratio of HCl was 0.6-0.8 and the ratio of ClONO2 was 0.2-0.3 before winter (day 130-140). The partitioning of HCl was three times larger than that of ClONO2 at

- 10 that time. The ratio of CIO increased to 0.5-0.6 during the enhanced period (day 240-260). The ratio of HCl was 0-0.2 and the ratio of CIONO2 was 0-0.6 during this same period. CIONO2 shows negative correlation with CIO, while HCl kept low even when CIO was low during this period. This negative correlation is shown in Figure 10 later. When CIO was enhanced, the O3 amount gradually decreased, and finally reached <0.5 ppmv (>80% destruction) in October (day 280) (See Figures 4 and 5). The ratios became 0.9-1.0 for HCl and 0-0.1 for CIONO2 after the recovery in spring (after day 290), indicating that
- 15 almost all chlorine reservoir species became HCl via reaction (R13), due to the lack of O3 and NO2 during this period. The ratios of Cly (FTIR) and Cly (MLS) were both around 0.7 at the time of ClO enhanced period (day 230-260). The remaining chlorine are thought to be Cl2O2, which will be shown in model simulation in Section 5 later. The ratio of Cly became close to 1 after the recovery period (after day 280).

For both in 2007 and 2011 at 22 km (Figure 9), the ratio of HCl was 0.4-0.9 and the ratio of ClONO2 was 0.2-0.3 before winter (day 110-140). The partitioning of HCl was two to three times larger than that of ClONO2. The ratio of ClO increased to 0.6-0.7 during the enhanced period (day 220-240). The ratio of HCl was 0-0.3 and the ratio of ClONO2 was 0-0.6 during this period. ClONO2 shows negative correlation with ClO, while HCl kept low even when ClO was low during this period like the case at 18 km. The O3 amount gradually decreased during the ClO enhanced period but remained >1.5 ppmv (less than half destruction) at this altitude (See Figures 6 and 7). When the ClO enhancement ended, temporal increase of ClONO2

- up to a ratio of 0.5 occurred in early spring (day 260-280). Then, the reservoir ratios became 0.6-0.8 for HCl and 0.2-0.4 for  $CIONO_2$  in spring (after day 280). This phenomenon shows that more chlorine deactivation via reaction (R12) occurred towards  $CIONO_2$  at 22 km rather than at 18km. This is attributed to the existence of  $O_3$  and  $NO_2$  during this period at 22 km, which was different from the case at 18 km. The ratios of  $Cl_y$  (FTIR) and  $Cl_y$  (MLS) were both around 0.8 at the time of ClO enhanced period (day 230-250). The remaining chlorine are thought to be  $Cl_2O_2$ . The ratio of  $Cl_y$  became around 1.1 after the
- 30 recovery period (after day 270). The reason why observed  $Cl_y$  values exceed calculated  $Cl_y^*$  values might be due to the difference in N2O-Cly correlation at this altitude from the one in the equation (2).

In 2011 at 18 km (Figure 8), another temporal increase of  $CIONO_2$  up to a ratio of 0.6 occurred in early spring (around day 280) in accordance with HCl increase, then the  $CIONO_2$  amount gradually decreased to nearly zero after late October (day 300-). This temporal increase in  $CIONO_2$  could be attributed to temporal change of the location of Syowa Station in the polar

vortex. Although Syowa Station was always located inside the polar vortex from day 195 to 350, the difference between the equivalent latitude over Syowa Station and that at inner edge became less than 10 degrees at around day 280, while it was typically between 15 and 20 degrees in other days.  $O_3$  and HNO3 showed higher values around day 280 (see Figure 5), indicating that Syowa Station was located close to the boundary region at this period. Therefore, the temporal increase of ClONO2 in 2011 at 18 km was attributed to spatial variation, not to chemical evolution.

5

**5. Discussion**

Figure 10 shows the correlation between ClO and ClONO2 during the ClO enhanced period (August 8-September 17; day 220-260) at 18 km in 2007 (a) and 2011 (b), and at 22 km in 2007 (c) and 2011 (d). Note that MLS ClO and MIPAS ClONO2

- 10 data were sampled on the same day at the nearest orbit for both satellites to Syowa Station. The maximum differences between these two satellites' observational times and locations are 9.0 hours in time and 587 km in distance. Mean differences are 6.8 hours in time and 270 km in distance, respectively. Solid lines show regression lines obtained by RMA (Reduced Major Axis) regression. Negative correlations of slope ~ -1.0 between ClO and ClONO2 are seen in all figures. The cause of this negative correlation might be due to the variation of the relative distance between Syowa Station and the boundary region of the polar
- 15 vortex. When Syowa Station was located deep inside the polar vortex, there was more CIO and less CIONO2. On the contrary when Syowa Station was located near the vortex edge, there was less CIO and more CIONO2. The potential vorticities (PV) over Syowa Station shown by color code generally show this tendency, that warm colored higher PV points are located more towards bottom right-hand side. This is further confirmed by 3D model simulation as is shown later in this section.
- Figure 11 shows simulated mixing ratios of O3, NO2, HNO3, ClO, HCl, and ClONO2 by the MIROC3.2 Chemistry-Climate
  Model (CCM) at 50 hPa (~18 km) for June 24 (day 175), September 1 (day 244), September 6 (day 249), and October 6 (day 279) in 2007. For a description of the MIROC3.2 CCM, please see Appendix A for detail. The location of Syowa Station is shown by a white star in each panel. Direct comparisons of mixing ratios of ClO, HCl, ClONO2, Cly, and O3 measured by FTIR and MLS, and modeled by MIROC3.2 CCM in 2007 and 2011 at 18 and 22 km are shown in Appendix B. In general, the model results are in good agreement with FTIR and satellite observations (Figure B1). Hereafter, the result of MIROC3.2
- 25 CCM is discussed.

On June 24 (day 175), stratospheric temperatures over Antarctica were already low enough to allow PSCs to form. Consequently,  $NO_2$  and  $HNO_3$  in the polar vortex condensed onto PSCs. Note that the depleted area of  $NO_2$  was greater than that of  $HNO_3$ . This might be due to reaction (R12) that converts ClO and  $NO_2$  to ClONO2 at the edge of the polar vortex, which is shown by the enhanced ClONO2 area at the vortex edge in Figure 11. Also, HCl and ClONO2 are depleted in the

30 polar vortex due to the heterogeneous reactions (R1), (R2), (R3), and (R4) on the surface of PSCs and aerosols. Some HCl remains near the core of the polar vortex, because the initial amount of the counter-part of heterogeneous reaction (R1)

(ClONO2) was less than that of HCl (see Figure 4 and/or 5). The  $O_3$  amount was only slightly depleted within the polar vortex on this day.

On September 1 (day 244), amounts of NO2, HNO3, HCl, and ClONO2 all show very depleted values in the polar vortex. The amount of ClO shows some enhanced values at the outer part of the polar vortex. Development of ozone depletion was seen

- 5 in the polar vortex. Note that ClONO2 shows enhanced values around the boundary region of the polar vortex. This might be due to the reaction (R12) at this location. On this day (day 244), Syowa Station was located inside the polar vortex close to the inner vortex edge, where ClO was smaller and ClONO2 was greater than the values deep inside the polar vortex. On September 6 (day 249), most features were the same as on September 1, but the shape of the polar vortex was different.
- Consequently, Syowa Station was located deep inside the polar vortex, where ClO was greater and ClONO2 was smaller than the values around the boundary region of the polar vortex. Hence, the negative correlation between ClO and ClONO2 seen in Figure 10 was due to variation of the relative distance between Syowa Station and the inner edge of the polar vortex.
- On October 6 (day 279), CIO enhancement has almost disappeared. Inside the polar vortex, O3, NO2, HNO3, and CIONO2 showed very low values. Ozone was almost fully destroyed at this altitude in the polar vortex. However, the amount of HCl increased deep inside the polar vortex. This might be due to the recovery of HCl by reaction (R13) deep inside the polar vortex,
- 15 where there is no  $O_3$  or  $NO_2$  left and reaction (R13) was favoured compared with reaction (R12). At Syowa Station, the amount of HCl was several times greater than that of ClONO2 on this day.

Three-hourly time series of zonal-mean active chlorine species,  $Cl_2O_2$  (b),  $Cl_2$  (c), ClO (d), and their sum  $(ClO+2*Cl_2O_2+2*Cl_2)$  (a), and chlorine reservoir species HCl (e) and  $ClONO_2$  (f) modeled by MIROC3.2 CCM at 68.4°S, 71.2°S, 76.7°S, and 87.9°S are plotted in Figure 12. The dates on which the distribution of each species is shown in Figure 11

- 20 are indicated by vertical dotted lines. In this figure, it is shown that HCl and ClONO2 rapidly decreased at around day 130 at 87.9°S, when PSCs started to form in the Antarctic polar vortex (Figures 12(e) and 12(f)). The decrease of HCl stopped when the counter-part of the heterogeneous reaction (R1) was missing at around day 140. Consequently, Cl2 was formed (Figure 12(c)). Similar chlorine activation was seen at 76.7°S about 5-10 days later than at 87.9°S. Gradual conversion from Cl2 into Cl2O2 (ClO-dimer) was seen at all latitudes at around day 150-160 (Figures 12(b) and 12(c)) through reactions (R5), (R8), and
- (R9). At 87.9°S, conversion from Cl2 to Cl2O2 was slow, due to lack of sunlight which is needed for reaction (R5). Increase of ClO occurred much later in winter (day 190 or later), because sun light is needed to form ClO by reactions (R5) and (R8) in the polar vortex (Figure 12(d)). Nevertheless, there were some enhancements of ClO in early winter, day 175, simulated at the edge of the polar vortex (Figure 11) where there was some sunlight available due to the distortion of the shape of the polar vortex. Increase of ClO occurred from lower latitude (68.4°S) at around day 195, towards higher latitude (87.9°S) at around
- 30 day 255 (Figure 12(d)). Diurnal variation of CIO was also seen at latitudes between 68.4°S and 76.7°S. When stratospheric temperature increased above PSC saturation temperature at around day 270 (Figure 3(a)), chlorine activation ended, and ClO was mainly converted into HCl at all latitudes inside the polar vortex (Figures 12(d) and 12(e)). This is because reaction (R13) occurs more frequently than reaction (R12) inside the polar vortex due to the depleted O3 amount there as was described in Section 1 (Douglass et al., 1995).

Continuous loss of HCl was seen at 87.9°S between days 160 and 200 even after the disappearance of the counterpart of heterogeneous reaction (R1) (Figure 12(e)). The cause of this continuous loss was unknown until recently, where a hypothesis was proposed that includes the effect of decomposition of particulate HNO3 by some process like ionisation caused by galactic cosmic rays during the winter polar vortex (Grooß et al., 2018). Solomon et al. (2015) proposed a new mechanism on this

5 issue: Continuous transport of CIONO2 from the subpolar regions near 55-65°S to higher latitudes near 65-75°S provides a flux of NOx from more sunlit latitudes into the polar vortex. Our result also shows the same phenomena indicated by some sporadic increase in CIONO2 at around days 158, 179, and 189 at 76.7°S as shown in Figure 12(f). Subsequently, HCl losses were observed at 76.7°S and 87.9°S during these episodes in Figure 12(e). The continuous loss of HCl at the most polar latitude (87.9°S) might be due to the gradual mixing of air within the polar vortex during the winter period, when polar vortex was still

10 strong.

**6. Conclusions**

Lower stratospheric vertical profiles of O3, HNO3, and HCl were retrieved using SFIT2 from solar spectra taken with a groundbased FTIR installed at Syowa Station, Antarctica from March to December 2007 and September to November 2011. This

15 was the first continuous measurements of chlorine species related to the ozone hole from the ground in Antarctica. Retrieved profiles were validated with Aura/MLS and ozonesonde data. The absolute differences between FTIR and Aura/MLS or ozonesonde measurements were within measurement error bars at the altitudes of interest.

To study the temporal variation of chlorine partitioning and ozone destruction from fall to spring in the Antarctic polar vortex, we analyzed temporal variations of measured minor species by FTIR over Syowa Station combined with satellite

- 20 measurements of ClO, HCl, ClONO2 and HNO3. When the stratospheric temperature over Syowa Station fell below PSC saturation temperature, PSCs started to form and heterogeneous reaction between HCl and ClONO2 occurred and ClONO2 was almost completely lost at both 18 km and 22 km in early winter. When the sun came back to the Antarctic in spring, enhancement of ClO and gradual O3 destruction were observed. During the ClO enhanced period, negative correlation between ClO and ClONO2 was observed in the time-series of the data at Syowa Station. This negative correlation is associated with
- 25 the distance between Syowa Station and the inner edge of the polar vortex.

To see the comprehensive behavior of chlorine and related species inside the polar vortex and the boundary region in more detail, results of MIROC3.2 CCM simulation were analyzed. The modeled  $O_3$  and day-to-day variations of HCl and ClONO2 are in good agreement with FTIR and satellite observations. Rapid conversion of chlorine reservoir species (HCl and ClONO2) into Cl2, gradual conversion of Cl2 into Cl2O2, increase of ClO when sunlight became available, and conversion of ClO into

30 HCl were successfully reproduced by the CCM. HCl decrease in the winter polar vortex core continued to occur due to the transport of ClONO2 from the subpolar region to higher latitudes, providing a flux of ClONO2 from more sunlit latitudes into the polar vortex. Temporal variation of chlorine species over Syowa Station was affected both by heterogeneous chemistry

related to PSC occurrence deep inside the polar vortex, and transport of  $NO_x$ -rich airmass from lower latitudinal polar vortex boundary region, which can produce additional ClONO2 by reaction (R12).

The deactivation pathways from active CIO into reservoir species (HCl and/or CIONO2) were found to be very dependent on the availability of ambient  $O_3$ . At an altitude (18 km) where most ozone was depleted in the Antarctic, most CIO was

5 converted to HCl. However, at an altitude (22 km) when there were some O3 available, additional increase of ClONO2 than initial value can occur, like it is the case in the Arctic, through reactions (R14) and (R12) (Douglass et al., 1995).

[revised manuscript text omitted]

  - Douglass, A. R., Schoeberl, M. R., Stolarski, R. S., Waters, J. W., Russell, J. M., III, Roche, A. E., and Massie, S. T.: Interhemispheric differences in springtime production of HCl and ClONO2 in the polar vortices, J. Geophys. Res., 100, 13,967-13,978, 1995.
- 30 Dufour, G., Nassar, R., Boone, C. D., Skelton, R., Walker, K. A., Bernath, P. F., Rinsland, C. P., Semeniuk, K., Jin, J. J., McConnell, J. C., and Manney, G. L.: Partitioning between the inorganic chlorine reservoirs HCl and ClONO2 during the Arctic winter 2005 from the ACE-FTS, Atmos. Chem. Phys., 6, 2355-2366, 2006.
  - Farman, J. C., Gardiner, B. G., and Shanklin, J. D.: Large losses of total ozone in Antarctica reveal seasonal ClOx/NOx interaction, Nature, 315, 207-210, 1985.

[revised manuscript text omitted]

Nakajima, H., Sugita, T., Yokota, T., Ishigaki, T., Mogi, Y, Araki, N, Waragai, K, Kimura, N, Iwazawa, T., Kuze, A, Tanii, J,

30 Kawasaki, H, Horikawa, M, Togami, T, Uemura, N, Kobayashi, H, and Sasano, Y.: Characteristics and performance of the Improved Limb Atmospheric Spectrometer-II (ILAS-II) on board the ADEOS-II satellite, J. Geophys. Res., 111, D11S01, doi:10.1029/2005JD006334, 2006.

- Nakajima, H., Wohltmann, I, Wegner, T., Takeda, M., Pitts, M. C., Poole, L. R., Lehmann, R., Santee, M. L., and Rex, M.: Polar stratospheric cloud evolution and chlorine activation measured by CALIPSO and MLS, and modeled by ATLAS, Atmos. Chem. Phys., 16, 3311-3325, doi:10.5194/acp-16-3311-2016, 2016.
- Nash, E. R., Newman, P. A., Rosenfield, J. E., and Schoeberl, M. R.: An objective determination of the polar vortex using Ertel's potential vorticity, J. Geophys. Res., 101, 9471–9478, 1996.
- Norton, R., and Rinsland, C.: ATMOS data processing and science analysis methods, Appl. Opt., 30, 389-400, 1991.

[revised manuscript text omitted]

20 http://dx.doi.org/10.1029/2002JD002579, 2003.

- Tomikawa, Y., Sato, K., Hirasawa, N., Tsutsumi, M., and Nakamura, T.: Balloon-borne observations of lower stratospheric water vapor at Syowa Station, Antarctica in 2013, Polar Sci., 9, 345–353, doi:10.1016/j.polar.2015.08.003, 2015.
- von Clarmann, T., Glatthor, N., Grabowski, U., Höpfner, M., Kellmann, S., Kiefer, M., Linden, A., Tsidu, G. M., Milz, M., Steck, T., Stiller, G. P., Wand, D. Y., and Fischer, H., Retrieval of temperature and tangent altitude pointing from limb
- emission spectra recorded from space by the Michelson Interferometer for Passive Atmospheric Sounding (MIPAS), J.
   Geophys. Res., 108, 4736, doi:10.1029/2003JD003602, 2003.
  - Waters, J. W., et al.: The Earth Observing System Microwave Limb Sounder (EOS MLS) on the Aura satellite, IEEE Trans. Geosci. Remote Sens., 44, 1075-1092, 2006.
  - Webster, C. R., May, R. D., Toohey, D. W., Avallone, L. M., Anderson, J. G., Newman, P., Lait, L., Schoeberl, M. R., Elkins,
- 30 J. W., and Chan, K. R.: Chlorine chemistry on polar stratospheric cloud particles in the Arctic winter, Science, 261, 1130-1134, 1993.
  - Winker, D. M., McGill, M., and Hunt, W. H.: Initial performance assessment of CALIOP, Geophys. Res. Lett., 34, L19803, doi:10.1029/2007GL30135, 2007.

- WMO: Scientific Assessment of Ozone Depletion: 2006, Global Ozone Research and Monitoring Project—Report No. 50, 572 pp., Geneva, Switzerland, 2007.
- WMO: Scientific Assessment of Ozone Depletion: 2010, Global Ozone Research and Monitoring Project—Report No. 52, 516 pp., Geneva, Switzerland, 2011.
- 5 WMO: Scientific Assessment of Ozone Depletion: 2014, Global Ozone Research and Monitoring Project—Report No. 55, 416 pp., Geneva, Switzerland, 2014.
  - Wood, S. W., Bodeker, G. E., Boyd, I. S., Jones, N. B., Connor, B. J., Johnston, P. V., Matthews, W. A., Nichol, S. E., Murcray, F. J., Nakajima, H., and Sasano, Y.: Validation of version 5.20 ILAS HNO3, CH4, N2O, O3, and NO2 using ground-based measurements at Arrival Heights and Kiruna, J. Geophys. Res., 107, 8208, 10.1029/2001jd000581, 2002.
- 10 Wood, S. W., Batchelor, R. L., Goldman, A., Rinsland, C. P., Connor, B. J., Murcray, F. J., Stephen, T. M., and Heuff, D. N.: Ground-based nitric acid measurements at Arrival Heights, Antarctica, using solar and lunar Fourier transform infrared observations, J. Geophys. Res., 109, D18307, 10.1029/2004jd004665, 2004.
  - Yang, E.-S., Cunnold, D. M., Newchurch, M. J., Salawitch, R. J., McCormick, M. P., Russell III, J. M., Zawodny, J. M., and Oltmans, S. J.: First stage of Antarctic ozone recovery, J. Geophys. Res., 113, D20308, 10.1029/2007JD009675, 2008.
- 15 Ziemke, J. R., Chandra, S., Labow, G. J., Bhartia, P. K., Froidevaux, L., and Witte, J. C.: A global climatology of tropospheric and stratospheric ozone derived from Aura OMI and MLS measurements, Atmos. Chem. Phys., 11, 9237-9251, 10.5194/acp-11-9237-2011, 2011.

**Tables**

| Month     | Dates                             | Dates                   | Number of days | Number of days | Number of days | Number of   |
|-----------|-----------------------------------|-------------------------|----------------|----------------|----------------|-------------|
|           | (2007)                            | (2011)                  | vortex         | region of the  | vortex         | days        |
|           |                                   |                         | (2007/2011)    | polar vortex   | (2007/2011)    | (2007/2011) |
|           |                                   |                         |                | (2007/2011)    |                |             |
| March     | 25                                |                         | 0 / 0          | 0 / 0          | 1 / 0          | 1 / 0       |
| April     | 1, 3, 4, 5, 8, 24, 26, 28         |                         | 0 / 0          | 0 / 0          | 8 / 0          | 8 / 0       |
| May       | 8, 9, 10, 13, 14, 15, 20, 21, 22  |                         | 7 / 0          | 0 / 0          | 2 / 0          | 9 / 0       |
| June      |                                   |                         | 0 / 0          | 0 / 0          | 0 / 0          | 0 / 0       |
| July      | 29, 30                            |                         | 0 / 0          | 2 / 0          | 0 / 0          | 2 / 0       |
| August    | 1, 8, 9, 10, 24, 25, 26, 28, 29   |                         | 8 / 0          | 1 / 0          | 0 / 0          | 9 / 0       |
| September | 1, 4, 5, 6, 7, 8, 16, 18, 23, 26, | 25, 29, 30              | 12/3           | 0 / 0          | 0 / 0          | 12/3        |
|           | 27, 30                            |                         |                |                |                |             |
| October   | 6, 10, 11, 14, 19, 20, 25, 26, 27 | 1, 3, 4, 8, 11, 22, 23, | 9 / 9          | 0 / 0          | 0 / 0          | 9 / 9       |
|           |                                   | 24, 26                  |                |                |                |             |
| November  | 2, 3, 5, 6, 7, 8, 9, 10, 11, 16,  | 1, 2, 3, 9, 11, 16, 19  | 12 / 7         | 1 / 0          | 4 / 0          | 17 / 7      |
|           | 17, 18, 19, 21, 27, 29, 30        |                         |                |                |                |             |
| December  | 4, 7, 8, 9, 13, 15, 16, 17, 20,   |                         | 8 / 0          | 0 / 0          | 3 / 0          | 11 / 0      |
|           | 22, 29                            |                         |                |                |                |             |
| Total     |                                   |                         | 56 / 19        | 4 / 0          | 18 / 0         | 78 / 19     |

 Table 1. FTIR observation dates at Syowa Station in 2007 and 2011

| Species                             | $O_3$                                                                             | HNO 3                                                                            | HCl                                                                  |  |
|-------------------------------------|-----------------------------------------------------------------------------------|---------------------------------------------------------------------------------------------|----------------------------------------------------------------------|--|
| Spectroscopy                        | HITRAN 2008                                                                       | HITRAN 2008                                                                                 | HITRAN 2008                                                          |  |
| PT Profile                          | Daily sonde (0-30 km)                                                             | Daily sonde (0-30 km)                                                                       | Daily sonde (0-30 km)                                                |  |
|                                     | CIRA 86 (30-100 km)                                                               | CIRA 86 (30-100 km)                                                                         | CIRA 86 (30-100 km)                                                  |  |
| A priori profiles                   | Monthly averaged by
ozonesonde (0-30 km)
& ILAS-II (30-100 km)              | Monthly averaged by
ILAS-II                                                              | Monthly averaged by
HALOE                                         |  |
| Microwindows
(cm -1 ) | 1002.578 - 1003.500                                                               | 867.000 - 869.591                                                                           | 2727.730 - 2727.830                                                  |  |
|                                     | 1003.900 - 1004.400                                                               | 872.800 - 874.000                                                                           | 2775.700 - 2775.800                                                  |  |
|                                     | 1004.578 - 1005.000                                                               |                                                                                             | 2925.800 - 2926.000                                                  |  |
| Retrieved
interfering
species | O 3 (668), O 3 (686),
CO 2 , H 2 O | H 2 O, OCS, NH 3 , CO 2 ,
C 2 H 6 | CO 2 , H 2 O, O 3 , NO 2 |  |

**Table 2. Retrieval parameters of SFIT2**

Table 3. Summary of validation results of FTIR profiles compared with ozonesonde and Aura/MLS measurements,and possible Aura/MLS biases from literatures

|                       | D (%) 18-22
km | Min/Max (%)
18-22 km | Agreement
15-25 km
(ppmv/ppbv) | Literature values                                                         |
|-----------------------|-------------------|-------------------------|--------------------------------------|---------------------------------------------------------------------------|
| O 3 | +6.2              | -10.4/+19.2             | -0.02~+0.40                          |                                                                           |
| HNO 3      | +13.2             | +0.2/+21.9              | -0.56~+0.57                          | Aura/MLS no bias with errors (0.6 ppbv) (Livesey et al., 2011)            |
| HCI                   | -9.7              | -14.6/-3.0              | -0.2~+0.09                           | Aura/MLS > HALOE by 10-15%, precision 0.2-0.6 ppbv (Livesey et al., 2013) |

**Table 4. Summary of minor atmospheric species variations**

| Altitude                                             | 18 km   |         | 22 km   |         |
|------------------------------------------------------|---------|---------|---------|---------|
| Year                                                 | 2007    | 2011    | 2007    | 2011    |
| Period of enhanced ClO (day)                         | 230-260 | 230-260 | 220-240 | 230-250 |
| Variation when ClO enhanced (ppbv)                   | 0–1.3   | 0–1.5   | 0–2.2   | 0–2.2   |
| HCl value before winter (ppbv)                       | 1.5–1.8 | 1.2–1.6 | 2.1–2.4 | 1.8–2.2 |
| HCl starting-ending day of decrease (day)            | 140-180 | 140-180 | 130-180 | 140-170 |
| Variation when HCl ~ 0 (ppbv)                        | 0–0.3   | 0-0.3   | 0.1–1.0 | 0.1–0.9 |
| HCl starting-ending day of increase (day)            | 250-300 | 250-300 | 240-280 | 240-300 |
| HCl Value after increase (ppbv)                      | 2.6–3.0 | 2.5–2.8 | 2.1–2.4 | 2.0–2.5 |
| HCl Value outside polar vortex (ppbv)                | 1.5-2.0 | 1.0-1.8 | 1.5-2.0 | 1.5-2.0 |
| ClONO 2 Value before winter (ppbv)        | ~0.5    | ~0.4    | 0.6–0.9 | 0.6–0.7 |
| Variation when ClONO 2 ~0 (ppbv)          | 0–1.5   | 0–1.5   | 0–2.0   | 0–2.0   |
| Day of CIONO 2 enhancement                | -       | 270-300 | 270-280 | 270–280 |
| Value of ClONO 2 enhancement (ppbv)       | -       | 1.5     | 1.5     | 1.5     |
| ClONO 2 value after enhancement (ppbv)    | 0–0.3   | 0-0.2   | 0.8–1.3 | 0.8–1.1 |
| CIONO 2 value outside polar vortex (ppbv) | 0.3–0.4 | 0.2–0.3 | 0.5–0.7 | 0.6–0.8 |
| O 3 value before winter (ppmv)            | 2.5     | 2.5     | 4.0     | 4.0     |
| O 3 starting-ending day of decrease (day) | 190-280 | 200-270 | 170-260 | 170-270 |
| O 3 minimum value (ppmv)                  | 0.3     | 0.5     | 2.0     | 1.0     |
| O 3 value after recovery (ppmv)           | 0.8     | 0.8     | 2.4-4.0 | 2.0–3.5 |
| HNO 3 value before winter (ppbv)          | 6-10    | 8-10    | 15-16   | 13–15   |
| HNO3 starting-ending day of decrease (day)           | 160-190 | 150-180 | 140-180 | 150-180 |
| HNO 3 minimum value (ppbv)                | 0       | 0       | 2       | 1       |
| HNO 3 value after recovery (ppbv)         | 3–4     | 3–4     | 4–6     | 4–5     |